# Laser- and Shock-Induced Droplet Dynamics: A Machine Learning Benchmark for Complex Multiphase Flows

## Abstract

Compressible multiphase flow is central to numerous engineering applications, characterized by complex wave dynamics and challenging shock-interface interactions. Despite their importance, they remain significantly missing from existing benchmarks in the Scientific Machine Learning (SciML) community, limiting progress on generalization to impactful real-world scenarios. To address this issue, we introduce two exemplary datasets from this class, Laser-Induced Droplet Explosion (LIDE) and Shock-Induced Droplet Aero-breakup (SIDA), providing researchers with valuable references to establish reliable baselines and push boundaries of SciML. Due to the high computational cost of simulating these processes with full fidelity, we explore data-driven surrogate models designed to efficiently approximate the underlying physics at reduced cost. We benchmark these datasets on diverse architectures trained autoregressively and compared across varying parameter counts. A comprehensive set of ablations is carried out to analyze the performance of the models. We identify key scenarios, such as incorporating temporal sequence information and conditioning, that enable the models to accurately capture the rich and nonlinear physics embedded in the datasets. Code and datasets will be made available upon acceptance.

## 1 Introduction

Modern technical applications of fluid dynamics exhibit a plethora of flow scenarios involving compressible and multiphase flows, which are characterized by discontinuities across shockwaves and phase boundaries. Gaining insights into the underlying physics of compressible flows is a cornerstone in many real-world systems. These include a wide range of scientific fields, spanning from astrophysics to engineering applications such as coating, fuel injection, biomedical treatment (Chaussy & Schmiedt, 1984), analysis of cavitation phenomena (Maeda et al., 2015), and nanoparticle synthesis (Riahi et al., 2023). Traditionally, domain experts have analyzed these phenomena through simulations and experiments. The downside of these methods is that they demand highly specialized facilities and substantial computational power.

Recent advancements in deep learning algorithms and data-driven modeling (Cai et al., 2021), (Ho et al., 2020), (Lipman et al., 2022), (Kovachki et al., 2023), (Vaswani et al., 2017)), coupled with the rapid growth of modern high-performance computing infrastructures, have accelerated discoveries in Scientific Machine Learning (SciML), enabling robust and reliable surrogate models. However, training these models requires large, multifaceted datasets that capture and correlate spatiotemporal information.

To the best of our knowledge, while datasets exist for either compressible single-phase flows (Takamoto et al., 2022), (Herde et al., 2024) or incompressible multiphase flows (Shadkhah et al., 2025), (Hassan et al., 2023), there is an absence of labeled datasets that capture the complexity of both simultaneously. We address this scarcity by providing two high-fidelity datasets pertaining to liquid droplet dynamics, called **Laser-Induced Droplet Explosion (LIDE)** and **Shock-Induced Droplet Aero-breakup (SIDA)**. This novel set of datasets involves intricate interactions of shocks with interfaces-Richtmyer-Meshkov, Rayleigh-Taylor, and Kelvin-Helmholtz instabilities. It further captures the evolution of multiscale vortical structures and wave dynamics. Therefore, it requires

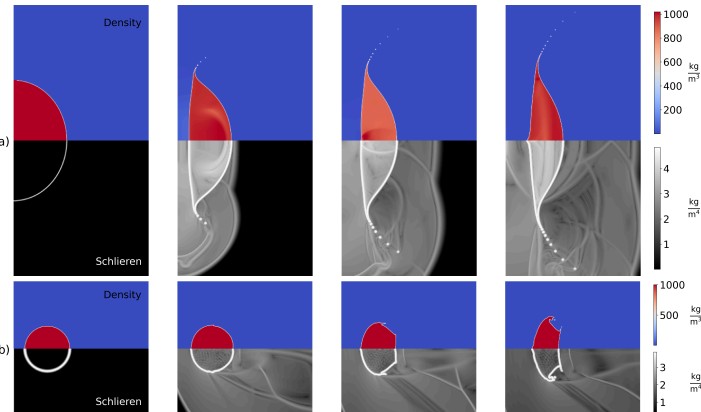

Figure 1: From left to right, Sample trajectory time-snapshots of (a) LIDE and (b) SIDA at: $\{0, \frac{T}{3}, \frac{2T}{3}, T\}$. $T$ denotes the end time of the respective datasets. The frames of the LIDE dataset are rotated clockwise.

profound domain expertise and computational resources, and our contribution lies in introducing this valuable dataset, which paves the way for advancing research in the community. An illustration of two field variables of each dataset is depicted in Figure 1. In LIDE (Paula et al., 2019), an initial high-pressure laser cavity is generated in a micro-droplet. Initiated shock-interface interactions lead to droplet breakup and cavitation events. In SIDA (Kaiser et al., 2020), a shock wave hits a droplet and initiates aero-breakup, where triggered interfacial instabilities generate small liquid fragments through different scenarios.

We propose a many-to-many training strategy (Shadkhah et al., 2025) to benchmark our datasets on a variety of neural architectures, ranging from convolution and spectral models to attention-based approaches. Specifically, we consider UNet, Residual Network (ResNet), Fourier Neural Operator (FNO), Convolutional Neural Operator (CNO), Vision Transformer (ViT), Transolver, alongside Scalable Operator Transformer (ScOT). Furthermore, we identify key parameters and fields with the goal of designing an extensive set of ablations to experiment with the generalization capabilities of the models. Additionally, we compute the metrics over domain-specific quantities of interest on In- and Out-of-Distribution datasets to evaluate baseline models. Although training these models is computationally intensive, once trained, these models are substantially faster when used as a forward simulator. The key contributions of this work are:

- **Datasets for Complex Flow Physics.** A new high-fidelity dataset for complex flow physics involving droplet dynamics and shock-interface interactions is generated and presented.

- **Dataset Validation.** Dataset fidelity is assessed and confirmed by high-resolution simulations and independent experiments.

- **Benchmarking.** A comprehensive set of experiments is performed through side-by-side comparison with different models to gain insights into generalization capabilities.

## 2 RELATED WORK

Existing benchmarks differ in scope and physical coverage. Among them, PDEBench (Takamoto et al., 2022) and the Well (Ohana et al., 2024) offer a wide variety of datasets, including single-phase compressible Navier–Stokes problems, BubbleML (Hassan et al., 2023) and MPF-Bench (Shadkhah et al., 2025) extend to multiphase problems and contribute an impressive collection of bubble and droplet datasets; however, both are limited to incompressible physics. It is noteworthy that Poseidon (Herde et al., 2024) provides an extensive set of datasets to train foundation models, although it considers only single-phase problems. Additionally, BLASTNet(Chung et al., 2023) includes compressible turbulent flow, AIRFRANS (Bonnet et al., 2022) mainly covers incompressible flow over

airfoils, FlowBench(Tali et al., 2024) incorporates fluid flow around arbitrary shapes, and CFD-Bench(Luo et al., 2023) features incompressible and single-phase CFD problems. However, there is no benchmark combining both compressible and multiphase physics in the same setting. Our work addresses this gap by integrating these two characteristics and further incorporates Symmetry, Dirichlet, and Neumann boundary conditions, thereby broadening the diversity of physical scenarios available for SciML research. A summary of the aforementioned references is presented in Table 1.

Table 1: Summary of related datasets.

| Name | Dimensions | Compressible | Multiphase |
|---|---|---|---|
| PDE Bench (Takamoto et al., 2022) | 2 | ✓ | ✗ |
| Poseidon (Herde et al., 2024) | 2 | ✓ | ✗ |
| The Well (Ohana et al., 2024) | 2, 3 | ✓ | ✗ |
| BubbleML (Hassan et al., 2023)(Hassan et al., 2025) | 2, 3 | ✗ | ✓ |
| MPF-Bench (Shadkhah et al., 2025) | 2, 3 | ✗ | ✓ |
| **Current study** | 2-Axisymmetric | ✓ | ✓ |

## 3 DATASETS

We focus on the class of compressible multiphase problems in this paper. Breakup of liquid droplets is a significant example in this class, which can be induced by laser irradiation (LIDE) or a shock (SIDA). These two transient problems are investigated intensely through experiments and numerical simulations. The Robust Discrete Equation Method for Interface Capturing (RDEMIC) (Paula et al., 2023) is used to generate targets through solving the two-dimensional (2D) axisymmetric compressible Euler equations. Adopting an axisymmetric setup reduces computational cost compared to the full three-dimensional treatment. The set of equations, without dissipative terms in vector notation, reads

$$\partial_t \mathbf{U}_l + \nabla \cdot \mathbf{F}_l = \mathbf{B}_l \cdot \nabla \alpha_l + \mathbf{S}_l, \tag{1}$$

where subscript $l$ denotes the index of the phase, $\mathbf{U}_l$ is the vector of conserved quantities, $\mathbf{F}_l$ is the flux tensor, $\mathbf{B}_l$ is the interaction tensor, and $\mathbf{S}_l$ is a source term to account for cylindrical symmetry,

$$\mathbf{U}_l = \begin{bmatrix} \alpha_l \\ \alpha_l \rho_l \\ \alpha_l \rho_l \mathbf{u}_l \\ \alpha_l E_l \end{bmatrix}, \quad \mathbf{F}_l = \begin{bmatrix} 0 \\ \alpha_l \rho_l \mathbf{u}_l^T \\ \alpha_l \rho_l \mathbf{u}_l \otimes \mathbf{u}_l + \alpha_l p_l \mathbf{I} \\ \alpha_l (E_l + p_l) \mathbf{u}_l^T \end{bmatrix}, \quad \mathbf{B}_l = \begin{bmatrix} -\mathbf{u}_{\text{int}}^T \\ 0 \\ p_{\text{int},l} \mathbf{I} \\ p_{\text{int},l} \mathbf{u}_{\text{int}}^T \end{bmatrix}, \quad \mathbf{S}_l = -\frac{\alpha_l u_{r,l}}{r} \begin{bmatrix} 0 \\ \rho_l \\ \rho_l \mathbf{u}_l \\ E_l + p_l \end{bmatrix}.$$

Above, $\alpha_l$, $\rho_l$, $\mathbf{u}_l$, $p_l$, $u_{r,l}$, and $E_l$ imply the volume fraction, mass density, velocity vector, pressure, velocity component in the radial direction, and total energy of phase $l$, respectively. Interface velocity vector and pressure are indicated by $\mathbf{u}_{\text{int}}$ and $p_{\text{int},l}$, respectively; without considering the surface tension, $p_{\text{int},l}$ is the same for all phases; $r$ denotes the distance from the symmetry axis and $\mathbf{I}$ is the identity tensor. This method is implemented and validated extensively through the Finite Volume solver, ALPACA (Hoppe et al., 2022). In cylindrical coordinate configuration, the domain revolves around the z-axis (south, as shown in Figure 2), resulting in an axisymmetric problem. In the following sections, a brief overview of each dataset is given. For more details, refer to Appendix A.

### 3.1 LASER-INDUCED DROPLET EXPLOSION (LIDE)

Experimental investigations of LIDE provide a valuable insight into pure liquid states and pressure-sensitive molecular dynamics in solutions (Stan et al., 2016a). When a laser pulse hits the transparent liquid droplet, energy is deposited within nanoseconds, forming a high-pressure filament along the laser trajectory. This induces shock and expansion waves, which are reflected and subsequently generate negative pressure waves inside the droplet. Consequently, the droplet undergoes deformation

and eventually ruptures if the tension is strong enough. Notably, the negative pressure at rupture is related to the tensile strength that the liquid can sustain during decompression (Stan et al., 2016b).

This problem is also numerically addressed in literature (Paula et al., 2019). Taking advantage of the symmetries, a droplet with radius $R_0$ is located in the bottom left corner of a square domain with length $3R_0$, as shown in Figure 2a. The filament, heated by the laser beam along the centerline, is also illustrated. The boundary conditions (BC) are Symmetry (west) and Zero-gradient (east and north). The latter refers to a special case of Neumann BC, where the normal derivative of the field variable at the boundary is set to zero. To explore the dynamics of the explosion, we vary the values for filament pressure, ambient pressure, laser half-width, and the droplet radii along perpendicular axes, which distinguishes spherical from ellipsoidal geometries. The aforementioned parameters are subsequently used as conditioning parameters during training. More details on the initial condition values are described in Appendix A.1.

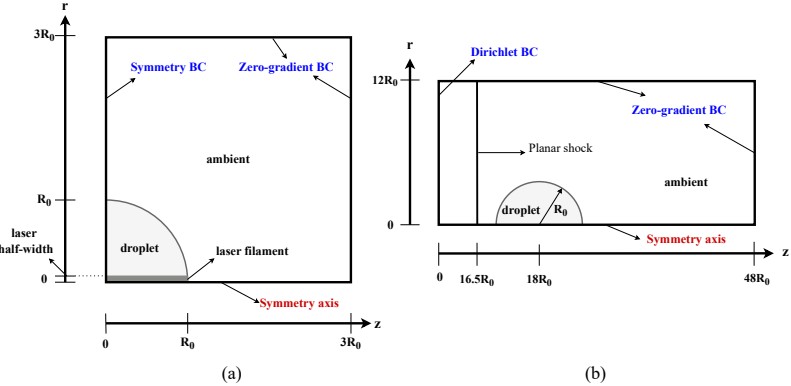

Figure 2: Initial setup for (a)LIDE and (b)SIDA.

### 3.2 SHOCK-INDUCED DROPLET AERO-BREAKUP (SIDA)

The droplet aero-breakup, which is caused by the sudden exposure of liquid droplets to external flow, is relevant in practical applications of fuel injection and shock-tube flow (Liang et al., 2020). The resulting shock–droplet interaction involves the evolution of reflected, transmitted, and diffracted waves, along with droplet displacement, deformation, and the development of surface instabilities. This high-speed phenomenon requires high spatiotemporal resolution to be accurately captured. Surface tension has a strong impact on the droplet breakup mode, which is characterized by the Weber number (Hinze, 1955). This non-dimensional parameter accounts for the relative dominance of aerodynamic force over surface tension. Furthermore, the external flow regime, from subsonic to supersonic, is governed by the Mach number (Kaiser et al., 2020).

Initially, we simulate the SIDA dataset in a domain of size $[48R_0, 12R_0]$, which is shown in Figure 2b. This large domain is essential to avoid undesirable boundary effects regarding wave dynamics. However, a fixed subdomain with size $[6R_0, 3R_0]$ around the droplet is saved and later used in training. This subdomain is chosen such that in the initial timestep, the shock wave is located at the west end.

Boundary conditions include Dirichlet (west) and Zero-gradient (east and north). This dataset is generated with various combinations of Mach and Weber numbers, which are later utilized as conditioning parameters in model training (Meng & Colonius, 2018), (Winter et al., 2019). More details on the initial condition values and the validation of the dataset are described in Appendix A.2.

### 3.3 DATASETS VALIDATION

For LIDE, we compare the expansion of the droplet outer diameter in the radial direction to validate our dataset against experiments (Stan et al., 2016b). The consistency with reference is shown in Figure 3 (a). For SIDA, the non-dimensional time ($t^*$) and displacement of the center of mass

(COM) in the droplet ($\Delta z^*$) are defined as $t^* = t \frac{u_2}{d} \sqrt{\frac{\rho_2}{\rho_{drop}}}$ and $\Delta z^* = \frac{z}{d}$, where $u_2$ and $\rho_2$ are post-shock flow velocity and density, $\rho_{drop}$ and $d$ are droplet density and diameter, and $t$ is the saved timestep, respectively. The displacement of the droplet COM aligns against literature in Figure 3 (b). More details on the dataset validation are described in Appendix A.

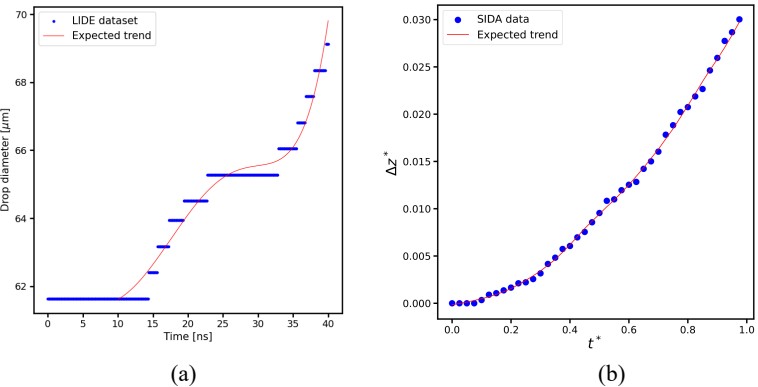

(a)  (b)

Figure 3: Dataset validation of (a) LIDE (Stan et al., 2016b) (b) SIDA (Winter et al., 2019).

## 3.4 METADATA

Each dataset[1] includes 128 trajectories, and the splitting for training/validation/inference is 86/10/32. In total, 6 fields are made available for each dataset, where density, pressure, X-velocity, Y-velocity, and schlieren are common in both datasets. The remaining channel is the total energy for LIDE, and vorticity for SIDA. The spatiotemporal parameters used in the numerical solver are presented in Table 2. The datasets are stored as HDF5 files, with sizes of 75 GB and 12 GB for LIDE and SIDA, respectively, and the shapes for both are **[num_of_trajectories][num_of_timesteps][fields][X-resolution][Y-resolution]**. Each trajectory in the dataset file is assigned a unique group name based on its corresponding conditioning parameters. Sample Out-of-Distribution (OOD) datasets with a similar size to the test part of the In-Distribution (ID) data are considered to assess the generalisation capabilities of trained ML models. For OOD LIDE, higher pressures within the laser width, and for OOD SIDA, higher shock Mach numbers are considered. Details on the parameter ranges considered to generate ID and OOD datasets are provided in Appendix A.

Table 2: Metadata for LIDE and SIDA datasets

| Dataset | Resolution [X, Y] | End time [s] | CFL[2] | $\Delta t_{\text{solver}}$ [s][3] | $\Delta x$ [m] | # trajectories | # timesteps |
|---------|-------------------|--------------|--------|-----------------------------------|----------------|----------------|-------------|
| LIDE | [256, 256] | $20 \times 10^{-9}$ | 0.35 | $6.80 \times 10^{-12}$ | $1.25 \times 10^{-7}$ | 128 | 201 |
| SIDA | [256, 128] | $15 \times 10^{-6}$ | 0.50 | $1.95 \times 10^{-9}$ | $1.17 \times 10^{-5}$ | 128 | 61 |

---

[1]The uploaded supplementary material as a .zip file includes metadata.json files for each LIDE and SIDA dataset. Also, sample video files for ID and OOD datasets are provided for visualization.

[2]CFL refers to Courant-Friedrichs-Lewy criterion.

[3]This is the average solver timestep among all trajectories.

# 4 EXPERIMENTS

## 4.1 DESIGN OF EXPERIMENTS

This section outlines the Design of Experiments (DOE). Each experiment is assigned a unique tag for easier identification and comparison. We use **'P'** for Pressure, **'D'** for Density, **'U'** for X-Velocity, **'V'** for Y-Velocity, **'E'** for Energy, **'S'** for Schlieren, and **'Vo'** for Vorticity. For example, an experiment with a tag 'PDUV[ES]_T_(3,2)' implies the input channels are Pressure (P), Density (D), X-Velocity (U), Y-Velocity (V), Energy (E), and Schlieren (S). '[ES]' shows that Energy and Schlieren are counted as conditioning fields and are not predicted in the output. Furthermore, 'T' indicates that the conditioning parameters are included in the experiment. Finally, '(3,2)' corresponds to 3 consecutive inputs and 2 consecutive predicted frames. The complete DOE table is provided in the Appendix B.1.

## 4.2 BASELINE MODELS

We investigate the performance of the datasets on a variety of neural architecture baselines, The models under consideration are: UNet (Ronneberger et al., 2015), ResNet (He et al., 2016), FNO (Li et al., 2020), CNO (Raonic et al., 2023), ViT (Dosovitskiy et al., 2020), Transolver (Wu et al., 2024), and ScOT (Herde et al., 2024). Each model was trained from scratch on two parameter categories, i.e., 1M and 50M. However, ResNet and Transolver are trained only with 1M model parameters due to their high computational requirements for the 50M size. For more details on model hyperparameters, refer to Appendix B.2.

## 4.3 INVESTIGATION SCENARIOS

We analyze our results by categorizing the experiments into three distinct scenarios. Each scenario addresses a certain learning problem, and experiments are grouped by altering only the learning parameter while holding all other parameters fixed. The following subsections give a brief overview of these learning problems.

### 4.3.1 TEMPORAL CONTEXT

Historic information, provided through additional temporal inputs (frames), has proved its efficacy (Hassan et al., 2023), (Shadkhah et al., 2025). In some experiments, to facilitate the understanding of the patterns, we incorporate multiple frames into the model. This provision is effective in learning transient trajectories. For both datasets, we experiment with either 1 input or including a sequence of 3 historic inputs. We also define a stride parameter during dataloading, which skips a fixed number of timesteps. In the LIDE and SIDA datasets, strides of 10 and 5 timesteps are employed, respectively.

### 4.3.2 CONDITIONING PARAMETERS

In many fluid dynamic problems, the physics are fundamentally characterized by non-dimensional and domain parameters, which influence the system's evolution. These provide crucial information as they dictate the governing dynamics, leading to distinct flow regimes. Conditioning the model with such parameters improves generalization (Kohl et al., 2023), (Peebles & Xie, 2023). The conditioning parameters for the LIDE and SIDA datasets are mentioned in section 3. These are injected into the models through the normalization layers (Herde et al., 2024). More details on the implementation are provided in Appendix B.3.

### 4.3.3 CONDITIONING FIELDS

In this experimental scenario, additional channels are appended to the inputs before passing them to the model. These extra channels are called conditioning fields, which are derived quantities from existing inputs. For the LIDE dataset, we incorporate energy and schlieren as the conditioning fields, whereas for the SIDA dataset, vorticity and schlieren are used. We aim to test the hypothesis that this type of conditioning guides the model towards generalization.

## 4.4 TRAINING AUTOREGRESSIVE MODELS

In this work, we use a many-to-many training style to train each of our baselines, $\mathcal{M}_\theta$. The dataset is a discrete spatiotemporal system, containing $c$ channels. For a particular trajectory, the mapping is given by $\mathbf{X}_t : \Omega \times [0, T] \rightarrow \mathbb{R}^c$, where $\Omega \subset \mathbb{R}^2$ and $T$ represents the last timestep of the trajectory.

During training, we split each of the training trajectories into $M$ windows. The length of each window is determined by the number of input and output sequences, denoted by $l_1$ and $l_2$, respectively, and $s$ denotes the stride, which are all hyperparameters of the temporal context study as mentioned in section 4.3.1.

The input sequence of the $m^{th}$ window is given by $\mathbf{X}_m = [\mathbf{X}_m, \dots \mathbf{X}_{m+(l_1 \times s)}] \in \mathbb{R}^{l_1 \times c}$ and the corresponding target is $\mathbf{Y}_m = [\mathbf{X}_{m+((l_1+1) \times s)} \dots \mathbf{X}_{m+((l_1+l_2) \times s)}] \in \mathbb{R}^{l_2 \times c}$. The training loss reads:

$$\text{MSE} := \frac{1}{M} \sum_{m=1}^{M} \| \mathcal{M}_\theta (\mathbf{X}_m) - \mathbf{Y}_m \|^2, \tag{2}$$

After each training epoch, the validation loss is computed by rolling out the model autoregressively for 5 steps and then computing the Root Mean Square Error (RMSE) over the output channels.

## 4.5 INFERENCE METRICS

During inference, we start from the initial condition of each trajectory and rollout the model in an autoregressive fashion to reach the final frame. The predictions across trajectories are accumulated into a tensor $(\hat{Y})$, and normalized with the target $(Y)$. The normalized Root Mean Square Error (nRMSE) metric is obtained as shown in Equation 3. **This metric is referred to as error-type 1.**

$$\text{nRMSE} = \sqrt{\frac{\sum_{\text{all dims}} (Y - \hat{Y})^2}{\sum_{\text{all dims}} Y^2 + \epsilon}} \tag{3}$$

We define an **error-type 2** starting from the initial frame and performing rollout until the end of the sequence. The model prediction and target tensors are of shape (N, R, T, C, spatial-dims), where N is the number of test trajectories, R is the number of rollout steps, T is the number of output timesteps per rollout step, C is the number of output channels, and spatial-dims is the resolution of the dataset. The error aggregation is performed in four stages:

1. In each trajectory, we first compute nRMSE between the prediction and the target tensor over the T, C, and spatial-dims, resulting in an overall tensor with shape (N, R).

2. We compute the cumulative summation along the rollout dimension, retaining the shape (N, R).

3. We compute mean and standard deviation across trajectories (N), which results in a tensor of shape (R).

4. Finally, we reduce across the rollout dimension to obtain the overall mean and standard deviation. **We denote this metric as error-type 2**.

In addition to the metrics over the output channels, the aforementioned error types are computed for the domain-specific quantities of interest (QoI) to better understand the underlying physics of the problem, namely, evolution of Kinetic Energy (KE) and Vorticity Production (VP). These QoIs are defined in Appendix B.5. Dataset-specific QoIs over the rollout for LIDE and SIDA are the change of droplet outer radius (OR) and the displacement of droplet center of mass (COM), respectively, over time.

## 5 RESULTS

### 5.1 EFFECT OF SEQUENCE INFORMATION

Within the many-to-many autoregressive training framework, we evaluate three configurations of sequence information: (1,1), (3,1), and (3,2), corresponding to one input–one prediction, three in-

puts–one prediction, and three inputs–two predictions, respectively. For both the LIDE and the SIDA datasets, we observe a consistent performance improvement across all models trained with three historical timesteps, with the single-prediction models having a slight metric advantage over the two-consecutive-prediction models. A further gain in accuracy is obtained upon increasing the parameter count, with UNet performing the best. These results for the SIDA are tabulated in Table 32. The results for the LIDE are shown in Table 19 and 20 in the Appendix C. As expected, the model's performance degrades on the OOD datasets; however, including temporal context improves its performance compared to single-frame inputs. The complete set of tables for both datasets and for all QoIs is presented in Appendix C. Moreover, Appendix D presents the cumulative nRMSE trends for both in-distribution and out-of-distribution datasets. Over longer rollouts, UNet consistently outperforms the other models for most QoIs, establishing itself as a robust baseline.

Table 3: Effect of sequence information for SIDA In-Distribution (ID) and Out-of-Distribution (OOD) datasets. Error-type 2 over output channels from section 4.5 is presented.

| MODEL | TAG | 1M | | 50M | |
|---|---|---|---|---|---|
| | | ID | OOD | ID | OOD |
| UNet | PDUV_F_(1,1) | 3.5661±2.4696 | 5.9914±3.2792 | 3.4983±2.5270 | 5.9438±3.2522 |
| | PDUV_F_(3,1) | **0.3383±0.2089** | 2.2998±1.7754 | **0.1356±0.0976** | 1.9424±1.5948 |
| | PDUV_F_(3,2) | 0.4406±0.2437 | **2.1979±1.6703** | 0.1785±0.1080 | **1.8812±1.4977** |
| FNO | PDUV_F_(1,1) | 3.5224±2.5528 | 6.0820±3.2896 | 3.4720±2.5577 | 5.9768±3.2572 |
| | PDUV_F_(3,1) | **0.5333±0.2829** | **2.1215±1.5319** | **0.2498±0.1387** | **2.0963±1.5242** |
| | PDUV_F_(3,2) | 0.7123±0.3624 | 2.1449±1.5747 | 0.3389±0.1827 | 2.0990±1.5458 |
| ViT | PDUV_F_(1,1) | 4.4758±2.8750 | 5.8219±3.1222 | 3.5527±2.7160 | 5.7645±3.1503 |
| | PDUV_F_(3,1) | 1.5891±0.9548 | 3.4253±2.4145 | **0.4430±0.2336** | **1.8428±1.3876** |
| | PDUV_F_(3,2) | **1.3832±0.7495** | **2.4443±1.6647** | 0.5441±0.2699 | 1.8678±1.3767 |
| ScOT | PDUV_F_(1,1) | 3.5764±2.5158 | 5.8985±3.2421 | 3.5173±2.6462 | 5.8306±3.1826 |
| | PDUV_F_(3,1) | **0.6884±0.4222** | **1.9005±1.3770** | **0.2059±0.1199** | 1.6318±1.3494 |
| | PDUV_F_(3,2) | 0.7995±0.4265 | 1.9084±1.3867 | 0.2639±0.1459 | **1.5164±1.2414** |
| CNO | PDUV_F_(1,1) | 3.9955±2.6786 | 5.8508±3.1777 | 3.5025±2.5455 | 5.9444±3.2344 |
| | PDUV_F_(3,1) | **0.6943±0.4235** | 2.1707±1.8099 | **0.1621±0.1032** | 1.8159±1.4794 |
| | PDUV_F_(3,2) | 0.8584±0.4459 | **1.8360±1.3758** | 0.2210±0.1248 | **1.7634±1.3875** |
| ResNet | PDUV_F_(1,1) | 3.7260±2.4574 | 6.0844±3.3406 | | |
| | PDUV_F_(3,1) | **0.5570±0.3461** | 2.3042±1.7139 | | |
| | PDUV_F_(3,2) | 0.6356±0.3669 | **2.3012±1.6856** | | |
| Transolver | PDUV_F_(1,1) | 5.7995±3.4502 | 6.9499±3.7154 | | |
| | PDUV_F_(3,1) | **4.7065±2.6888** | **4.8852±2.7289** | | |
| | PDUV_F_(3,2) | 5.2026±2.7770 | 5.1580±2.7698 | | |

## 5.2 EFFECT OF CONDITIONING PARAMETERS

We conduct several studies to assess if including conditioning parameters as described in section 4.3.2 has a pronounced influence on the inference metrics. For the LIDE dataset, we observe that embedding these parameters into the baselines generally has a positive impact, whereas metrics deteriorate for the SIDA dataset, as shown in Table 4 and Table 36, respectively. It is worth noting that the characteristics of the conditioning parameters in the SIDA dataset are different from those of the LIDE dataset. In the former, the parameters are geometry-based, and for the latter, these are flow properties. Furthermore, we observe that conditioning parameters substantially enhance the prediction in the single-input single-output experiments, motivating their adoption in scenarios where generating temporal context windows poses challenges. The complete set of tables for both datasets and for all QoIs is presented in Appendix C.

Table 4: Effect of conditioning parameters for LIDE In-Distribution (ID) and Out-of-Distribution (OOD) datasets. Error-type 2 over output channels from section 4.5 is presented.

| MODEL | TAG | 1M | | 50M | |
|---|---|---|---|---|---|
| | | ID | OOD | ID | OOD |
| UNet | PDUV_F_(1,1) | 2.9972±2.4755 | **9.0795±5.9229** | 2.1209±1.8598 | **8.0827±5.3407** |
| | PDUV_T_(1,1) | **2.9636±2.5183** | 10.1979±7.5259 | **1.2553±1.1437** | 8.1129±5.4866 |
| | PDUV_F_(3,1) | **1.5341±1.2526** | **6.0177±5.0873** | **1.0029±0.8511** | **5.0866±4.2680** |
| | PDUV_T_(3,1) | 2.6413±2.3683 | 9.6531±7.8398 | 1.0383±0.9354 | 6.6239±5.2895 |
| FNO | PDUV_F_(1,1) | 3.6399±3.1425 | **7.8868±5.3663** | 2.5836±2.2495 | **7.2695±5.3881** |
| | PDUV_T_(1,1) | **2.7562±2.4957** | 9.1299±6.1320 | **1.6632±1.4450** | 8.7610±6.3761 |
| | PDUV_F_(3,1) | 2.4656±2.4281 | **5.9441±4.8157** | 1.6334±1.4492 | **6.4107±5.4997** |
| | PDUV_T_(3,1) | **2.1101±2.2230** | 6.4283±5.1802 | **1.5349±1.2236** | 7.3136±5.1163 |
| ViT | PDUV_F_(1,1) | 6.7641±4.8388 | **8.5556±5.3823** | 2.1804±1.7873 | 6.3454±4.3554 |
| | PDUV_T_(1,1) | **4.6571±3.6689** | 11.1820±12.6204 | **1.0744±0.7655** | **6.0053±5.3625** |
| | PDUV_F_(3,1) | 4.1118±3.0304 | **6.4825±5.1926** | 1.4109±1.1031 | **3.9230±3.2763** |
| | PDUV_T_(3,1) | **3.4518±2.4023** | 10.4698±11.1001 | **1.0612±0.7537** | 4.4118±4.0008 |
| ScOT | PDUV_F_(1,1) | 2.8022±2.2122 | **7.4147±4.9714** | 2.2992±1.8961 | **7.0310±4.6946** |
| | PDUV_T_(1,1) | **2.1729±1.6478** | 15.8824±16.0901 | **1.0326±0.8473** | 10.2492±10.8710 |
| | PDUV_F_(3,1) | **1.7180±1.3379** | **6.4316±7.3418** | 1.1854±0.9577 | **5.1242±5.4815** |
| | PDUV_T_(3,1) | 1.7619±1.3709 | 8.0149±6.9873 | **0.8772±0.6840** | 7.8435±7.3701 |
| CNO | PDUV_F_(1,1) | **2.1786±1.7884** | **7.7764±6.8129** | 2.1271±1.7921 | **6.7606±4.9076** |
| | PDUV_T_(1,1) | 2.6620±2.2500 | 10.6073±7.5735 | **2.0716±1.8737** | 9.5014±6.0985 |
| | PDUV_F_(3,1) | **1.2258±0.9966** | **6.4241±6.3085** | **1.1850±0.9985** | **4.7852±3.9415** |
| | PDUV_T_(3,1) | 2.6061±2.4252 | 9.0774±6.4503 | 1.8087±1.6317 | 8.0127±5.4545 |
| ResNet | PDUV_F_(1,1) | 7.1277±5.7086 | 9.8022±6.0704 | | |
| | PDUV_T_(1,1) | **4.8602±3.8324** | **9.1651±5.5656** | | |
| | PDUV_F_(3,1) | 5.0517±4.3040 | **6.4456±4.5672** | | |
| | PDUV_T_(3,1) | **3.3461±2.5509** | 8.8525±6.2045 | | |
| Transolver | PDUV_F_(1,1) | **4.9459±3.6294** | **8.3506±5.2692** | | |
| | PDUV_T_(1,1) | 8.3954±5.2724 | 8.7039±5.2462 | | |
| | PDUV_F_(3,1) | **3.4782±2.3452** | **5.5601±4.1509** | | |
| | PDUV_T_(3,1) | 4.9456±3.2063 | 8.5088±5.8755 | | |

## 5.3 EFFECT OF CONDITIONING FIELDS

Considering the selected conditioning fields for each dataset, as described in Section 4.3.3, we conclude that, across all models and parameter counts, incorporating these fields, in general, degrades the predictions, resulting in increased errors during inference. The complete set of tables for both datasets and for all QoIs is presented in Appendix C.

## 5.4 BASELINE MODEL PERFORMANCE STUDY

We investigate error-type 1 (section 4.5) in baseline models on an identical experiment for each dataset. As a sample experiment, we present Table 5, which shows that a higher parameter count improves the prediction accuracy across all models. UNet consistently achieves superior performance compared to all the other baselines in both the 1M and 50M categories. Remaining tables for In-Distribution (ID) and Out-of-Distribution (OOD) datasets are available in Appendix E.

## 5.5 COMPARISON BETWEEN ERROR TYPES

We compare the two error types, **defined in section 4.5**, to correlate the metrics with the predicted rollout. It is worth emphasizing that from our ablations, error-type 2 demonstrates better coherence

Table 5: Error-type 1 from section 4.5 for experiment PDUV_F_(3,1) for the LIDE In-Distribution (ID) dataset across all 1M and 50M models.

| MODEL | 1M | | | | 50M | | | |
|---|---|---|---|---|---|---|---|---|
| | output channels | OR | KE | VP | output channels | OR | KE | VP |
| CNO | 0.1641 | 0.7208 | 0.1182 | 0.2360 | 0.1658 | 0.7696 | 0.0995 | 0.1413 |
| FNO | 0.3814 | 0.7405 | 1.5689 | 2.0840 | 0.2538 | 0.7069 | 0.7270 | 0.9354 |
| ScOT | 0.2234 | 0.4091 | 0.3476 | 0.3303 | 0.1570 | 0.8316 | 0.1831 | 0.1294 |
| UNet | 0.2140 | 0.3373 | 0.1471 | 0.1766 | 0.1383 | 0.3408 | 0.0601 | 0.0677 |
| ViT | 0.5237 | 0.8853 | 1.5604 | 1.3256 | 0.1792 | 0.8206 | 0.2179 | 0.1896 |
| ResNet | 0.7244 | 1.0878 | 0.2947 | 1.4805 | – | – | – | – |
| Transolver | 0.4091 | 1.2833 | 0.7547 | 0.4055 | – | – | – | – |

with predicted rollouts in some cases. For example, as shown in Table 6, UNet-50M achieves higher accuracy according to error-type 2 compared to ViT-50M; UNet captures the droplet interface more precisely, indicating better performance as a surrogate relative to ViT. The corresponding plot is available in Figure 54 in Appendix F. In contrast, error-type 1 suggests that ViT predicts better. This discrepancy highlights the importance of selecting an error metric that aligns with the qualitative behavior observed in rollout plots (Luo et al., 2023). Sample rollout prediction plots, during inference, for the LIDE-ID and the SIDA-ID datasets are shown in Appendix F.

Table 6: Error-type 1 and 2 over output channels from section 4.5 for the experiment PDUV_T_(3,1) for the LIDE In-Distribution (ID) dataset across models with 50M parameters.

| MODEL | Error type 1 | Error type 2 |
|---|---|---|
| UNet | 0.1407 | 1.0383 |
| FNO | 0.1969 | 1.5349 |
| ViT | 0.1293 | 1.0612 |
| ScOT | 0.1111 | 0.8772 |
| CNO | 0.2463 | 1.8087 |

# 6 CONCLUSION

This study presents two novel datasets in the domain of compressible multiphase fluid dynamics. We benchmarked seven baseline models on these datasets with varying parameter counts. Domain-specific quantities of interest (QoIs) are considered, and our study scenarios explore the influence of historical information, conditioning parameters, and fields on all QoIs. The inference results of the trained baseline models on both the LIDE and SIDA datasets showed superior prediction accuracy upon incorporating additional temporal context. Subsequently, introducing additional channels as conditioning fields to the input degraded the prediction accuracy during inference on both datasets. Furthermore, injecting conditional parameters into the baselines yielded bifurcating results for the datasets. Despite poor performance on the SIDA dataset, models show better accuracy on the LIDE dataset. Finally, we examined the interpretation of two error types and their correlation with the rollout plots, which illustrates the importance of selecting a suitable error metric in choosing an appropriate surrogate. In conclusion, it is essential to highlight that representing the complex physics and patterns through the current datasets by surrogates still poses a challenge. This observation motivates the integration of such datasets in the SciML community to further the development of data-driven surrogates.

**Limitations and Future works.** Extending the dataset diversity to include multi-droplet and bubble scenarios, analyzing different combinations of conditioning fields and parameters, and advancing toward more effective conditioning algorithms are promising future directions.

# 7 REPRODUCIBILITY STATEMENT

We introduce two datasets in this paper, which are reproducible based on our description in the main text (section 3) and supplements in the Appendix A. These explanations include the referenced Finite Volume solver, numerical setup, and initial conditions. More details on the solver code and generator scripts for creating datasets are provided in the HuggingFace repository. In addition, for reproducing model evaluations, we will provide the datasets, the benchmarking code, and the trained model weights upon acceptance.

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

## A  DATASETS

As mentioned in the main text, we solve the compressible multiphase Euler equations (Equation 1) with the RDEMIC, which captures the interface as a diffuse zone on a Cartesian grid. This method combines the solutions of pairwise Riemann problems to obtain the finite-volume flux. By a modified partitioning of the Riemann solutions and a specific combination of fluxes and non-conservative terms, the method is made practically applicable for high-resolution shock-interface problems (Paula et al., 2023). We use this method in ALPACA (Hoppe et al., 2022), which is a well-suited environment for compressible single-phase simulations and other multi-phase methods, although originally developed as a level-set-based sharp-interface solver. Its standout features include a wide variety of Riemann solvers, high-resolution reconstruction schemes, and a state-of-the-art multiresolution algorithm for high computational efficiency. In both datasets in this study, the cell face fluxes are reconstructed with the fifth-order WENO scheme (Jiang & Shu, 1996). Furthermore, a third-order Runge-Kutta Total Variation Diminishing scheme is applied for time discretization (Gottlieb & Shu, 1998).

To close the governing equation (Equation 1), an Equation of State (EOS) is used, which relates pressure to density and internal energy. We adopt the stiffened-gas EOS to generate both datasets, which reads

$$p(\rho, e) = (\gamma - 1)\rho e - \gamma p_{\text{stiff}} \iff e(\rho, p) = \frac{p + \gamma p_{\text{stiff}}}{(\gamma - 1)\rho}, \tag{4}$$

with $p$ being the pressure of the fluid, $\rho$ the mass density, $e$ the internal energy, $\gamma$ the model constant. In addition, $p_{\text{stiff}}$ accounts for a pre-compression of the fluid. To degenerate the aforementioned equation to an ideal-gas EOS for air, we adopt $\gamma = 1.4$ and $p_{\text{stiff}} = 0$. The total energy density, $E[\frac{J}{m^3}]$, is obtained by considering internal energy from Equation 4 and kinetic energy, as shown in Equation 5:

$$E = \rho e + 1/2\rho(u_r^2 + u_z^2) \tag{5}$$

Here, $u_r$ and $u_z$ are the velocity components in the $r$ and $z$ directions, respectively. Schlieren $[\frac{kg}{m^4}]$ is computed in the solver by Equation 6:

$$\text{schlieren} = \nabla\rho \tag{6}$$

Additionally, vorticity $[s^{-1}]$ is defined in Equation 7:

$$\text{vorticity} = \nabla \times \mathbf{u} \tag{7}$$

### A.1  THE LIDE DATASET

To simulate this problem, careful considerations must be taken into account. The filament along the centerline, which is heated by a laser in a very short time, is pre-initialized with vapor instead of liquid water. However, it is important to note that the density of the vapor in this zone remains equal to that of liquid water, since the laser energy heats the liquid rapidly. A summary of initial condition values is presented in Table 7.

Table 7: Initial conditions for the LIDE dataset

| Phase-l | $\rho_l$ [kg m$^{-3}$] | $\{u_{r,l}, u_{z,l}\}$ [m s$^{-1}$] | $p_l$ [Pa] | |
|---|---|---|---|---|
| 1 (Ambient air) | 0.74 | 0.0, 0.0 | $p_{\text{ambient}}$ | |
| 2 (Liquid droplet) | 998.2 | 0.0, 0.0 | $p_{\text{ambient}}$ | $z >$ laser half-width |
| | 998.2 | 0.0, 0.0 | $p_{\text{filament}}$ | $z <$ laser half-width |

**Validation.** We compare the evolution of the droplet diameter in the radial direction to validate our dataset against experiments (Stan et al., 2016b). According to experimental observations, the droplet starts to expand upon the arrival of the radial shock wave, which is induced by high pressure in the filament. Due to the wave interactions, a decrease in the expansion rate is observed, which is again followed by an increase. This trend is depicted in Figure 4 and is in good agreement with experiments.

In this problem, it is crucial to analyze and understand the wave interactions inside the droplet. After rapid energy deposition along the centerline, the main shock spreads radially, approaching the droplet surface. The corresponding reflection results in a curved negative-pressure wave, which increases the tension. Shortly after, this wave collapses toward the z-axis and impacts the motion of the droplet's surface (Paula et al., 2019). These phenomena are depicted step-by-step from top to bottom in Figure 5.

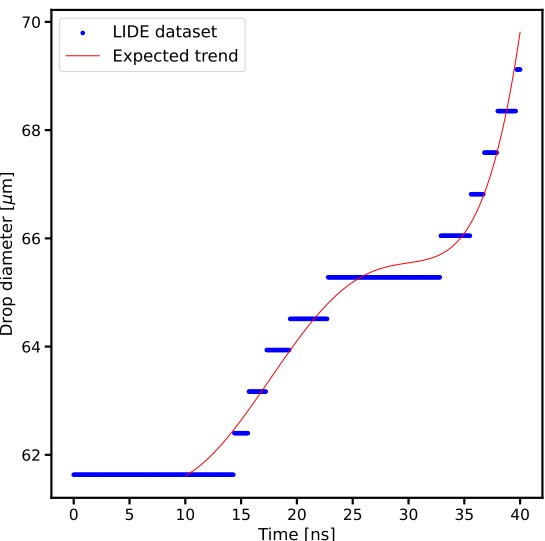

Figure 4: Validation of the LIDE data compared to the expected trend (Stan et al., 2016b).

**Parameter Range**. Considering that different laser pulse energies result in different pressures in the filament ($p_{filament}$), we cover a range from $10^8$ to $10^{10}$ [Pa] in our dataset. Alongside the high-pressure, the ambient pressure ($p_{ambient}$) varies between $10^5$ and $10^6$ [Pa]. In addition, the laser half-width changes in the range of $2 \times 10^{-7}$ to $1.5 \times 10^{-6}$ [m]. The droplet radius along the r and z axes varies from $1 \times 10^{-5}$ to $1.6 \times 10^{-5}$ [m]. For Out-of-Distribution (OOD) trajectories, we increase the high pressure to a higher range: $10^{10}$ - $8 \times 10^{10}$ [Pa].

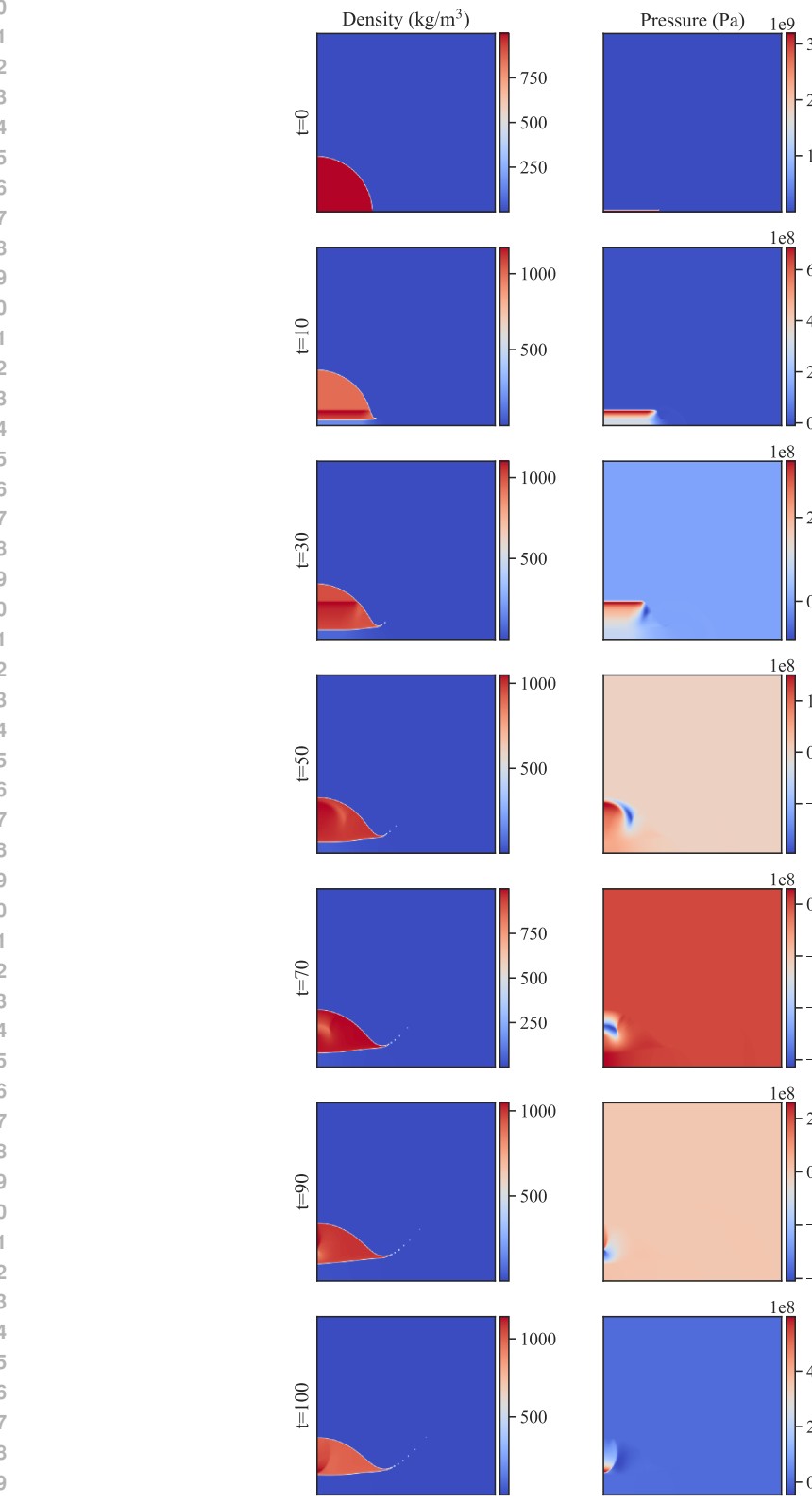

Figure 5: Visualization of droplet's motion and deformation in the LIDE dataset.

## A.2 THE SIDA DATASET

To get a better understanding of this problem, both wave dynamics and droplet breakup modes are studied extensively in the literature (Sharma et al., 2021), (Theofanous & Li, 2008). Breakup modes are characterized by the Weber number (Hinze, 1955), which is defined as follows:

$$We = \frac{\rho_2 u_2^2 d}{\sigma} \tag{8}$$

In this definition, $\rho_2$ and $u_2$ refer to post-shock density and velocity of the external flow, respectively. Additionally, $d$ is the droplet diameter and $\sigma$ denotes the surface tension coefficient. For droplet aero-breakup, two major breakup modes are introduced: Rayleigh-Taylor Piercing (RTP) and shear-induced entrainment (SIE). RTP is the main instability mode for small Weber numbers (starting at $We \approx 28$), and SIE is the terminal instability mode for increasing Weber numbers ($We > 10^3$) (Theofanous & Li, 2008). For this study, we cover the Weber number in the range [530, 40000], which corresponds to the transition regions from RTP to SIE and also the SIE region itself.

After the shock impact, the post-shock flow plays a significant role in droplet deformation and breakup. The post-shock flow regime is identified by the Mach number, which is a non-dimensional parameter that relates flow velocity to the speed of sound. We compute the post-shock flow properties using the normal shock relation. These relations are given by (Anderson, 1990):

$$u_s = M_s \cdot c_1 \tag{9}$$

$$u_{1,\text{rel}} = -u_s \tag{10}$$

$$u_1 = u_{1,\text{rel}} + u_s \tag{11}$$

$$T_2 = T_1 \left(1 + \frac{2\gamma \left(M_s^2 - 1\right)}{\gamma + 1}\right) \left(\frac{2 + (\gamma - 1)M_s^2}{(\gamma + 1)M_s^2}\right) \tag{12}$$

$$c_2 = \sqrt{\gamma \cdot R_1 \cdot T_2} \tag{13}$$

$$M_{f2,\text{rel}} = \sqrt{\frac{1 + \frac{\gamma - 1}{2}M_s^2}{\gamma M_s^2 - \frac{\gamma - 1}{2}}} \tag{14}$$

$$u_{2,\text{rel}} = M_{f2,\text{rel}} \cdot c_2 \tag{15}$$

$$u_2 = u_s - u_{2,\text{rel}} \tag{16}$$

$$\rho_2 = \rho_1 \cdot \frac{(\gamma + 1)M_s^2}{2 + (\gamma - 1)M_s^2} \tag{17}$$

$$p_2 = p_1 \left(1 + \frac{2\gamma \left(M_s^2 - 1\right)}{\gamma + 1}\right) \tag{18}$$

We use $M_s$ for the shock and $M_f$ for the post-shock flow Mach number. The flow states before and after the shock wave are referred to with subscripts 1 and 2, respectively. Furthermore, $T$ is the temperature, $c$ is the speed of sound, $\gamma = \frac{c_p}{c_v}$ is the ratio of specific heat, and $R$ is the specific gas constant. We consider shock Mach numbers spanning from 1.2 to 3.5. Then, based on the selected shock Mach number, we calculate $\rho_2$, $u_2$, and $p_2$ for the west Dirichlet boundary condition. Next, the surface tension coefficient is computed from the Weber number. A summary of initial condition values is presented in Table 8. It should be noted that the value $16.5R_0$ in the table, shows the location of the shock wave in the initial setup (refer to Figure 2).

Table 8: Initial conditions for the SIDA dataset

| $l$ | $\rho_l$ [kg m$^{-3}$] | $\{u_{r,l}, u_{z,l}\}$ [m s$^{-1}$] | $p_l$ [Pa] | $\mathbf{p_l}$ [Pa] |
|---|---|---|---|---|
| 1 (Ambient air) | $\rho_2$ | 0.0, $u_2$ | $p_2$ | $z < 16.5R_0$ |
| | 1.2 | 0.0, 0.0 | 101325.0 | $z > 16.5R_0$ |
| 2 (Liquid droplet) | 998.2 | 0.0, 0.0 | 101325.0 | |

**Parameter Range**. As mentioned above, we adopt shock Mach numbers spanning from 1.2 to 3.5, and Weber numbers between 530 and 40000. For the Out-of-Distribution (OOD) dataset, we extend the shock Mach number to range between 3.5 to 5.

**Validation.** We compare the SIDA dataset against numerical studies. Since we employ an axisymmetric setup in our simulation, a full three-dimensional study is referenced for validation (Winter et al., 2019), (Meng & Colonius, 2018). For this purpose, the non-dimensional time ($t^*$) and displacement of the center of mass (COM) in the droplet ($\Delta z^*$) are defined as

$$t^* = t \frac{u_2}{d} \sqrt{\frac{\rho_2}{\rho_{drop}}}, \tag{19}$$

and

$$\Delta z^* = \frac{z}{d}, \tag{20}$$

where $t$ is the saved timestep, and $\rho_{drop}$ is density of the liquid droplet. Upon shock and post-shock flow impact, the droplet COM accelerates. This trend is clearly observable in our dataset, which aligns with results from the literature. In Figure 7, the flattening of the droplet surface and the hat-shaped deformation are shown in order from top to bottom. Noteworthy, the perturbations on the surface of the droplet are related to shear-induced instabilities (Sharma et al., 2021).

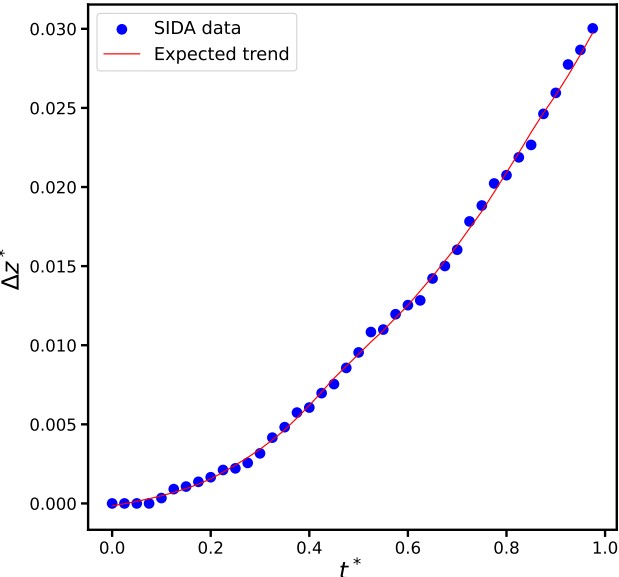

Figure 6: Validation of SIDA data against numerical studies (Winter et al., 2019),(Meng & Colonius, 2018).

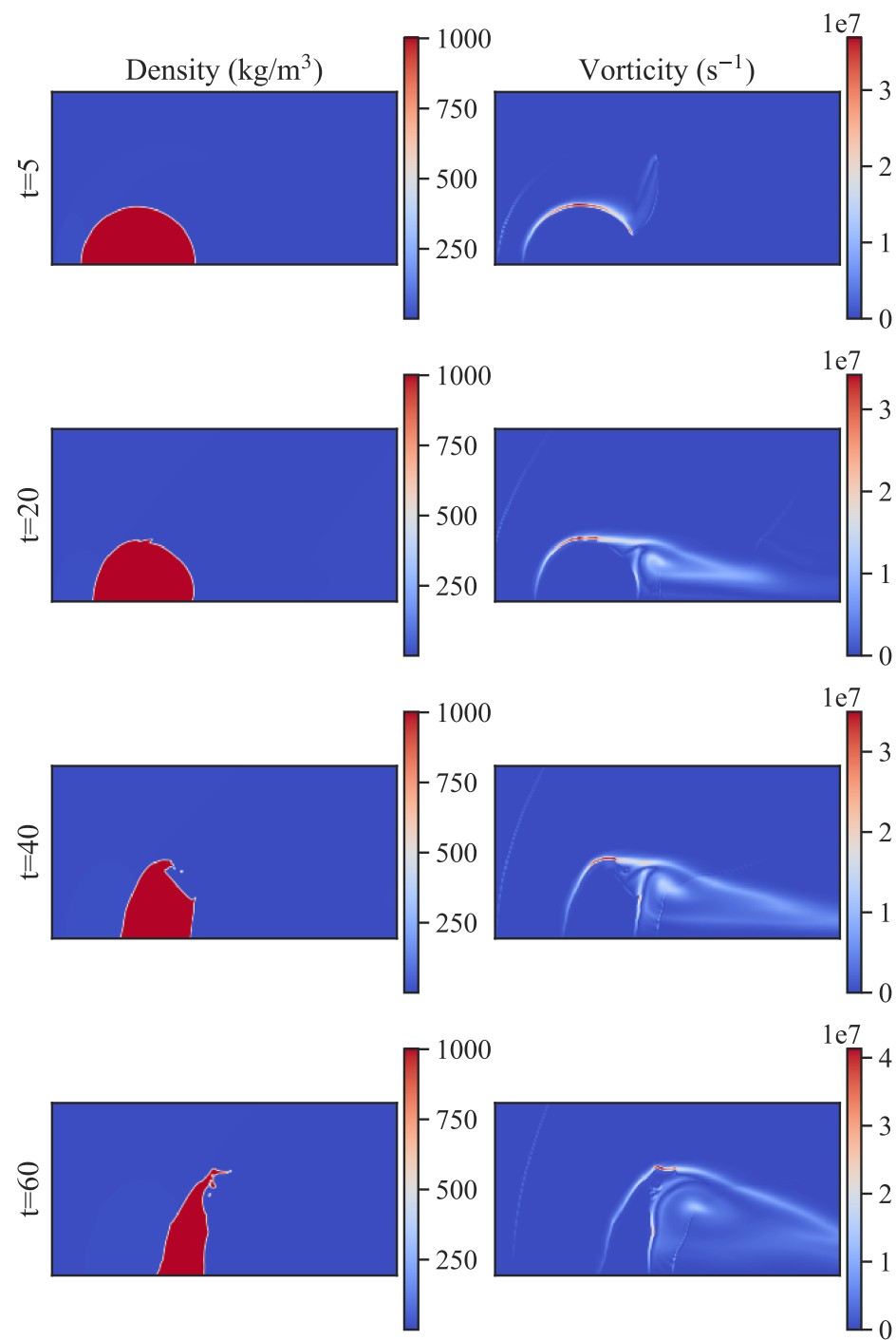

Figure 7: Visualization of droplet's motion and deformation in the SIDA dataset.

# B EXPERIMENT DETAILS

## B.1 DESIGN OF EXPERIMENTS

The complete set of experiments for the SIDA and the LIDE datasets is shown in Tables 9 and 10, respectively. We experiment with a variety of input, conditioning, and output channels, along with the combinations of sequence info and conditioning parameters.

Table 9: SIDA (PDUVVoS) experiments with Tag identifiers.

| # Expt | Tag | Input Channels | Output Channels | Cond[4] | Seq Info [5] |
|---|---|---|---|---|---|
| 1 | PDUV_F_(1,1) | Pressure, Density X-velocity, Y-Velocity | Pressure, Density X-Velocity, Y-Velocity | F | 1, 1, 5 |
| 2 | PDUV_F_(3,1) | Pressure, Density X-velocity, Y-velocity | Pressure, Density Velocity_x, Velocity_y | F | 3, 1, 5 |
| 3 | PDUV_T_(3,1) | Pressure, Density X-velocity, Y-velocity | Pressure, Density X-velocity, Y-velocity | T | 3, 1, 5 |
| 4 | PDUV[VoS]_F_(1,1) | Pressure, Density X-velocity, Y-velocity [Vorticity, Schlieren] | Pressure, Density X-velocity, Y-velocity | F | 1, 1, 5 |
| 5 | PDUV[VoS]_F_(3,1) | Pressure, Density X-velocity, Y-velocity [Vorticity, Schlieren] | Pressure, Density X-velocity, Y-velocity | F | 3, 1, 5 |
| 6 | PDUV[VoS]_T_(3,1) | Pressure, Density X-velocity, Y-velocity [Vorticity, Schlieren] | Pressure, Density X-velocity, Y-velocity | T | 3, 1, 5 |
| 7 | PDUV_F_(3,2) | Pressure, Density X-velocity, Y-velocity | Pressure, Density X-velocity, Y-velocity | F | 3, 2, 5 |
| 8 | PDUV[VoS]_F_(3,2) | Pressure, Density X-velocity, Y-velocity [Vorticity, Schlieren] | Pressure, Density X-velocity, Y-velocity | F | 3, 2, 5 |

---

[4]refers to the boolean flag indicating whether conditioning parameters are injected into the normalization layer.

[5]refers to the sequence information: [number of historic inputs, number of bundled predictions, stride between timesteps].

Table 10: LIDE (PDUVES) experiments with Tag identifiers.

| # Expt | Tag | Input Channels | Output Channels | Cond | Seq Info |
|---|---|---|---|---|---|
| 1 | P_F_(1,1) | Pressure | Pressure | F | 1, 1, 10 |
| 2 | P_F_(3,1) | Pressure | Pressure | F | 3, 1, 10 |
| 3 | P_T_(1,1) | Pressure | Pressure | T | 1, 1, 10 |
| 4 | P_T_(3,1) | Pressure | Pressure | T | 3, 1, 10 |
| 5 | PDUV_F_(1,1) | Pressure, Density X-velocity, Y-velocity | Pressure, Density X-velocity, Y-velocity | F | 1, 1, 10 |
| 6 | PDUV_F_(3,1) | Pressure, Density X-velocity, Y-velocity | Pressure, Density X-velocity, Y-velocity | F | 3, 1, 10 |
| 7 | P[ES]_F_(1,1) | Pressure, [Energy, Schlieren] | Pressure | F | 1, 1, 10 |
| 8 | P[ES]_F_(3,1) | Pressure [Energy, Schlieren] | Pressure | F | 3, 1, 10 |
| 9 | PDUV[ES]_F_(1,1) | Pressure, Density X-velocity, Y-velocity [Energy, Schlieren] | Pressure, Density X-velocity, Y-velocity | F | 1, 1, 10 |
| 10 | PDUV[ES]_F_(3,1) | Pressure, Density X-velocity, Y-velocity [Energy, Schlieren] | Pressure, Density X-velocity, Y-velocity | F | 3, 1, 10 |
| 11 | P_F_(3,2) | Pressure | Pressure | F | 3, 2, 10 |
| 12 | PDUV_F_(3,2) | Pressure, Density X-velocity, Y-velocity | Pressure, Density X-velocity, Y-velocity | F | 3, 2, 10 |
| 13 | PDUV_T_(1,1) | Pressure, Density X-velocity, Y-velocity | Pressure, Density X-velocity, Y-velocity | T | 1, 1, 10 |
| 14 | PDUV_T_(3,1) | Pressure, Density X-velocity, Y-velocity | Pressure, Density X-velocity, Y-velocity | T | 3, 1, 10 |

## B.2 BASELINE MODEL DETAILS

In this section, we provide a brief overview of all the models used as baselines. In all the models described in this section, the LayerNorm (Ba et al., 2016) is used as the default choice of normalization layer, and the normalized grid X- and Y-coordinates are appended as additional channels with the input channels.

1. **UNet:** We implement the UNet variant as described in Gupta & Brandstetter (2022). UNets follow a structure that first performs spatial downsampling and then spatial upsampling, with each block composed of multiple convolutional layers. A distinctive feature of UNet is the inclusion of skip connections that link activations from the downsampling path to their corresponding upsampling layers. Table 11 shows the hyperparameters chosen for the two model parameter categories. The number of latent channels corresponds to the feature dimension produced after the first convolutional layer. Along the downsampling path, the base latent channel dimension is adjusted according to a channel multiplier list, with each element specifying the factor used to increase the number of channels at successive levels of the model.

Table 11: UNet hyperparameters.

| Hyperparameters | 1M | 50M |
|---|---|---|
| Latent channels | 28 | 48 |
| Channel Multiplier | [1,4] | [1,2,2,4] |
| Activation | GELU | GELU |

2. **Residual Network (ResNet):** The baseline ResNet is implemented as described in Gupta & Brandstetter (2022), where no up- or down-projection techniques have been used. The input channels are projected to the latent channels by a convolutional layer and subsequently passed through four ResNet blocks. Each block consists of two 3x3 convolutional layers, each followed by an activation function and a norm layer. The convolutional layers employ a stride and padding of 1, preserving the spatial resolution of the feature maps. The final output is then obtained by adding the original input to the convolutional output. Refer to Table 12 for the hyperparameters.

Table 12: ResNet hyperparameters.

| Hyperparameters | 1M |
|---|---|
| Latent channels | 112 |
| # residual blocks | [1, 1, 1, 1] |
| Activation | GELU |

3. **Fourier Neural Operator (FNO):** The FNO is designed to approximate mappings between function spaces by performing computations directly in the Fourier domain. Its architecture can be divided into three main components: a lifting network, a sequence of Fourier layers, and a decoder network. We adopt the implementation described in Contributors (2023) and use the hyperparameters as shown in Table 13 for our experiments.

   The lifting network first maps the input channels into a higher-dimensional latent space using pointwise convolutions. The dimension of this latent space is described by the latent channels. The core of the model is composed of Fourier layers that have spectral convolution with a point-wise linear convolution layer acting as a skip connection. The activation is applied to the summation of the spectral convolutions and this convolutional skip layer. In each spectral convolution, the input is transformed into the Fourier domain using Fast Fourier Transform (FFT), where a specified number of modes are retained and updated with

learned complex weights, and the result is projected back to the spatial domain through the decoder network.

Table 13: FNO hyperparameters.

| Hyperparameters | 1M | 50M |
|---|---|---|
| Latent channels | 16 | 32 |
| FNO Layers | 4 | 6 |
| Modes | 16 | 45 |
| Padding | 8 | 8 |
| Padding Type | constant | constant |
| Activation in Fourier Layers | GELU | GELU |
| Decoder layers | 2 | 2 |
| Decoder layers size | 128 | 256 |
| Decoder activation | SiLU | SiLU |

4. **Vision Transformer (ViT):** A modified ViT (Dosovitskiy et al., 2020) architecture was adopted. The implementation follows the general ViT paradigm, splitting the image into square patches of size 8, embedding and passing them through a transformer encoder, and reconstructing the spatial output from the resulting latent representations with the additional capability to handle non-square inputs. The ViT model consists of a patch-based embedding, an encoder, and a decoder. Passing the input through the embedding-encoder-decoder pipeline results in a reconstruction of the original input shape. The Embedding divides the image into non-overlapping patches, embeds them via a linear projection, and adds positional encodings. For each patch, this results in a sequence of token vectors, each with dimensions specified by the latent channels. The transformer encoder processes this sequence using standard Multi-Head Self-Attention (MHSA) and feedforward layers, with the hidden size denoted by the intermediate size variable. The number of the hidden layers determines the number of the encoder layers. The number of MHSA in each layer is specified by the number of attention heads. This attention stage allows global spatial interactions across the patch grid, enabling the model to learn long-range dependencies. Table 14 shows the hyperparameters for the two learnable parameter categories.

Table 14: ViT hyperparameters.

| Hyperparameters | 1M | 50M |
|---|---|---|
| Latent channels | 128 | 504 |
| Patch size | 8 | 8 |
| # hidden layers | 2 | 12 |
| # attention heads | 4 | 14 |
| intermediate size | 512 | 1024 |
| Activation | GELU | GELU |

5. **Scalable Operator Transformer (ScOT):** The ScOT model is based on the Poseidon framework (Herde et al., 2024). At its core, ScOT adopts a hierarchical transformer architecture inspired by vision transformers with a window-based approach. The input is partitioned into a uniform grid of non-overlapping patches. We implement an additional capability to process non-square inputs. Each patch undergoes an averaging operation using a shared spatial weight matrix, followed by a linear projection into a latent embedding space, whose size is described by the latent channels. This procedure produces a piecewise-constant latent function representation over the domain, which serves as the input to the

transformer backbone. The motivation for this patch-based embedding is to reduce the computational complexity associated with global attention while preserving essential local information about the input field.

Once embedded, the representation is processed through a series of hierarchical SwinV2 Transformer blocks (Liu et al., 2021), arranged in multiple stages that progressively down-sample and subsequently upsample the latent feature maps, forming a UNet-like architecture. The number of blocks per stage is defined by the variable 'depths' in Table 15. Each stage applies windowed MHSA, where attention computations are restricted to local windows rather than the entire spatial domain, significantly reducing the quadratic cost of global attention. The number of parallel MHSA per stage is determined by the number of attention heads. To ensure information exchange across windows and avoid locality bias, the attention windows are shifted between consecutive layers, enabling effective global context modeling over multiple layers.

The hierarchical design incorporates patch merging operations during the encoder phase to reduce spatial resolution and increase the feature dimension, thereby allowing deeper layers to capture global structures. Conversely, the decoder phase employs patch expansion to restore resolution, and skip connections in the form of ConvNext blocks (Liu et al., 2022), bridging the corresponding encoder and decoder stages. The number of blocks per stage in the ConvNext blocks is specified by the hyperparameter 'skip-connections'.

Table 15: ScOT hyperparameters.

| Hyperparameters | 1M | 50M |
|---|---|---|
| Latent channels | 27 | 150 |
| Patch size | 4 | 4 |
| Depths | [3,3,3] | [4,4,4] |
| # attention heads | [3,6,12] | [6,12,24] |
| Skip connections | [2,2,0] | [3,3,0] |
| Window size | 16 | 16 |
| MLP ratio | 2.0 | 4.0 |
| Activation | GELU | GELU |

6. **Convolutional Neural Operator (CNO):** A CNO, similar to a UNet, processes an input function by first feeding it into a series of encoder layers, which progressively reduce the spatial resolution while increasing the number of channels. The representation is then passed through decoder layers, which perform the opposite operation: they restore spatial resolution while compressing the channel dimension. In parallel, the encoder and decoder stages, operating at the same spatial scale or spectral band, are linked through ResNet-style skip connections.

   The key idea lies in the model's upsampling and downsampling strategy, where a low-pass filter is applied to prevent the introduction of new high-frequency noise during the sampling process. This design complies with the Shannon sampling theorem, allowing the discrete data points to remain consistent with their corresponding continuous functions. Refer to Table 16 for the hyperparameters.

7. **Transolver** It is a transformer-based model designed to learn how physical fields evolve according to PDEs. Instead of treating the input like an image, it views the spatial grid as a set of points and learns how information should flow between them using attention. The model embeds the input fields, applies transformer layers that capture both local interactions and long-range dependencies, and then reconstructs the output field. Its design includes spatially meaningful positional encodings and mechanisms to compare points across a reference grid, enabling it to generalize across various resolutions and geometries. Refer to Table 17 for the hyperparameters.

Table 16: CNO hyperparameters.

| Hyperparameters | 1M | 50M |
| --- | --- | --- |
| Latent channels | 32 | 64 |
| Depth | 4 | 5 |
| # blocks (except at neck) | 2 | 6 |
| # blocks (at neck) | 2 | 7 |
| Channel multiplier | 16 | 32 |
| Activation | custom LeakyReLU | custom LeakyReLU |

Table 17: Transolver hyperparameters.

| Hyperparameters | 1M |
| --- | --- |
| Latent channels | 112 |
| # layers | 4 |
| # attention heads | 8 |
| Dropout | 0.0 |
| # slices | 32 |
| reference grid resolution | 12 |
| MLP ratio | 1.0 |
| Activation | GELU |

## B.3 CONDITIONING

In this section, we describe the formulation of the strategy used to integrate conditioning parameters into the model (Herde et al., 2024). For an input $x \in \mathbb{R}^d$, and $k$ being the conditioning parameters, the conditional layer norm formulation is given by Equation 21. Figure 8 illustrates this injection of conditioning parameters into the layer norm. Here $\gamma$ and $\beta$ are simple linear layers and $\hat{\gamma}$ and $\hat{\beta}$ are learnable affine transform parameters of the layer norm, respectively.

$$
\hat{\mathbf{x}} = \gamma(k) \odot \mathbf{x} + \beta(k)
$$

$$
LayerNorm(\mathbf{x}) = \hat{\gamma} \odot \frac{\hat{\mathbf{x}} - \mu(\hat{\mathbf{x}})}{\sqrt{\sigma^2(\hat{\mathbf{x}}) + \epsilon}} + \hat{\beta},
$$

$$
\mu(\hat{\mathbf{x}}) = \frac{1}{d} \sum_{j=1}^{d} \hat{x}_j, \quad \sigma^2(\hat{\mathbf{x}}) = \frac{1}{d} \sum_{j=1}^{d} \left( \hat{x}_j - \mu(\hat{\mathbf{x}}) \right)^2.
$$

(21)

## B.4 TRAINING HYPERPARAMETERS

Table 18 shows the training hyperparameters that are common for all the models. Each model has its own specific hyperparameters, which are described in Appendix B.2. All models were trained on NVIDIA RTX A6000 48GB GPU with bf16 mixed-precision, except for the FNO and CNO, which were trained on fp32.

## B.5 ERROR METRICS

To compute the Kinetic Energy (KE) of the domain comprising N cells for each rollout step, we employ Equation 22.

Figure 8: Conditional LayerNorm.

Table 18: Training hyperparameters.

| Hyperparameter | Value |
|---|---|
| Number of Epochs | 128 |
| Batch Size | 32 |
| Optimizer | AdamW |
| Weight Decay | 0.000001 |
| Learning Rate(LR) | 0.00005 |
| LR Scheduler | Cosine |
| Warmup Ratio | 0.0 |
| Mix-precision | bf16 (except FNO: fp32) |

$$\text{KE} = \Sigma_{i=0}^{N} \frac{1}{2} \rho_i (u_{r\,i}^{\,2} + u_{z\,i}^{\,2}) \Delta V_i \tag{22}$$

Similarly, for Vorticity Production (VP) for N cells, we use Equation 23.

$$\text{VP} = \Sigma_{i=0}^{N} \left( \frac{\partial u_{z\,i}}{\partial r} - \frac{\partial u_{r\,i}}{\partial z} \right)^2 \tag{23}$$

Here, $u_r$ and $u_z$ are the velocity components in the $r$ and $z$ directions, respectively. We compute the square to avoid negative vorticity values.

## C  RESULTS

The complete set of tables, including the evaluation results on both LIDE and SIDA, is shown in this section. These tables cover three scenarios, which are already discussed in the main text, section 5. These scenarios are the effect of sequence information, conditioning parameters, and fields. Moreover, we introduced two error types, 1 and 2, in section 4.5. For each experiment, we compute error type 2 over **output channels, the evolution of Kinetic Energy (KE), and Vorticity Production (VP)**. In addition, for each dataset, we introduce a quantity of interest (QoI) and repeat the computation over this Dataset-specific QoI. These are **the change of droplet outer radius (OR) and displacement of droplet center of mass (COM)** for LIDE and SIDA, respectively.

### C.1  ERROR-TYPE 2 METRICS FOR THE LIDE DATASET

Table 19: Effect of sequence information for LIDE In-Distribution (ID) and Out-of-Distribution (OOD) datasets. Error-type 2 over output channels from section 4.5 is presented. MODELS: UNet, FNO, ViT, ScOT, CNO

| MODEL | TAG | 1M | | 50M | |
|---|---|---|---|---|---|
| | | ID | OOD | ID | OOD |
| UNet | P_F_(1,1) | 9.1478±8.0895 | 13.0943±8.7709 | 5.3090±6.7340 | 11.9953±7.5931 |
| | P_F_(3,1) | 3.4922±3.9388 | 9.1548±6.0536 | 1.9950±2.6557 | 7.8092±7.1191 |
| | P_F_(3,2) | 3.9751±3.7768 | 9.0973±5.7687 | 2.1201±2.8103 | 7.2636±5.6294 |
| | P_T_(1,1) | 5.3515±4.9418 | 17.8080±15.0191 | 2.3514±2.6213 | 9.0930±6.1429 |
| | P_T_(3,1) | 4.3477±3.9575 | 12.7650±9.2722 | 1.6726±2.1158 | 7.5000±5.2114 |
| | PDUV_F_(1,1) | 2.9972±2.4756 | 9.0796±5.9230 | 2.1210±1.8598 | 8.0828±5.3407 |
| | PDUV_F_(3,1) | 1.5341±1.2526 | 6.0177±5.0874 | 1.0030±0.8511 | 5.0866±4.2680 |
| | PDUV_F_(3,2) | 1.8878±1.4960 | 5.1885±3.9663 | 1.0518±0.8712 | 5.4376±4.7212 |
| | P[ES]_F_(1,1) | 23.0535±18.8769 | 16.0039±11.8845 | 29.6090±24.6381 | 36.0516±30.8501 |
| | P[ES]_F_(3,1) | 13.0605±13.3488 | 20.2864±17.7559 | 11.1466±10.3795 | 19.2790±16.1179 |
| | PDUV[ES]_F_(1,1) | 7.5569±6.2939 | 10.5505±6.1862 | 9.9430±8.6244 | 11.4681±7.2812 |
| | PDUV[ES]_F_(3,1) | 5.1957±4.6966 | 9.3330±7.0384 | 4.4018±3.7968 | 8.5123±6.5495 |
| FNO | P_F_(1,1) | 6.3022±5.1934 | 10.3828±7.0152 | 5.2016±5.4331 | 12.0931±7.6183 |
| | P_F_(3,1) | 4.6875±4.6579 | 9.5000±6.8683 | 3.6339±4.4195 | 8.7828±6.6895 |
| | P_F_(3,2) | 4.9082±4.6673 | 8.6019±5.6522 | 3.4477±4.2682 | 8.0590±5.7641 |
| | P_T_(1,1) | 6.0708±5.0622 | 9.8894±6.0049 | 4.3767±4.2430 | 18.1397±15.7189 |
| | P_T_(3,1) | 3.9302±3.4157 | 8.7390±5.2368 | 2.8506±3.0168 | 8.6539±5.2262 |
| | PDUV_F_(1,1) | 3.6399±3.1425 | 7.8868±5.3663 | 2.5836±2.2495 | 7.2695±5.3881 |
| | PDUV_F_(3,1) | 2.4656±2.4281 | 5.9441±4.8157 | 1.6334±1.4492 | 6.4107±5.4997 |
| | PDUV_F_(3,2) | 2.0686±1.6025 | 5.8610±4.4382 | 1.5864±1.2262 | 6.3866±5.0470 |
| | P[ES]_F_(1,1) | 27.6326±22.1648 | 26.2085±20.8494 | 32.7032±25.4675 | 27.6014±23.5059 |
| | P[ES]_F_(3,1) | 10.8859±8.1119 | 14.1592±10.4569 | 14.2710±10.8054 | 14.5738±10.2457 |
| | PDUV[ES]_F_(1,1) | 8.6627±6.7686 | 12.3647±8.0489 | 8.9522±7.0050 | 11.1104±6.8842 |
| | PDUV[ES]_F_(3,1) | 6.6485±5.3640 | 9.6179±6.0460 | 5.0849±4.2891 | 8.7880±5.6131 |
| ViT | P_F_(1,1) | 9.1603±5.5769 | 10.2389±5.8189 | 6.2536±7.1400 | 10.8063±7.6196 |
| | P_F_(3,1) | 8.6900±5.5057 | 10.2917±6.1372 | 4.1896±4.9856 | 7.6857±6.3458 |
| | P_F_(3,2) | 7.6380±5.2494 | 10.4696±6.9706 | 4.4838±5.4155 | 7.6581±5.9456 |
| | P_T_(1,1) | 7.4830±5.7665 | 11.5443±7.7521 | 1.7560±2.0376 | 8.9017±6.6578 |
| | P_T_(3,1) | 5.0959±3.8155 | 10.5010±7.0844 | 1.5175±1.4244 | 7.7419±5.9593 |
| | PDUV_F_(1,1) | 6.7641±4.8388 | 8.5556±5.3823 | 2.1804±1.7873 | 6.3454±4.3554 |
| | PDUV_F_(3,1) | 4.1118±3.0304 | 6.4825±5.1926 | 1.4109±1.1031 | 3.9230±3.2763 |
| | PDUV_F_(3,2) | 3.3079±2.3757 | 6.0555±5.4917 | 1.3113±0.9807 | 4.1262±3.6042 |
| | P[ES]_F_(1,1) | 19.4751±19.8881 | 17.3331±13.6654 | 12.7624±10.7407 | 15.7572±12.5764 |
| | P[ES]_F_(3,1) | 9.2608±7.0138 | 11.8868±9.0272 | 7.3133±5.8014 | 10.2300±8.2406 |
| | PDUV[ES]_F_(1,1) | 9.2205±8.8594 | 11.4403±9.2148 | 4.3167±3.4105 | 7.1662±4.9084 |
| | PDUV[ES]_F_(3,1) | 5.4353±4.1444 | 7.7203±6.2456 | 3.8999±3.2830 | 5.4419±4.3560 |
| ScOT | P_F_(1,1) | 8.2057±6.9389 | 11.6114±7.4053 | 5.1421±6.1064 | 11.3345±7.6428 |
| | P_F_(3,1) | 4.8706±4.7652 | 7.0995±5.8052 | 3.1855±4.2143 | 6.1995±5.6304 |
| | P_F_(3,2) | 4.4133±4.2731 | 7.0349±6.0759 | 3.4873±4.3623 | 6.4399±5.7289 |
| | P_T_(1,1) | 7.8716±19.5111 | 77.7586±85.1101 | 2.2581±2.5968 | 11.4039±8.8884 |
| | P_T_(3,1) | 4.8011±4.6080 | 41.1675±43.1742 | 1.8432±2.0658 | 23.1959±25.7497 |
| | PDUV_F_(1,1) | 2.8022±2.2122 | 7.4147±4.9714 | 2.2992±1.8961 | 7.0310±4.6946 |
| | PDUV_F_(3,1) | 1.7180±1.3379 | 6.4316±7.3418 | 1.1854±0.9577 | 5.1242±5.4815 |
| | PDUV_F_(3,2) | 1.6306±1.1924 | 4.5044±4.0265 | 1.2412±0.9675 | 4.3705±3.9198 |
| | P[ES]_F_(1,1) | 21.4050±22.1739 | 22.0405±17.6376 | 16.8751±13.1522 | 15.5203±9.7885 |
| | P[ES]_F_(3,1) | 6.2541±5.3469 | 7.7537±5.7890 | 6.0439±4.9679 | 8.0174±5.8438 |
| | PDUV[ES]_F_(1,1) | 8.5861±7.6685 | 8.7523±6.9290 | 6.4329±6.2919 | 10.0934±7.6721 |
| | PDUV[ES]_F_(3,1) | 4.2951±4.1789 | 9.7228±10.4728 | 4.2373±4.2770 | 8.4763±8.7977 |
| CNO | P_F_(1,1) | 5.8675±6.4694 | 28.5730±20.1015 | 5.5428±6.2562 | 35.9518±35.2050 |
| | P_F_(3,1) | 2.1754±2.6216 | 10.0666±8.6685 | 2.3462±3.2362 | 7.4284±6.6611 |
| | P_F_(3,2) | 2.6451±2.9804 | 7.2544±6.5288 | 2.2003±2.9196 | 10.2079±9.7935 |
| | P_T_(1,1) | 5.7219±5.7462 | 16.1748±13.3171 | 3.7626±4.6015 | 11.5224±7.8040 |
| | P_T_(3,1) | 4.2187±4.2207 | 10.7036±7.3115 | 3.1813±3.2968 | 8.9759±5.4488 |
| | PDUV_F_(1,1) | 2.1786±1.7884 | 7.7764±6.8129 | 2.1271±1.7921 | 6.7606±4.9076 |
| | PDUV_F_(3,1) | 1.2258±0.9966 | 6.4241±6.3085 | 1.1850±0.9985 | 4.7852±3.9415 |
| | PDUV_F_(3,2) | 1.3169±1.0001 | 4.1531±3.4367 | 1.3296±1.0634 | 4.6685±3.9123 |
| | P[ES]_F_(1,1) | 26.0960±22.0672 | 39.5163±34.1637 | 27.8891±22.8285 | 33.5685±28.1269 |
| | P[ES]_F_(3,1) | 8.6612±7.3315 | 10.0280±7.1972 | 8.4329±7.1614 | 14.1459±12.8208 |
| | PDUV[ES]_F_(1,1) | 8.7094±7.9076 | 13.4745±9.9670 | 7.5888±6.7066 | 11.8868±9.0768 |
| | PDUV[ES]_F_(3,1) | 4.9778±4.7576 | 7.3581±6.0097 | 4.7545±4.2252 | 7.9192±6.5797 |

Table 20: Effect of sequence information for LIDE In-Distribution (ID) and Out-of-Distribution (OOD) datasets. Error-type 2 over output channels from section 4.5 is presented. MODELS: ResNet and Transolver

| MODEL | TAG | 1M | |
|---|---|---|---|
| | | ID | OOD |
| ResNet | P_F_(1,1) | 15.8603±14.2650 | 14.5966±10.5899 |
| | P_F_(3,1) | 9.7710±7.7492 | 10.9950±7.2943 |
| | P_F_(3,2) | 6.3019±5.0498 | 9.5956±5.5532 |
| | P_T_(1,1) | 8.4347±6.6584 | 18.5181±14.7422 |
| | P_T_(3,1) | 6.2434±5.2237 | 20.0036±15.6632 |
| | PDUV_F_(1,1) | 7.1277±5.7086 | 9.8022±6.0704 |
| | PDUV_F_(3,1) | 5.0517±4.3040 | 6.4456±4.5672 |
| | PDUV_F_(3,2) | 2.6025±1.9887 | 5.3443±3.8090 |
| | P[ES]_F_(1,1) | 22.7663±16.9184 | 30.5975±25.0920 |
| | P[ES]_F_(3,1) | 10.6258±7.4852 | 14.4755±10.1972 |
| | PDUV[ES]_F_(1,1) | 6.5783±4.6303 | 10.2784±6.3609 |
| | PDUV[ES]_F_(3,1) | 6.3316±4.7105 | 8.5529±5.2428 |
| Transolver | P_F_(1,1) | 7.7697±6.8997 | 11.3624±7.4919 |
| | P_F_(3,1) | 3.7175±3.7892 | 6.7653±5.1699 |
| | P_F_(3,2) | 4.6809±4.1168 | 8.7001±6.3872 |
| | P_T_(1,1) | 3.8780±3.8781 | 9.3457±6.6854 |
| | P_T_(3,1) | 3.0103±2.9045 | 7.5745±5.2939 |
| | PDUV_F_(1,1) | 4.9459±3.6294 | 8.3506±5.2692 |
| | PDUV_F_(3,1) | 3.4782±2.3452 | 5.5601±4.1509 |
| | PDUV_F_(3,2) | 5.8298±3.8836 | 6.7510±4.4488 |
| | P[ES]_F_(1,1) | 16.7733±12.4990 | 14.5373±9.9606 |
| | P[ES]_F_(3,1) | 9.6835±7.1451 | 10.8923±8.3330 |
| | PDUV[ES]_F_(1,1) | 9.1508±7.5811 | 9.7693±7.6113 |
| | PDUV[ES]_F_(3,1) | 6.0474±4.3954 | 7.4736±5.4511 |

Table 21: Effect of sequence information for LIDE In-Distribution (ID) and Out-of-Distribution (OOD) datasets. Error-type 2 over KE from section 4.5 is presented.

| MODEL | TAG | 1M | | 50M | |
|---|---|---|---|---|---|
| | | ID | OOD | ID | OOD |
| UNet | PDUV_F_(1,1) | 2.9558±3.4607 | 7.4869±9.0017 | 0.7912±0.9653 | 8.0747±7.5999 |
| | PDUV_F_(3,1) | 1.5158±1.7694 | 5.8522±5.8268 | 0.6309±0.9576 | 4.6977±3.9619 |
| | PDUV_F_(3,2) | 2.0264±2.2633 | 7.9273±6.9269 | 1.1124±1.2723 | 5.7533±5.6323 |
| | PDUV[ES]_F_(1,1) | 20.3706±34.1137 | 10.1271±15.9696 | 85.3670±95.5683 | 153.1264±176.1153 |
| | PDUV[ES]_F_(3,1) | 16.7911±23.1446 | 106.7279±127.6807 | 7.1791±9.1559 | 38.0896±41.2699 |
| FNO | PDUV_F_(1,1) | 17.3105±56.5599 | 10.9310±16.6896 | 7.7942±39.8016 | 20.4284±40.8841 |
| | PDUV_F_(3,1) | 9.4252±25.9660 | 13.1689±17.0462 | 5.0986±27.2035 | 5.3432±9.4238 |
| | PDUV_F_(3,2) | 3.5499±5.2362 | 7.0161±4.5337 | 1.4311±1.4455 | 5.6575±4.4884 |
| | PDUV[ES]_F_(1,1) | 25.1967±25.9655 | 28.1146±31.9155 | 20.8824±31.6092 | 9.1176±11.8356 |
| | PDUV[ES]_F_(3,1) | 11.2583±9.6838 | 4.7826±3.8797 | 6.1821±7.2471 | 21.7777±30.0173 |
| ViT | PDUV_F_(1,1) | 30.7485±75.0957 | 13.4051±28.2157 | 1.9576±1.9775 | 2.7012±2.0355 |
| | PDUV_F_(3,1) | 8.4927±9.5889 | 21.5649±28.8912 | 2.0578±1.8248 | 11.1058±6.4700 |
| | PDUV_F_(3,2) | 5.8635±4.6720 | 42.0109±101.9154 | 3.1128±2.6023 | 14.1639±8.6045 |
| | PDUV[ES]_F_(1,1) | 241.4799±870.9689 | 145.0038±397.7887 | 16.0164±19.8772 | 6.9168±8.2187 |
| | PDUV[ES]_F_(3,1) | 27.0586±39.7306 | 25.4535±23.4193 | 9.8028±13.6173 | 14.4241±9.5125 |
| ScOT | PDUV_F_(1,1) | 3.3588±3.5977 | 3.6852±3.2128 | 1.5700±1.4611 | 3.0450±2.3063 |
| | PDUV_F_(3,1) | 3.4918±4.2602 | 22.0304±39.1454 | 1.8417±1.7165 | 7.0220±18.5086 |
| | PDUV_F_(3,2) | 4.6926±4.0309 | 14.9502±8.3347 | 2.1463±1.7221 | 5.9104±4.1972 |
| | PDUV[ES]_F_(1,1) | 62.5426±145.8651 | 25.8621±39.7799 | 43.8361±70.6359 | 27.6410±45.7599 |
| | PDUV[ES]_F_(3,1) | 10.6228±13.9594 | 46.5500±63.7638 | 15.5375±19.7119 | 40.3355±42.3768 |
| CNO | PDUV_F_(1,1) | 2.4282±2.7867 | 14.8098±19.7260 | 1.6192±1.9654 | 7.1715±7.9620 |
| | PDUV_F_(3,1) | 1.6514±1.7726 | 31.4155±37.0237 | 1.2797±1.4745 | 4.4545±4.1664 |
| | PDUV_F_(3,2) | 2.4189±2.1014 | 6.7205±5.8820 | 1.8983±1.6440 | 4.5032±4.4261 |
| | PDUV[ES]_F_(1,1) | 42.3454±45.6934 | 147.1579±157.3773 | 21.5265±26.4024 | 99.8242±121.7740 |
| | PDUV[ES]_F_(3,1) | 14.6887±19.6394 | 41.0990±52.3916 | 2.7269±3.3062 | 6.9642±9.2008 |
| ResNet | PDUV_F_(1,1) | 72.0392±186.8011 | 21.9601±35.3048 | | |
| | PDUV_F_(3,1) | 5.2020±11.0591 | 5.9362±3.9994 | | |
| | PDUV_F_(3,2) | 3.2099±4.3074 | 8.1963±5.5746 | | |
| | PDUV[ES]_F_(1,1) | 30.4190±41.2347 | 64.9705±64.7465 | | |
| | PDUV[ES]_F_(3,1) | 26.2392±33.9826 | 25.1360±32.8936 | | |
| Transolver | PDUV_F_(1,1) | 5.6585±6.2504 | 5.4874±3.5771 | | |
| | PDUV_F_(3,1) | 8.2177±8.3123 | 31.5978±26.8641 | | |
| | PDUV_F_(3,2) | 19.5804±13.9786 | 37.3732±20.2799 | | |
| | PDUV[ES]_F_(1,1) | 72.7505±228.9720 | 27.5679±43.1728 | | |
| | PDUV[ES]_F_(3,1) | 38.8649±37.2626 | 72.0413±61.4863 | | |

Table 22: Effect of sequence information for LIDE In-Distribution (ID) and Out-of-Distribution (OOD) datasets. Error-type 2 over VP from section 4.5 is presented.

| MODEL | TAG | 1M | | 50M | |
|---|---|---|---|---|---|
| | | ID | OOD | ID | OOD |
| UNet | PDUV_F_(1,1) | 2.2364±1.8249 | 7.6716±4.9818 | 1.1246±1.3118 | 6.6250±4.5504 |
| | PDUV_F_(3,1) | 1.4657±1.3658 | 4.2129±2.3582 | 0.5786±0.6629 | 4.8546±3.6289 |
| | PDUV_F_(3,2) | 2.8379±1.8601 | 5.5938±3.3693 | 0.8377±0.7912 | 4.2676±2.6929 |
| | PDUV[ES]_F_(1,1) | 5.9239±6.3310 | 6.2247±4.3015 | 9.2649±8.3096 | 15.2664±17.0709 |
| | PDUV[ES]_F_(3,1) | 3.7189±4.2770 | 18.7705±29.1679 | 2.3040±2.8389 | 8.0838±8.7216 |
| FNO | PDUV_F_(1,1) | 15.4477±38.8379 | 10.0575±11.7609 | 8.1643±39.6190 | 10.5453±15.8533 |
| | PDUV_F_(3,1) | 8.7514±19.1512 | 10.0428±11.7263 | 4.8705±22.6135 | 5.1937±3.2168 |
| | PDUV_F_(3,2) | 4.7165±3.5565 | 6.9232±4.4139 | 2.6684±2.0407 | 5.2543±3.3209 |
| | PDUV[ES]_F_(1,1) | 7.6566±9.2270 | 6.3084±3.6235 | 6.2573±7.7260 | 6.8818±7.8352 |
| | PDUV[ES]_F_(3,1) | 4.1110±4.3578 | 4.6076±3.7051 | 2.6139±2.7369 | 20.8548±24.9235 |
| ViT | PDUV_F_(1,1) | 35.9765±50.1389 | 10.7088±14.2735 | 2.1195±1.7062 | 6.0996±3.5933 |
| | PDUV_F_(3,1) | 13.7822±15.4450 | 11.8592±20.5723 | 1.7101±1.8497 | 3.5671±2.1539 |
| | PDUV_F_(3,2) | 3.7469±4.5967 | 5.0032±5.8750 | 2.0180±1.3555 | 4.2558±2.5245 |
| | PDUV[ES]_F_(1,1) | 116.2920±166.9602 | 51.8742±99.6273 | 5.3122±6.8443 | 5.0159±5.2584 |
| | PDUV[ES]_F_(3,1) | 31.3124±44.7175 | 21.5597±25.9508 | 5.0579±7.6161 | 5.0811±7.9904 |
| ScOT | PDUV_F_(1,1) | 3.2438±2.3717 | 7.2868±3.9963 | 1.4184±1.3680 | 5.8183±3.3720 |
| | PDUV_F_(3,1) | 2.7103±2.0030 | 6.1239±5.2132 | 0.8984±0.8154 | 4.7614±3.2915 |
| | PDUV_F_(3,2) | 3.3650±2.1891 | 5.9972±3.2918 | 1.3082±0.9690 | 4.8981±2.9342 |
| | PDUV[ES]_F_(1,1) | 6.2914±11.0882 | 6.8750±4.1989 | 8.1067±12.0643 | 7.6245±5.7289 |
| | PDUV[ES]_F_(3,1) | 2.9604±3.3728 | 6.7903±6.5895 | 2.5907±3.6683 | 4.8924±3.7627 |
| CNO | PDUV_F_(1,1) | 2.3980±1.9352 | 5.0360±2.8973 | 1.4503±1.5786 | 5.6487±3.7962 |
| | PDUV_F_(3,1) | 1.9368±1.4496 | 4.8935±2.9276 | 1.1844±1.1809 | 5.1373±3.3016 |
| | PDUV_F_(3,2) | 2.7687±1.7760 | 5.9649±3.6874 | 1.7443±1.3692 | 5.1603±3.1993 |
| | PDUV[ES]_F_(1,1) | 5.6469±6.5604 | 7.1679±7.5588 | 4.5431±5.7530 | 6.5345±7.5133 |
| | PDUV[ES]_F_(3,1) | 2.5447±2.8231 | 4.2803±2.7221 | 2.1570±2.6265 | 4.6820±3.2973 |
| ResNet | PDUV_F_(1,1) | 35.3231±75.9315 | 5.7896±4.4612 | | |
| | PDUV_F_(3,1) | 28.5191±65.6205 | 6.3865±7.7030 | | |
| | PDUV_F_(3,2) | 17.1056±37.6935 | 6.7059±5.9792 | | |
| | PDUV[ES]_F_(1,1) | 66.6398±120.4167 | 21.7128±30.8889 | | |
| | PDUV[ES]_F_(3,1) | 29.7433±46.1951 | 7.7186±9.6800 | | |
| Transolver | PDUV_F_(1,1) | 3.7311±4.9342 | 6.1950±3.9039 | | |
| | PDUV_F_(3,1) | 3.2161±4.3176 | 6.8555±5.0112 | | |
| | PDUV_F_(3,2) | 11.5979±11.1597 | 13.8273±10.7180 | | |
| | PDUV[ES]_F_(1,1) | 48.7057±82.3430 | 19.3961±24.8031 | | |
| | PDUV[ES]_F_(3,1) | 6.6554±8.8829 | 8.8493±14.1979 | | |

Table 23: Effect of sequence information for LIDE In-Distribution (ID) and Out-of-Distribution (OOD) datasets. Error-type 2 over OR from section 4.5 is presented.

| MODEL | TAG | 1M | | 50M | |
|---|---|---|---|---|---|
| | | ID | OOD | ID | OOD |
| UNet | PDUV_F_(1,1) | 27.6161±83.5283 | 288.1661±361.6815 | 25.9451±78.9227 | 336.0100±416.4188 |
| | PDUV_F_(3,1) | 22.5440±64.7437 | 360.2897±422.2532 | 21.6422±62.6203 | 325.0154±380.1980 |
| | PDUV_F_(3,2) | 25.2007±72.3719 | 304.1421±357.8917 | 11.5376±35.9400 | 246.6969±285.4076 |
| | PDUV[ES]_F_(1,1) | 10.1121±13.9343 | 12.7286±27.3411 | 19.4516±58.1035 | 27.6148±52.5871 |
| | PDUV[ES]_F_(3,1) | 21.5989±67.8911 | 68.7352±141.6686 | 18.9192±53.5520 | 74.0707±178.7413 |
| FNO | PDUV_F_(1,1) | 29.6074±93.7013 | 313.8395±437.8267 | 9.9817±46.2403 | 320.7908±453.1373 |
| | PDUV_F_(3,1) | 37.0998±106.2921 | 413.0769±490.3942 | 26.7387±80.1169 | 331.4658±421.8210 |
| | PDUV_F_(3,2) | 29.6000±82.8274 | 343.0459±387.5149 | 29.7442±83.5604 | 278.5166±342.8986 |
| | PDUV[ES]_F_(1,1) | 42.1162±111.6572 | 389.1060±525.8061 | 13.9013±41.0257 | 100.3914±190.0753 |
| | PDUV[ES]_F_(3,1) | 11.3997±36.5709 | 65.5979±245.8531 | 17.4764±49.3967 | 227.7179±364.7697 |
| ViT | PDUV_F_(1,1) | 28.3570±84.8627 | 316.4301±381.2929 | 7.3652±28.7282 | 154.0128±218.9231 |
| | PDUV_F_(3,1) | 26.6128±87.9115 | 106.3928±240.9438 | 16.3596±58.4039 | 108.5414±239.3181 |
| | PDUV_F_(3,2) | 31.4939±89.9586 | 403.3719±457.6138 | 39.1299±108.5654 | 395.2225±461.4427 |
| | PDUV[ES]_F_(1,1) | 24.9565±69.7981 | 240.2715±326.7355 | 19.9529±60.1380 | 212.9958±261.6513 |
| | PDUV[ES]_F_(3,1) | 11.2870±38.3220 | 171.8278±256.1028 | 6.3234±11.7187 | 105.0288±216.5524 |
| ScOT | PDUV_F_(1,1) | 19.0514±57.4875 | 213.0902±263.9498 | 29.8124±91.7401 | 360.4875±460.3578 |
| | PDUV_F_(3,1) | 25.3996±76.6864 | 361.9640±412.0892 | 8.1928±41.8663 | 176.1184±329.3044 |
| | PDUV_F_(3,2) | 25.3838±74.0512 | 370.2090±439.8857 | 16.7229±61.8938 | 238.5913±329.3854 |
| | PDUV[ES]_F_(1,1) | 28.4080±86.3271 | 297.9673±404.0781 | 19.1925±58.2088 | 216.5683±272.0256 |
| | PDUV[ES]_F_(3,1) | 33.4385±101.7622 | 333.1158±415.8457 | 21.1220±59.9334 | 271.1644±309.5756 |
| CNO | PDUV_F_(1,1) | 20.3428±62.2283 | 407.2689±527.2424 | 19.7226±60.0229 | 261.2177±332.9367 |
| | PDUV_F_(3,1) | 22.7767±66.8182 | 163.3376±283.7012 | 19.7672±71.7488 | 91.9920±200.5455 |
| | PDUV_F_(3,2) | 15.4841±49.3636 | 144.9798±221.2885 | 35.2070±104.4445 | 407.2021±461.4344 |
| | PDUV[ES]_F_(1,1) | 34.5318±101.6591 | 298.9503±438.7167 | 16.0693±48.4838 | 71.8442±107.9373 |
| | PDUV[ES]_F_(3,1) | 7.3676±11.9052 | 9.7813±5.3788 | 17.2166±48.6478 | 207.3200±232.2929 |
| ResNet | PDUV_F_(1,1) | 48.2119±127.4267 | 443.6026±559.3933 | | |
| | PDUV_F_(3,1) | 50.6830±131.6622 | 507.9845±594.5909 | | |
| | PDUV_F_(3,2) | 43.3286±119.8816 | 285.9452±379.6090 | | |
| | PDUV[ES]_F_(1,1) | 20.2926±66.2371 | 74.7534±168.2826 | | |
| | PDUV[ES]_F_(3,1) | 5.0466±4.8902 | 8.6086±7.1586 | | |
| Transolver | PDUV_F_(1,1) | 50.4440±122.6158 | 479.4698±587.8039 | | |
| | PDUV_F_(3,1) | 54.6837±128.0420 | 312.8717±446.2977 | | |
| | PDUV_F_(3,2) | 28.0606±64.1206 | 222.4606±249.9115 | | |
| | PDUV[ES]_F_(1,1) | 11.5269±6.4621 | 315.8390±533.3240 | | |
| | PDUV[ES]_F_(3,1) | 15.6974±29.7730 | 306.3072±530.8621 | | |

Table 24: Effect of conditioning parameters for LIDE In-Distribution (ID) and Out-of-Distribution (OOD) datasets. Error-type 2 over output channels from section 4.5 is presented.

| MODEL | TAG | 1M | | 50M | |
|---|---|---|---|---|---|
| | | ID | OOD | ID | OOD |
| UNet | P_F_(1,1) | 9.1478±8.0895 | 13.0942±8.7709 | 5.3089±6.7339 | 11.9953±7.5931 |
| | P_T_(1,1) | 5.3515±4.9417 | 17.8080±15.0190 | 2.3513±2.6213 | 9.0930±6.1429 |
| | P_F_(3,1) | 3.4922±3.9388 | 9.1547±6.0535 | 1.9950±2.6556 | 7.8092±7.1190 |
| | P_T_(3,1) | 4.3477±3.9574 | 12.7650±9.2722 | 1.6726±2.1157 | 7.4999±5.2113 |
| | PDUV_F_(1,1) | 2.9972±2.4755 | 9.0795±5.9229 | 2.1209±1.8598 | 8.0827±5.3407 |
| | PDUV_T_(1,1) | 2.9636±2.5183 | 10.1979±7.5259 | 1.2553±1.1437 | 8.1129±5.4866 |
| | PDUV_F_(3,1) | 1.5341±1.2526 | 6.0177±5.0873 | 1.0029±0.8511 | 5.0866±4.2680 |
| | PDUV_T_(3,1) | 2.6413±2.3683 | 9.6531±7.8398 | 1.0383±0.9354 | 6.6239±5.2895 |
| FNO | P_F_(1,1) | 6.3022±5.1934 | 10.3828±7.0152 | 5.2016±5.4331 | 12.0931±7.6183 |
| | P_T_(1,1) | 6.0708±5.0622 | 9.8894±6.0049 | 4.3767±4.2430 | 18.1397±15.7189 |
| | P_F_(3,1) | 4.6875±4.6579 | 9.5000±6.8683 | 3.6339±4.4195 | 8.7828±6.6895 |
| | P_T_(3,1) | 3.9302±3.4157 | 8.7390±5.2368 | 2.8506±3.0168 | 8.6539±5.2262 |
| | PDUV_F_(1,1) | 3.6399±3.1425 | 7.8868±5.3663 | 2.5836±2.2495 | 7.2695±5.3881 |
| | PDUV_T_(1,1) | 2.7562±2.4957 | 9.1299±6.1320 | 1.6632±1.4450 | 8.7610±6.3761 |
| | PDUV_F_(3,1) | 2.4656±2.4281 | 5.9441±4.8157 | 1.6334±1.4492 | 6.4107±5.4997 |
| | PDUV_T_(3,1) | 2.1101±2.2230 | 6.4283±5.1802 | 1.5349±1.2236 | 7.3136±5.1163 |
| ViT | P_F_(1,1) | 9.1603±5.5769 | 10.2389±5.8189 | 6.2536±7.1400 | 10.8063±7.6196 |
| | P_T_(1,1) | 7.4830±5.7665 | 11.5443±7.7521 | 1.7560±2.0376 | 8.9017±6.6578 |
| | P_F_(3,1) | 8.6900±5.5057 | 10.2917±6.1372 | 4.1896±4.9856 | 7.6857±6.3458 |
| | P_T_(3,1) | 5.0959±3.8155 | 10.5010±7.0844 | 1.5175±1.4244 | 7.7419±5.9593 |
| | PDUV_F_(1,1) | 6.7641±4.8388 | 8.5556±5.3823 | 2.1804±1.7873 | 6.3454±4.3554 |
| | PDUV_T_(1,1) | 4.6571±3.6689 | 11.1820±12.6204 | 1.0744±0.7655 | 6.0053±5.3625 |
| | PDUV_F_(3,1) | 4.1118±3.0304 | 6.4825±5.1926 | 1.4109±1.1031 | 3.9230±3.2763 |
| | PDUV_T_(3,1) | 3.4518±2.4023 | 10.4698±11.1001 | 1.0612±0.7537 | 4.4118±4.0008 |
| ScOT | P_F_(1,1) | 8.2057±6.9389 | 11.6114±7.4053 | 5.1421±6.1064 | 11.3345±7.6428 |
| | P_T_(1,1) | 7.8716±19.5111 | 77.7586±85.1101 | 2.2581±2.5968 | 11.4039±8.8884 |
| | P_F_(3,1) | 4.8706±4.7652 | 7.0995±5.8052 | 3.1855±4.2143 | 6.1995±5.6304 |
| | P_T_(3,1) | 4.8011±4.6080 | 41.1675±43.1742 | 1.8432±2.0658 | 23.1959±25.7497 |
| | PDUV_F_(1,1) | 2.8022±2.2122 | 7.4147±4.9714 | 2.2992±1.8961 | 7.0310±4.6946 |
| | PDUV_T_(1,1) | 2.1729±1.6478 | 15.8824±16.0901 | 1.0326±0.8473 | 10.2492±10.8710 |
| | PDUV_F_(3,1) | 1.7180±1.3379 | 6.4316±7.3418 | 1.1854±0.9577 | 5.1242±5.4815 |
| | PDUV_T_(3,1) | 1.7619±1.3709 | 8.0149±6.9873 | 0.8772±0.6840 | 7.8435±7.3701 |
| CNO | P_F_(1,1) | 5.8675±6.4694 | 28.5730±20.1015 | 5.5428±6.2562 | 35.9518±35.2050 |
| | P_T_(1,1) | 5.7219±5.7462 | 16.1748±13.3171 | 3.7626±4.6015 | 11.5224±7.8040 |
| | P_F_(3,1) | 2.1754±2.6216 | 10.0666±8.6685 | 2.3462±3.2362 | 7.4284±6.6611 |
| | P_T_(3,1) | 4.2187±4.2207 | 10.7036±7.3115 | 3.1813±3.2968 | 8.9760±5.4488 |
| | PDUV_F_(1,1) | 2.1786±1.7884 | 7.7764±6.8129 | 2.1271±1.7921 | 6.7606±4.9076 |
| | PDUV_T_(1,1) | 2.6620±2.2500 | 10.6073±7.5735 | 2.0716±1.8737 | 9.5014±6.0985 |
| | PDUV_F_(3,1) | 1.2258±0.9966 | 6.4241±6.3085 | 1.1850±0.9985 | 4.7852±3.9415 |
| | PDUV_T_(3,1) | 2.6061±2.4252 | 9.0774±6.4503 | 1.8087±1.6317 | 8.0127±5.4545 |
| ResNet | P_F_(1,1) | 15.8603±14.2650 | 14.5966±10.5899 | | |
| | P_T_(1,1) | 8.4347±6.6584 | 18.5181±14.7422 | | |
| | P_F_(3,1) | 9.7710±7.7492 | 10.9950±7.2943 | | |
| | P_T_(3,1) | 6.2434±5.2237 | 20.0036±15.6632 | | |
| | PDUV_F_(1,1) | 7.1277±5.7086 | 9.8022±6.0704 | | |
| | PDUV_T_(1,1) | 4.8602±3.8324 | 9.1651±5.5656 | | |
| | PDUV_F_(3,1) | 5.0517±4.3040 | 6.4456±4.5672 | | |
| | PDUV_T_(3,1) | 3.3461±2.5509 | 8.8525±6.2045 | | |
| Transolver | P_F_(1,1) | 7.7697±6.8997 | 11.3624±7.4919 | | |
| | P_T_(1,1) | 3.8780±3.8781 | 9.3457±6.6854 | | |
| | P_F_(3,1) | 3.7175±3.7892 | 6.7653±5.1699 | | |
| | P_T_(3,1) | 3.0103±2.9045 | 7.5745±5.2939 | | |
| | PDUV_F_(1,1) | 4.9459±3.6294 | 8.3506±5.2692 | | |
| | PDUV_T_(1,1) | 8.3954±5.2724 | 8.7039±5.2462 | | |
| | PDUV_F_(3,1) | 3.4782±2.3452 | 5.5601±4.1509 | | |
| | PDUV_T_(3,1) | 4.9456±3.2063 | 8.5088±5.8755 | | |

Table 25: Effect of conditioning parameters for LIDE In-Distribution (ID) and Out-of-Distribution (OOD) datasets. Error-type 2 over KE from section 4.5 is presented.

| MODEL | TAG | 1M | | 50M | |
|---|---|---|---|---|---|
| | | ID | OOD | ID | OOD |
| UNet | PDUV_F_(1,1) | 2.9558±3.4607 | 7.4869±9.0017 | 0.7912±0.9653 | 8.0747±7.5999 |
| | PDUV_T_(1,1) | 5.8103±14.5419 | 44.3624±67.1609 | 1.8841±2.2318 | 21.0380±39.8297 |
| | PDUV_F_(3,1) | 1.5158±1.7694 | 5.8522±5.8268 | 0.6309±0.9576 | 4.6977±3.9619 |
| | PDUV_T_(3,1) | 5.9552±13.5994 | 178.5762±265.3987 | 2.0062±6.6965 | 65.4653±124.1800 |
| FNO | PDUV_F_(1,1) | 17.3105±56.5599 | 10.9310±16.6896 | 7.7942±39.8016 | 20.4284±40.8841 |
| | PDUV_T_(1,1) | 3.5652±5.4341 | 9.1283±13.5591 | 2.3204±8.9024 | 30.3218±54.8095 |
| | PDUV_F_(3,1) | 9.4252±25.9660 | 13.1689±17.0462 | 5.0986±27.2035 | 5.3432±9.4238 |
| | PDUV_T_(3,1) | 2.6480±5.6620 | 25.1748±42.5476 | 0.9550±1.2139 | 9.1987±11.1568 |
| ViT | PDUV_F_(1,1) | 30.7485±75.0957 | 13.4051±28.2157 | 1.9576±1.9775 | 2.7012±2.0355 |
| | PDUV_T_(1,1) | 13.7787±15.8051 | 206.6721±490.4529 | 1.8299±1.7364 | 17.8208±22.1968 |
| | PDUV_F_(3,1) | 8.4927±9.5889 | 21.5649±28.8912 | 2.0578±1.8248 | 11.1058±6.4700 |
| | PDUV_T_(3,1) | 8.7343±10.8018 | 101.8180±153.4608 | 1.9404±1.7064 | 10.8021±11.9168 |
| ScOT | PDUV_F_(1,1) | 3.3588±3.5977 | 3.6852±3.2128 | 1.5700±1.4611 | 3.0450±2.3063 |
| | PDUV_T_(1,1) | 6.8205±8.3661 | 169.6440±370.9922 | 2.1943±2.2499 | 672.6685±1149.7342 |
| | PDUV_F_(3,1) | 3.4918±4.2602 | 22.0304±39.1454 | 1.8417±1.7165 | 7.0220±18.5086 |
| | PDUV_T_(3,1) | 5.2982±6.3912 | 322.9135±558.6424 | 2.3892±2.2477 | 101.1714±152.1318 |
| CNO | PDUV_F_(1,1) | 2.4282±2.7867 | 14.8098±19.7260 | 1.6192±1.9654 | 7.1715±7.9620 |
| | PDUV_T_(1,1) | 4.9821±11.5265 | 281.5538±507.0057 | 4.5075±7.8777 | 49.5062±65.8603 |
| | PDUV_F_(3,1) | 1.6514±1.7726 | 31.4155±37.0237 | 1.2797±1.4745 | 4.4545±4.1664 |
| | PDUV_T_(3,1) | 3.1501±4.0803 | 18.7740±44.7674 | 2.6809±4.0172 | 42.1340±79.9313 |
| ResNet | PDUV_F_(1,1) | 72.0392±186.8011 | 21.9601±35.3048 | | |
| | PDUV_T_(1,1) | 11.2374±19.0098 | 38.5417±50.4448 | | |
| | PDUV_F_(3,1) | 5.2020±11.0591 | 5.9362±3.9994 | | |
| | PDUV_T_(3,1) | 6.0783±8.6927 | 12.5039±20.1806 | | |
| Transolver | PDUV_F_(1,1) | 5.6585±6.2504 | 5.4874±3.5771 | | |
| | PDUV_T_(1,1) | 27.3894±25.6446 | 12.6030±15.6500 | | |
| | PDUV_F_(3,1) | 8.2177±8.3123 | 31.5978±26.8641 | | |
| | PDUV_T_(3,1) | 23.1919±17.7832 | 25.8517±31.2784 | | |

Table 26: Effect of conditioning parameters for LIDE In-Distribution (ID) and Out-of-Distribution (OOD) datasets. Error-type 2 over VP from section 4.5 is presented.

| MODEL | TAG | 1M | | 50M | |
|---|---|---|---|---|---|
| | | ID | OOD | ID | OOD d |
| UNet | PDUV_F_(1,1) | 2.2364±1.8249 | 7.6716±4.9818 | 1.1246±1.3118 | 6.6250±4.5504 |
| | PDUV_T_(1,1) | 2.6919±1.9304 | 6.7822±3.8915 | 1.0655±1.4727 | 8.4409±5.2510 |
| | PDUV_F_(3,1) | 1.4657±1.3658 | 4.2129±2.3582 | 0.5786±0.6629 | 4.8546±3.6289 |
| | PDUV_T_(3,1) | 2.0459±2.2480 | 6.5417±3.4596 | 0.9025±1.1001 | 7.4573±4.5266 |
| FNO | PDUV_F_(1,1) | 15.4477±38.8379 | 10.0575±11.7609 | 8.1643±39.6190 | 10.5453±15.8533 |
| | PDUV_T_(1,1) | 6.9028±11.2910 | 21.9416±30.2423 | 3.4257±8.3260 | 28.1019±38.8927 |
| | PDUV_F_(3,1) | 8.7514±19.1512 | 10.0428±11.7263 | 4.8705±22.6135 | 5.1937±3.2168 |
| | PDUV_T_(3,1) | 5.3098±8.0500 | 8.4202±10.7254 | 2.4009±1.7602 | 9.3546±9.0295 |
| ViT | PDUV_F_(1,1) | 35.9765±50.1389 | 10.7088±14.2735 | 2.1195±1.7062 | 6.0996±3.5933 |
| | PDUV_T_(1,1) | 12.1881±18.9528 | 64.7967±102.2082 | 1.7690±1.5420 | 7.8840±10.1525 |
| | PDUV_F_(3,1) | 13.7822±15.4450 | 11.8592±20.5723 | 1.7101±1.8497 | 3.5671±2.1539 |
| | PDUV_T_(3,1) | 12.6794±18.8376 | 82.6483±125.2612 | 1.5897±1.7060 | 3.8315±4.8235 |
| ScOT | PDUV_F_(1,1) | 3.2438±2.3717 | 7.2868±3.9963 | 1.4184±1.3680 | 5.8183±3.3720 |
| | PDUV_T_(1,1) | 4.4437±3.0729 | 10.4074±11.7337 | 1.3951±1.3767 | 24.3249±38.3291 |
| | PDUV_F_(3,1) | 2.7103±2.0030 | 6.1239±5.2132 | 0.8984±0.8154 | 4.7614±3.2915 |
| | PDUV_T_(3,1) | 3.2888±2.5007 | 4.6767±2.4503 | 1.3000±0.9832 | 9.1858±11.4396 |
| CNO | PDUV_F_(1,1) | 2.3980±1.9352 | 5.0360±2.8973 | 1.4503±1.5786 | 5.6487±3.7962 |
| | PDUV_T_(1,1) | 3.6897±3.5841 | 8.5344±9.0559 | 2.2872±2.1839 | 7.4226±4.2026 |
| | PDUV_F_(3,1) | 1.9368±1.4496 | 4.8935±2.9276 | 1.1844±1.1809 | 5.1373±3.3016 |
| | PDUV_T_(3,1) | 3.4163±3.1853 | 5.5930±3.7933 | 2.2287±2.5476 | 5.6225±3.0315 |
| ResNet | PDUV_F_(1,1) | 35.3231±75.9315 | 5.7896±4.4612 | | |
| | PDUV_T_(1,1) | 18.5294±30.0486 | 10.7451±8.9306 | | |
| | PDUV_F_(3,1) | 28.5191±65.6205 | 6.3865±7.7030 | | |
| | PDUV_T_(3,1) | 11.3432±14.3506 | 8.8084±6.2336 | | |
| Transolver | PDUV_F_(1,1) | 3.7311±4.9342 | 6.1950±3.9039 | | |
| | PDUV_T_(1,1) | 74.0015±271.6266 | 18.6063±27.1426 | | |
| | PDUV_F_(3,1) | 3.2161±4.3176 | 6.8555±5.0112 | | |
| | PDUV_T_(3,1) | 10.9489±11.6813 | 9.6702±11.7623 | | |

Table 27: Effect of conditioning parameters for LIDE In-Distribution (ID) and Out-of-Distribution (OOD) datasets. Error-type 2 over OR from section 4.5 is presented.

| MODEL | TAG | 1M | | 50M | |
|---|---|---|---|---|---|
| | | ID | OOD | ID | OOD |
| UNet | PDUV_F_(1,1) | 27.6161±83.5283 | 288.1661±361.6815 | 25.9451±78.9227 | 336.0100±416.4188 |
| | PDUV_T_(1,1) | 24.3380±73.8961 | 131.6247±268.9813 | 17.2210±58.2018 | 79.1639±218.8088 |
| | PDUV_F_(3,1) | 22.5440±64.7437 | 360.2897±422.2532 | 21.6422±62.6203 | 325.0154±380.1980 |
| | PDUV_T_(3,1) | 26.7619±76.1744 | 78.3667±208.0575 | 28.4698±82.3074 | 26.1189±97.2739 |
| FNO | PDUV_F_(1,1) | 29.6074±93.7013 | 313.8395±437.8267 | 9.9817±46.2403 | 320.7908±453.1373 |
| | PDUV_T_(1,1) | 32.7552±98.8249 | 426.3703±548.5769 | 30.9927±94.9173 | 374.0674±491.5785 |
| | PDUV_F_(3,1) | 37.0998±106.2921 | 413.0769±490.3942 | 26.7387±80.1169 | 331.4658±421.8210 |
| | PDUV_T_(3,1) | 38.9718±112.4069 | 415.5112±491.0000 | 27.6664±83.7146 | 170.6939±302.2608 |
| ViT | PDUV_F_(1,1) | 28.3570±84.8627 | 316.4301±381.2929 | 7.3652±28.7282 | 154.0128±218.9231 |
| | PDUV_T_(1,1) | 34.7868±106.0204 | 42.1618±147.9682 | 12.5737±41.5759 | 400.0511±491.7083 |
| | PDUV_F_(3,1) | 26.6128±87.9115 | 106.3928±240.9438 | 16.3596±58.4039 | 108.5414±239.3181 |
| | PDUV_T_(3,1) | 38.9649±110.5403 | 396.1875±465.9020 | 27.1187±87.2383 | 163.6112±277.0810 |
| ScOT | PDUV_F_(1,1) | 19.0514±57.4875 | 213.0902±263.9498 | 29.8124±91.7401 | 360.4875±460.3578 |
| | PDUV_T_(1,1) | 43.7381±125.7196 | 89.7623±238.1051 | 23.4326±79.5363 | 472.5784±600.7889 |
| | PDUV_F_(3,1) | 25.3996±76.6864 | 361.9640±412.0892 | 8.1928±41.8663 | 176.1184±329.3044 |
| | PDUV_T_(3,1) | 27.3812±81.1700 | 159.3863±310.1257 | 9.5601±40.0058 | 140.4161±333.8770 |
| CNO | PDUV_F_(1,1) | 20.3428±62.2283 | 407.2689±527.2424 | 19.7226±60.0229 | 261.2177±332.9367 |
| | PDUV_T_(1,1) | 19.5992±59.2119 | 170.7035±352.2779 | 12.9402±48.1851 | 28.8141±91.3571 |
| | PDUV_F_(3,1) | 22.7767±66.8182 | 163.3376±283.7012 | 19.7672±71.7488 | 91.9920±200.5455 |
| | PDUV_T_(3,1) | 21.2636±59.1526 | 159.2449±359.0494 | 11.8448±45.1271 | 147.4758±372.0305 |
| ResNet | PDUV_F_(1,1) | 48.2119±127.4267 | 443.6026±559.3933 | | |
| | PDUV_T_(1,1) | 7.3037±25.1349 | 190.7057±284.2230 | | |
| | PDUV_F_(3,1) | 50.6830±131.6622 | 507.9845±594.5909 | | |
| | PDUV_T_(3,1) | 23.5701±83.5540 | 71.7497±176.7698 | | |
| Transolver | PDUV_F_(1,1) | 50.4440±122.6158 | 479.4698±587.8039 | | |
| | PDUV_T_(1,1) | 26.2207±79.1707 | 317.7513±430.3635 | | |
| | PDUV_F_(3,1) | 54.6837±128.0420 | 312.8717±446.2977 | | |
| | PDUV_T_(3,1) | 9.0898±5.5820 | 9.7856±5.3570 | | |

Table 28: Effect of conditioning fields for LIDE In-Distribution (ID) and Out-of-Distribution (OOD) datasets. Error-type 2 over output channels from section 4.5 is presented.

| MODEL | TAG | 1M | | 50M | |
|---|---|---|---|---|---|
| | | ID | OOD | ID | OOD |
| UNet | P_F_(1,1) | 9.1478±8.0895 | 13.0943±8.7709 | 5.3090±6.7340 | 11.9953±7.5931 |
| | P[ES]_F_(1,1) | 23.0535±18.8769 | 16.0039±11.8845 | 29.6090±24.6381 | 36.0516±30.8501 |
| | P_F_(3,1) | 3.4922±3.9388 | 9.1548±6.0536 | 1.9950±2.6557 | 7.8092±7.1191 |
| | P[ES]_F_(3,1) | 13.0605±13.3488 | 20.2864±17.7559 | 11.1466±10.3795 | 19.2790±16.1179 |
| | PDUV_F_(1,1) | 2.9972±2.4756 | 9.0796±5.9230 | 2.1210±1.8598 | 8.0828±5.3407 |
| | PDUV[ES]_F_(1,1) | 7.5569±6.2939 | 10.5505±6.1862 | 9.9430±8.6244 | 11.4681±7.2812 |
| | PDUV_F_(3,1) | 1.5341±1.2526 | 6.0177±5.0874 | 1.0030±0.8511 | 5.0866±4.2680 |
| | PDUV[ES]_F_(3,1) | 5.1957±4.6966 | 9.3330±7.0384 | 4.4018±3.7968 | 8.5123±6.5495 |
| FNO | P_F_(1,1) | 6.3022±5.1934 | 10.3828±7.0152 | 5.2016±5.4331 | 12.0931±7.6183 |
| | P[ES]_F_(1,1) | 27.6326±22.1648 | 26.2085±20.8494 | 32.7032±25.4675 | 27.6014±23.5059 |
| | P_F_(3,1) | 4.6875±4.6579 | 9.5000±6.8683 | 3.6339±4.4195 | 8.7828±6.6895 |
| | P[ES]_F_(3,1) | 10.8859±8.1119 | 14.1592±10.4569 | 14.2710±10.8054 | 14.5738±10.2457 |
| | PDUV_F_(1,1) | 3.6399±3.1425 | 7.8868±5.3663 | 2.5836±2.2495 | 7.2695±5.3881 |
| | PDUV[ES]_F_(1,1) | 8.6627±6.7686 | 12.3647±8.0489 | 8.9522±7.0050 | 11.1104±6.8842 |
| | PDUV_F_(3,1) | 2.4656±2.4281 | 5.9441±4.8157 | 1.6334±1.4492 | 6.4107±5.4997 |
| | PDUV[ES]_F_(3,1) | 6.6485±5.3640 | 9.6179±6.0060 | 5.0849±4.2891 | 8.7880±5.6131 |
| ViT | P_F_(1,1) | 9.1603±5.5769 | 10.2389±5.8189 | 6.2536±7.1400 | 10.8063±7.6196 |
| | P[ES]_F_(1,1) | 19.4751±19.8881 | 17.3331±13.6654 | 12.7624±10.7407 | 15.7572±12.5764 |
| | P_F_(3,1) | 8.6900±5.5057 | 10.2917±6.1372 | 4.1896±4.9856 | 7.6857±6.3458 |
| | P[ES]_F_(3,1) | 9.2608±7.0138 | 11.8868±9.0272 | 7.3133±5.8014 | 10.2299±8.2406 |
| | PDUV_F_(1,1) | 6.7641±4.8388 | 8.5556±5.3823 | 2.1804±1.7873 | 6.3454±4.3554 |
| | PDUV[ES]_F_(1,1) | 9.2205±8.8594 | 11.4403±9.2148 | 4.3167±3.4105 | 7.1662±4.9084 |
| | PDUV_F_(3,1) | 4.1118±3.0304 | 6.4825±5.1926 | 1.4109±1.1031 | 3.9230±3.2763 |
| | PDUV[ES]_F_(3,1) | 5.4353±4.1444 | 7.7203±6.2456 | 3.8999±3.2830 | 5.4419±4.3560 |
| ScOT | P_F_(1,1) | 8.2057±6.9389 | 11.6114±7.4053 | 5.1421±6.1064 | 11.3345±7.6428 |
| | P[ES]_F_(1,1) | 21.4050±22.1739 | 22.0405±17.6376 | 16.8751±13.1522 | 15.5203±9.7885 |
| | P_F_(3,1) | 4.8706±4.7652 | 7.0995±5.8052 | 3.1855±4.2143 | 6.1995±5.6304 |
| | P[ES]_F_(3,1) | 6.2541±5.3469 | 7.7537±5.7890 | 6.0439±4.9679 | 8.0174±5.8438 |
| | PDUV_F_(1,1) | 2.8022±2.2122 | 7.4147±4.9714 | 2.2992±1.8961 | 7.0310±4.6946 |
| | PDUV[ES]_F_(1,1) | 8.5861±7.6685 | 8.7523±6.9290 | 6.4329±6.2919 | 10.0934±7.6721 |
| | PDUV_F_(3,1) | 1.7180±1.3379 | 6.4316±7.3418 | 1.1854±0.9577 | 5.1242±5.4815 |
| | PDUV[ES]_F_(3,1) | 4.2951±4.1789 | 9.7228±10.4728 | 4.2373±4.2770 | 8.4763±8.7977 |
| CNO | P_F_(1,1) | 5.8675±6.4694 | 28.5730±20.1015 | 5.5428±6.2562 | 35.9518±35.2050 |
| | P[ES]_F_(1,1) | 26.0960±22.0672 | 39.5163±34.1637 | 27.8891±22.8285 | 33.5685±28.1269 |
| | P_F_(3,1) | 2.1754±2.6216 | 10.0666±8.6685 | 2.3462±3.2362 | 7.4284±6.6611 |
| | P[ES]_F_(3,1) | 8.6612±7.3315 | 10.0280±7.1972 | 8.4329±7.1614 | 14.1459±12.8208 |
| | PDUV_F_(1,1) | 2.1786±1.7884 | 7.7764±6.8129 | 2.1271±1.7921 | 6.7606±4.9076 |
| | PDUV[ES]_F_(1,1) | 8.7094±7.9076 | 13.4745±9.9670 | 7.5888±6.7066 | 11.8868±9.0768 |
| | PDUV_F_(3,1) | 1.2258±0.9966 | 6.4241±6.3085 | 1.1850±0.9985 | 4.7852±3.9415 |
| | PDUV[ES]_F_(3,1) | 4.9778±4.7576 | 7.3581±6.0097 | 4.7545±4.2252 | 7.9192±6.5797 |
| ResNet | P_F_(1,1) | 15.8603±14.2650 | 14.5966±10.5899 | | |
| | P[ES]_F_(1,1) | 22.7663±16.9184 | 30.5975±25.0920 | | |
| | P_F_(3,1) | 9.7710±7.7492 | 10.9950±7.2943 | | |
| | P[ES]_F_(3,1) | 10.6258±7.4852 | 14.4755±10.1972 | | |
| | PDUV_F_(1,1) | 7.1277±5.7086 | 9.8022±6.0704 | | |
| | PDUV[ES]_F_(1,1) | 6.5783±4.6303 | 10.2784±6.3609 | | |
| | PDUV_F_(3,1) | 5.0517±4.3040 | 6.4456±4.5672 | | |
| | PDUV[ES]_F_(3,1) | 6.3316±4.7105 | 8.5529±5.2428 | | |
| Transolver | P_F_(1,1) | 7.7697±6.8997 | 11.3624±7.4919 | | |
| | P[ES]_F_(1,1) | 16.7733±12.4990 | 14.5373±9.9606 | | |
| | P_F_(3,1) | 3.7175±3.7892 | 6.7653±5.1699 | | |
| | P[ES]_F_(3,1) | 9.6835±7.1451 | 10.8923±8.3330 | | |
| | PDUV_F_(1,1) | 4.9459±3.6294 | 8.3506±5.2692 | | |
| | PDUV[ES]_F_(1,1) | 9.1508±7.5811 | 9.7693±7.6113 | | |
| | PDUV_F_(3,1) | 3.4782±2.3452 | 5.5601±4.1509 | | |
| | PDUV[ES]_F_(3,1) | 6.0474±4.3954 | 7.4736±5.4511 | | |

Table 29: Effect of conditioning fields for LIDE In-Distribution (ID) and Out-of-Distribution (OOD) datasets. Error-type 2 over KE from section 4.5 is presented.

| MODEL | TAG | 1M | | 50M | |
|---|---|---|---|---|---|
| | | ID | OOD | ID | OOD |
| UNet | PDUV_F_(1,1) | 2.9558±3.4607 | 7.4869±9.0017 | 0.7912±0.9653 | 8.0747±7.5999 |
| | PDUV[ES]_F_(1,1) | 20.3706±34.1137 | 10.1271±15.9696 | 85.3670±95.5683 | 153.1264±176.1153 |
| | PDUV_F_(3,1) | 1.5158±1.7694 | 5.8522±5.8268 | 0.6309±0.9576 | 4.6977±3.9619 |
| | PDUV[ES]_F_(3,1) | 16.7911±23.1446 | 106.7279±127.6807 | 7.1791±9.1559 | 38.0896±41.2699 |
| FNO | PDUV_F_(1,1) | 17.3105±56.5599 | 10.9310±16.6896 | 7.7942±39.8016 | 20.4284±40.8841 |
| | PDUV[ES]_F_(1,1) | 25.1967±25.9655 | 28.1146±31.9155 | 20.8824±31.6092 | 9.1176±11.8356 |
| | PDUV_F_(3,1) | 9.4252±25.9660 | 13.1689±17.0462 | 5.0986±27.2035 | 5.3432±9.4238 |
| | PDUV[ES]_F_(3,1) | 11.2583±9.6838 | 4.7826±3.8797 | 6.1821±7.2471 | 21.7777±30.0173 |
| ViT | PDUV_F_(1,1) | 30.7485±75.0957 | 13.4051±28.2157 | 1.9576±1.9775 | 2.7012±2.0355 |
| | PDUV[ES]_F_(1,1) | 241.4799±870.9689 | 145.0038±397.7887 | 16.0164±19.8772 | 6.9168±8.2187 |
| | PDUV_F_(3,1) | 8.4927±9.5889 | 21.5649±28.8912 | 2.0578±1.8248 | 11.1058±6.4700 |
| | PDUV[ES]_F_(3,1) | 27.0586±39.7306 | 25.4535±23.4193 | 9.8028±13.6173 | 14.4241±9.5125 |
| ScOT | PDUV_F_(1,1) | 3.3588±3.5977 | 3.6852±3.2128 | 1.5700±1.4611 | 3.0450±2.3063 |
| | PDUV[ES]_F_(1,1) | 62.5426±145.8651 | 25.8621±39.7799 | 43.8361±70.6359 | 27.6410±45.7599 |
| | PDUV_F_(3,1) | 3.4918±4.2602 | 22.0304±39.1454 | 1.8417±1.7165 | 7.0220±18.5086 |
| | PDUV[ES]_F_(3,1) | 10.6228±13.9594 | 46.5500±63.7638 | 15.5375±19.7119 | 40.3355±42.3768 |
| CNO | PDUV_F_(1,1) | 2.4282±2.7867 | 14.8098±19.7260 | 1.6192±1.9654 | 7.1715±7.9620 |
| | PDUV[ES]_F_(1,1) | 42.3454±45.6934 | 147.1579±157.3773 | 21.5265±26.4024 | 99.8242±121.7740 |
| | PDUV_F_(3,1) | 1.6514±1.7726 | 31.4155±37.0237 | 1.2797±1.4745 | 4.4545±4.1664 |
| | PDUV[ES]_F_(3,1) | 14.6887±19.6394 | 41.0990±52.3916 | 2.7269±3.3062 | 6.9642±9.2008 |
| ResNet | PDUV_F_(1,1) | 72.0392±186.8011 | 21.9601±35.3048 | | |
| | PDUV[ES]_F_(1,1) | 30.4190±41.2347 | 64.9705±64.7465 | | |
| | PDUV_F_(3,1) | 5.2020±11.0591 | 5.9362±3.9994 | | |
| | PDUV[ES]_F_(3,1) | 26.2392±33.9826 | 25.1360±32.8936 | | |
| Transolver | PDUV_F_(1,1) | 5.6585±6.2504 | 5.4874±3.5771 | | |
| | PDUV[ES]_F_(1,1) | 72.7505±228.9720 | 27.5679±43.1728 | | |
| | PDUV_F_(3,1) | 8.2177±8.3123 | 31.5978±26.8641 | | |
| | PDUV[ES]_F_(3,1) | 38.8649±37.2626 | 72.0413±61.4863 | | |

Table 30: Effect of conditioning fields for LIDE In-Distribution (ID) and Out-of-Distribution (OOD) datasets. Error-type 2 over VP from section 4.5 is presented.

| MODEL | TAG | 1M | | 50M | |
|---|---|---|---|---|---|
| | | ID | OOD | ID | OOD |
| UNet | PDUV_F_(1,1) | 2.2364±1.8249 | 7.6716±4.9818 | 1.1246±1.3118 | 6.6250±4.5504 |
| | PDUV[ES]_F_(1,1) | 5.9239±6.3310 | 6.2247±4.3015 | 9.2649±8.3096 | 15.2664±17.0709 |
| | PDUV_F_(3,1) | 1.4657±1.3658 | 4.2129±2.3582 | 0.5786±0.6629 | 4.8546±3.6289 |
| | PDUV[ES]_F_(3,1) | 3.7189±4.2770 | 18.7705±29.1679 | 2.3040±2.8389 | 8.0838±8.7216 |
| FNO | PDUV_F_(1,1) | 15.4477±38.8379 | 10.0575±11.7609 | 8.1643±39.6190 | 10.5453±15.8533 |
| | PDUV[ES]_F_(1,1) | 7.6566±9.2270 | 6.3084±3.6235 | 6.2573±7.7260 | 6.8818±7.8352 |
| | PDUV_F_(3,1) | 8.7514±19.1512 | 10.0428±11.7263 | 4.8705±22.6135 | 5.1937±3.2168 |
| | PDUV[ES]_F_(3,1) | 4.1110±4.3578 | 4.6076±3.7051 | 2.6139±2.7369 | 20.8548±24.9235 |
| ViT | PDUV_F_(1,1) | 35.9765±50.1389 | 10.7088±14.2735 | 2.1195±1.7062 | 6.0996±3.5933 |
| | PDUV[ES]_F_(1,1) | 116.2920±166.9602 | 51.8742±99.6273 | 5.3122±6.8443 | 5.0159±5.2584 |
| | PDUV_F_(3,1) | 13.7822±15.4450 | 11.8592±20.5723 | 1.7101±1.8497 | 3.5671±2.1539 |
| | PDUV[ES]_F_(3,1) | 31.3124±44.7175 | 21.5597±25.9508 | 5.0579±7.6161 | 5.0811±7.9904 |
| ScOT | PDUV_F_(1,1) | 3.2438±2.3717 | 7.2868±3.9963 | 1.4184±1.3680 | 5.8183±3.3720 |
| | PDUV[ES]_F_(1,1) | 6.2914±11.0882 | 6.8750±4.1989 | 8.1067±12.0643 | 7.6245±5.7289 |
| | PDUV_F_(3,1) | 2.7103±2.0030 | 6.1239±5.2132 | 0.8984±0.8154 | 4.7614±3.2915 |
| | PDUV[ES]_F_(3,1) | 2.9604±3.3728 | 6.7903±6.5895 | 2.5907±3.6683 | 4.8924±3.7627 |
| CNO | PDUV_F_(1,1) | 2.3980±1.9352 | 5.0360±2.8973 | 1.4503±1.5786 | 5.6487±3.7962 |
| | PDUV[ES]_F_(1,1) | 5.6469±6.5604 | 7.1679±7.5588 | 4.5431±5.7530 | 6.5345±7.5133 |
| | PDUV_F_(3,1) | 1.9368±1.4496 | 4.8935±2.9276 | 1.1844±1.1809 | 5.1373±3.3016 |
| | PDUV[ES]_F_(3,1) | 2.5447±2.8231 | 4.2803±2.7221 | 2.1570±2.6265 | 4.6820±3.2973 |
| ResNet | PDUV_F_(1,1) | 35.3231±75.9315 | 5.7896±4.4612 | | |
| | PDUV[ES]_F_(1,1) | 66.6398±120.4167 | 21.7128±30.8889 | | |
| | PDUV_F_(3,1) | 28.5191±65.6205 | 6.3865±7.7030 | | |
| | PDUV[ES]_F_(3,1) | 29.7433±46.1951 | 7.7186±9.6800 | | |
| Transolver | PDUV_F_(1,1) | 3.7311±4.9342 | 6.1950±3.9039 | | |
| | PDUV[ES]_F_(1,1) | 48.7057±82.3430 | 19.3961±24.8031 | | |
| | PDUV_F_(3,1) | 3.2161±4.3176 | 6.8555±5.0112 | | |
| | PDUV[ES]_F_(3,1) | 6.6554±8.8829 | 8.4493±14.1979 | | |

Table 31: Effect of conditioning fields for LIDE In-Distribution (ID) and Out-of-Distribution (OOD) datasets. Error-type 2 over OR from section 4.5 is presented.

| MODEL | TAG | 1M | | 50M | |
|---|---|---|---|---|---|
| | | ID | OOD | ID | OOD |
| UNet | PDUV_F_(1,1) | 27.6161±83.5283 | 288.1661±361.6815 | 25.9451±78.9227 | 336.0100±416.4188 |
| | PDUV[ES]_F_(1,1) | 10.1121±13.9343 | 12.7286±27.3411 | 19.4516±58.1035 | 27.6148±52.5871 |
| | PDUV_F_(3,1) | 22.5440±64.7437 | 360.2897±422.2532 | 21.6422±62.6203 | 325.0154±380.1980 |
| | PDUV[ES]_F_(3,1) | 21.5989±67.8911 | 68.7352±141.6686 | 18.9192±53.5520 | 74.0707±178.7413 |
| FNO | PDUV_F_(1,1) | 29.6074±93.7013 | 313.8395±437.8267 | 9.9817±46.2403 | 320.7908±453.1373 |
| | PDUV[ES]_F_(1,1) | 42.1162±111.6572 | 389.1060±525.8061 | 13.9013±41.0257 | 100.3914±190.0753 |
| | PDUV_F_(3,1) | 37.0998±106.2921 | 413.0769±490.3942 | 26.7387±80.1169 | 331.4658±421.8210 |
| | PDUV[ES]_F_(3,1) | 11.3997±36.5709 | 65.5979±245.8531 | 17.4764±49.3967 | 227.7179±364.7697 |
| ViT | PDUV_F_(1,1) | 28.3570±84.8627 | 316.4301±381.2929 | 7.3652±28.7282 | 154.0128±218.9231 |
| | PDUV[ES]_F_(1,1) | 24.9565±69.7981 | 240.2715±326.7355 | 19.9529±60.1380 | 212.9958±261.6513 |
| | PDUV_F_(3,1) | 26.6128±87.9115 | 106.3928±240.9438 | 16.3596±58.4039 | 108.5414±239.3181 |
| | PDUV[ES]_F_(3,1) | 11.2870±38.3220 | 171.8278±256.1028 | 6.3234±11.7187 | 105.0288±216.5524 |
| ScOT | PDUV_F_(1,1) | 19.0514±57.4875 | 213.0902±263.9498 | 29.8124±91.7401 | 360.4875±460.3578 |
| | PDUV[ES]_F_(1,1) | 28.4080±86.3271 | 297.9673±404.0781 | 19.1925±58.2088 | 216.5683±272.0256 |
| | PDUV_F_(3,1) | 25.3996±76.6864 | 361.9640±412.0892 | 8.1928±41.8663 | 176.1184±329.3044 |
| | PDUV[ES]_F_(3,1) | 33.4385±101.7622 | 333.1158±415.8457 | 21.1220±59.9334 | 271.1644±309.5756 |
| CNO | PDUV_F_(1,1) | 20.3428±62.2283 | 407.2689±527.2424 | 19.7226±60.0229 | 261.2177±332.9367 |
| | PDUV[ES]_F_(1,1) | 34.5318±101.6591 | 298.9503±438.7167 | 16.0693±48.4838 | 71.8442±107.9373 |
| | PDUV_F_(3,1) | 22.7767±66.8182 | 163.3376±283.7012 | 19.7672±71.7488 | 91.9920±200.5455 |
| | PDUV[ES]_F_(3,1) | 7.3676±11.9052 | 9.7813±5.3788 | 17.2166±48.6478 | 207.3200±232.2929 |
| ResNet | PDUV_F_(1,1) | 48.2119±127.4267 | 443.6026±559.3933 | | |
| | PDUV[ES]_F_(1,1) | 20.2926±66.2371 | 74.7534±168.2826 | | |
| | PDUV_F_(3,1) | 50.6830±131.6622 | 507.9845±594.5909 | | |
| | PDUV[ES]_F_(3,1) | 5.0466±4.8902 | 8.6086±7.1586 | | |
| Transolver | PDUV_F_(1,1) | 50.4440±122.6158 | 479.4698±587.8039 | | |
| | PDUV[ES]_F_(1,1) | 11.5269±6.4621 | 315.8390±533.3240 | | |
| | PDUV_F_(3,1) | 54.6837±128.0420 | 312.8717±446.2977 | | |
| | PDUV[ES]_F_(3,1) | 15.6974±29.7730 | 306.3072±530.8621 | | |

## C.2 ERROR-TYPE 2 METRICS FOR THE SIDA DATASET

Table 32: Effect of sequence information for SIDA In-Distribution (ID) and Out-of-Distribution (OOD) datasets. Error-type 2 over output channels from section 4.5 is presented.

| MODEL | TAG | 1M | | 50M | |
|---|---|---|---|---|---|
| | | ID | OOD | ID | OOD |
| UNet | PDUV_F_(1,1) | 3.5661±2.4696 | 5.9914±3.2792 | 3.4983±2.5270 | 5.9438±3.2522 |
| | PDUV_F_(3,1) | **0.3383±0.2089** | 2.2998±1.7754 | **0.1356±0.0976** | 1.9424±1.5948 |
| | PDUV_F_(3,2) | 0.4406±0.2437 | **2.1979±1.6703** | 0.1785±0.1080 | **1.8812±1.4977** |
| FNO | PDUV_F_(1,1) | 3.5224±2.5528 | 6.0820±3.2896 | 3.4720±2.5577 | 5.9768±3.2572 |
| | PDUV_F_(3,1) | **0.5333±0.2829** | **2.1215±1.5319** | **0.2498±0.1387** | **2.0963±1.5242** |
| | PDUV_F_(3,2) | 0.7123±0.3624 | 2.1449±1.5747 | 0.3389±0.1827 | 2.0990±1.5458 |
| ViT | PDUV_F_(1,1) | 4.4758±2.8750 | 5.8219±3.1222 | 3.5527±2.7160 | 5.7645±3.1503 |
| | PDUV_F_(3,1) | 1.5891±0.9548 | 3.4253±2.4145 | **0.4430±0.2336** | **1.8428±1.3876** |
| | PDUV_F_(3,2) | **1.3832±0.7495** | **2.4443±1.6647** | 0.5441±0.2699 | 1.8678±1.3767 |
| ScOT | PDUV_F_(1,1) | 3.5764±2.5158 | 5.8985±3.2421 | 3.5173±2.6462 | 5.8306±3.1826 |
| | PDUV_F_(3,1) | **0.6884±0.4222** | **1.9005±1.3770** | **0.2059±0.1199** | 1.6318±1.3494 |
| | PDUV_F_(3,2) | 0.7995±0.4265 | 1.9084±1.3867 | 0.2639±0.1459 | **1.5164±1.2414** |
| CNO | PDUV_F_(1,1) | 3.9955±2.6786 | 5.8508±3.1777 | 3.5025±2.5455 | 5.9444±3.2344 |
| | PDUV_F_(3,1) | **0.6943±0.4235** | 2.1707±1.8099 | **0.1621±0.1032** | 1.8159±1.4794 |
| | PDUV_F_(3,2) | 0.8584±0.4459 | **1.8360±1.3758** | 0.2210±0.1248 | **1.7634±1.3875** |
| ResNet | PDUV_F_(1,1) | 3.7260±2.4574 | 6.0844±3.3406 | | |
| | PDUV_F_(3,1) | **0.5570±0.3461** | 2.3042±1.7139 | | |
| | PDUV_F_(3,2) | 0.6356±0.3669 | **2.3012±1.6856** | | |
| Transolver | PDUV_F_(1,1) | 5.7995±3.4502 | 6.9499±3.7154 | | |
| | PDUV_F_(3,1) | **4.7065±2.6888** | **4.8852±2.7289** | | |
| | PDUV_F_(3,2) | 5.2026±2.7770 | 5.1580±2.7698 | | |

Table 33: Effect of sequence information for SIDA In-Distribution (ID) and Out-of-Distribution (OOD) datasets. Error-type 2 over KE from section 4.5 is presented.

| MODEL | TAG | 1M | | 50M | |
|---|---|---|---|---|---|
| | | ID | OOD | ID | OOD |
| UNet | PDUV_F_(1,1) | 11.2017±29.5656 | 5.6983±3.0230 | 11.7372±31.0932 | 5.6613±2.9967 |
| | PDUV_F_(3,1) | 0.5052±0.2615 | 2.5368±1.0710 | 0.1532±0.0897 | 1.4758±0.8257 |
| | PDUV_F_(3,2) | 0.9620±0.6433 | 9.6801±5.2064 | 0.4389±0.2677 | 2.2230±1.1932 |
| FNO | PDUV_F_(1,1) | 10.8110±27.7074 | 5.7480±3.0231 | 10.7965±28.7023 | 5.6957±2.9859 |
| | PDUV_F_(3,1) | 0.7682±0.5172 | 1.9919±1.3654 | 0.2823±0.1765 | 1.5754±1.9135 |
| | PDUV_F_(3,2) | 1.2933±0.6185 | 1.5242±0.8227 | 0.5000±0.3997 | 2.7004±1.5806 |
| ViT | PDUV_F_(1,1) | 31.9704±87.7762 | 4.0911±2.0809 | 15.7706±44.1921 | 5.3135±2.7717 |
| | PDUV_F_(3,1) | 6.6527±8.3450 | 32.5122±28.4652 | 0.9694±0.6501 | 3.3619±1.4849 |
| | PDUV_F_(3,2) | 5.3889±3.2322 | 5.4872±2.9669 | 1.9649±1.1585 | 2.7149±1.9905 |
| ScOT | PDUV_F_(1,1) | 16.2807±44.8844 | 5.3495±2.8561 | 15.2877±42.6846 | 5.3579±2.8067 |
| | PDUV_F_(3,1) | 2.6928±1.9141 | 2.5941±1.1618 | 1.0306±0.6112 | 2.1724±0.9115 |
| | PDUV_F_(3,2) | 4.2225±2.2977 | 9.6165±5.0281 | 1.7483±0.9528 | 5.3631±2.7718 |
| CNO | PDUV_F_(1,1) | 17.6490±49.1966 | 5.0935±2.5799 | 12.7633±35.2539 | 5.5010±2.8628 |
| | PDUV_F_(3,1) | 3.9209±2.3251 | 10.1072±9.3279 | 0.4011±0.2343 | 1.3602±0.6405 |
| | PDUV_F_(3,2) | 4.9192±3.6170 | 20.5139±13.7615 | 0.8113±0.5196 | 1.3482±0.7874 |
| ResNet | PDUV_F_(1,1) | 10.9262±28.4670 | 5.7403±3.0356 | | |
| | PDUV_F_(3,1) | 0.8521±0.5177 | 1.9815±0.8412 | | |
| | PDUV_F_(3,2) | 1.4069±0.6998 | 5.6830±3.2346 | | |
| Transolver | PDUV_F_(1,1) | 21.6155±58.9766 | 5.0188±2.6784 | | |
| | PDUV_F_(3,1) | 67.9430±41.9429 | 59.5887±20.7706 | | |
| | PDUV_F_(3,2) | 181.1406±151.2849 | 96.0656±48.8018 | | |

Table 34: Effect of sequence information for SIDA In-Distribution (ID) and Out-of-Distribution (OOD) datasets. Error-type 2 over VP from section 4.5 is presented.

| MODEL | TAG | 1M | | 50M | |
|---|---|---|---|---|---|
| | | ID | OOD | ID | OOD |
| UNet | PDUV_F_(1,1) | 4.8548±9.7684 | 5.1872±2.8746 | 5.1272±10.2187 | 5.1860±2.8710 |
| | PDUV_F_(3,1) | 0.4678±0.5580 | 1.9011±1.4640 | 0.1094±0.1362 | 2.0724±1.6647 |
| | PDUV_F_(3,2) | 0.7461±1.0477 | 2.4576±1.7732 | 0.3118±0.3532 | 2.1101±1.7997 |
| FNO | PDUV_F_(1,1) | 4.6886±8.4082 | 5.4445±2.9026 | 4.7255±9.1404 | 5.3154±2.8825 |
| | PDUV_F_(3,1) | 0.9580±0.6336 | 2.1666±1.2573 | 0.4874±0.4576 | 1.7614±1.5075 |
| | PDUV_F_(3,2) | 1.1520±0.7034 | 2.7981±1.6515 | 0.6015±0.4159 | 1.3866±0.9442 |
| ViT | PDUV_F_(1,1) | 28.3750±59.3870 | 2.6103±1.1959 | 5.4192±11.6520 | 5.0160±2.7830 |
| | PDUV_F_(3,1) | 13.1278±16.1199 | 4.9536±5.5894 | 0.5882±0.9129 | 1.4598±2.5938 |
| | PDUV_F_(3,2) | 4.1804±4.5308 | 2.1003±1.7241 | 0.8756±1.1214 | 1.7605±2.6311 |
| ScOT | PDUV_F_(1,1) | 5.0246±10.1023 | 5.1421±2.8596 | 5.4551±12.0375 | 4.9934±2.7667 |
| | PDUV_F_(3,1) | 1.6140±2.9631 | 1.4442±0.6758 | 0.4774±0.8474 | 1.1108±0.9826 |
| | PDUV_F_(3,2) | 1.3736±2.0323 | 0.7779±0.5431 | 0.8046±1.0890 | 0.7494±0.6484 |
| CNO | PDUV_F_(1,1) | 5.0839±10.2831 | 5.0351±2.8432 | 5.0165±10.3994 | 5.0809±2.8365 |
| | PDUV_F_(3,1) | 1.9935±6.3802 | 0.9771±0.7020 | 0.2560±0.2127 | 2.0421±1.5173 |
| | PDUV_F_(3,2) | 3.4611±6.6855 | 2.8181±1.6026 | 0.5898±0.4428 | 2.1937±1.6745 |
| ResNet | PDUV_F_(1,1) | 5.1266±10.6196 | 5.1755±2.8330 | | |
| | PDUV_F_(3,1) | 0.5801±0.4023 | 1.6962±1.3323 | | |
| | PDUV_F_(3,2) | 0.6297±0.4655 | 2.2821±1.5949 | | |
| Transolver | PDUV_F_(1,1) | 6.9632±15.3887 | 5.0256±2.5608 | | |
| | PDUV_F_(3,1) | 42.3382±30.2457 | 23.1503±16.1705 | | |
| | PDUV_F_(3,2) | 11.3000±14.7913 | 1.8346±1.1888 | | |

Table 35: Effect of sequence information for SIDA In-Distribution (ID) and Out-of-Distribution (OOD) datasets. Error-type 2 over COM from section 4.5 is presented.

| MODEL | TAG | 1M | | 50M | |
|---|---|---|---|---|---|
| | | ID | OOD | ID | OOD |
| UNet | PDUV_F_(1,1) | 0.2325±0.2905 | 0.9394±0.9670 | 0.2235±0.3098 | 0.9910±0.9754 |
| | PDUV_F_(3,1) | 0.0228±0.0126 | 0.5918±0.4201 | 0.0088±0.0047 | 0.7818±0.7129 |
| | PDUV_F_(3,2) | 0.0566±0.0265 | 0.4628±0.4387 | 0.0251±0.0148 | 0.5248±0.4867 |
| FNO | PDUV_F_(1,1) | 0.2377±0.3125 | 1.0179±0.9760 | 0.2188±0.2944 | 0.9605±0.9621 |
| | PDUV_F_(3,1) | 0.0271±0.0130 | 0.2420±0.3222 | 0.0118±0.0078 | 0.2427±0.2805 |
| | PDUV_F_(3,2) | 0.0561±0.0319 | 0.2477±0.3338 | 0.0267±0.0141 | 0.2091±0.2934 |
| ViT | PDUV_F_(1,1) | 0.3214±0.2879 | 0.8025±0.8487 | 0.2269±0.2774 | 0.9255±0.9327 |
| | PDUV_F_(3,1) | 0.1844±0.1821 | 1.0956±0.8733 | 0.0326±0.0211 | 0.2025±0.2268 |
| | PDUV_F_(3,2) | 0.2007±0.1252 | 0.4172±0.3822 | 0.0751±0.0418 | 0.1544±0.1388 |
| ScOT | PDUV_F_(1,1) | 0.2393±0.3107 | 0.9747±0.9872 | 0.2238±0.2732 | 0.9072±0.9265 |
| | PDUV_F_(3,1) | 0.0654±0.0359 | 0.3114±0.3727 | 0.0360±0.0209 | 0.2175±0.2388 |
| | PDUV_F_(3,2) | 0.1993±0.0869 | 0.3583±0.2814 | 0.0623±0.0293 | 0.2439±0.3005 |
| CNO | PDUV_F_(1,1) | 0.4844±0.6753 | 0.6387±0.6294 | 0.2202±0.2753 | 0.9190±0.9273 |
| | PDUV_F_(3,1) | 0.1067±0.0623 | 0.5085±0.6775 | 0.0152±0.0097 | 0.2450±0.2505 |
| | PDUV_F_(3,2) | 0.1791±0.1004 | 0.3669±0.2622 | 0.0337±0.0189 | 0.2031±0.2339 |
| ResNet | PDUV_F_(1,1) | 0.2284±0.3093 | 0.9806±0.9846 | | |
| | PDUV_F_(3,1) | 0.0387±0.0329 | 0.7315±0.6383 | | |
| | PDUV_F_(3,2) | 0.0729±0.0535 | 0.4976±0.3697 | | |
| Transolver | PDUV_F_(1,1) | 0.6920±0.4419 | 0.7175±0.7466 | | |
| | PDUV_F_(3,1) | 2.9121±1.6738 | 2.3465±0.9107 | | |
| | PDUV_F_(3,2) | 4.0355±2.0120 | 2.7159±1.0297 | | |

Table 36: Effect of conditioning parameters for SIDA In-Distribution (ID) and Out-of-Distribution (OOD) datasets. Error-type 2 over output channels from section 4.5 is presented.

| MODEL | TAG | 1M | | 50M | |
|---|---|---|---|---|---|
| | | ID | OOD | ID | OOD |
| UNet | PDUV_F_(3,1) | 0.3383±0.2089 | 2.2998±1.7754 | 0.1356±0.0976 | 1.9424±1.5948 |
| | PDUV_T_(3,1) | 0.8690±0.4994 | 2.5683±1.8388 | 0.2272±0.1439 | 2.2561±1.7777 |
| | PDUV[VoS]_F_(3,1) | 1.4203±1.1512 | 2.6100±1.8673 | 1.2162±1.0310 | 2.4740±1.8269 |
| | PDUV[VoS]_T_(3,1) | 1.6310±1.2358 | 2.7767±1.8269 | 1.1111±0.9569 | 2.4399±1.7482 |
| FNO | PDUV_F_(3,1) | 0.5333±0.2829 | 2.1215±1.5319 | 0.2498±0.1387 | 2.0963±1.5242 |
| | PDUV_T_(3,1) | 0.7013±0.3424 | 1.8361±1.2131 | 0.4075±0.2168 | 1.7610±1.2616 |
| | PDUV[VoS]_F_(3,1) | 1.8914±1.3113 | 2.6203±1.7532 | 1.7002±1.3234 | 2.6674±1.8160 |
| | PDUV[VoS]_T_(3,1) | 1.8383±1.2061 | 2.6599±1.7452 | 1.6667±1.0564 | 2.4769±1.6513 |
| ViT | PDUV_F_(3,1) | 1.5891±0.9548 | 3.4253±2.4145 | 0.4430±0.2336 | 1.8428±1.3876 |
| | PDUV_T_(3,1) | 1.5949±1.0077 | 2.6376±1.7133 | 0.4249±0.2343 | 1.7359±1.3060 |
| | PDUV[VoS]_F_(3,1) | 2.2152±1.5536 | 3.2082±2.2406 | 1.3497±1.0659 | 2.3674±1.7135 |
| | PDUV[VoS]_T_(3,1) | 2.0499±1.4162 | 3.0158±1.9680 | 1.2139±0.9433 | 2.0682±1.4535 |
| ScOT | PDUV_F_(3,1) | 0.6884±0.4222 | 1.9005±1.3770 | 0.2059±0.1199 | 1.6318±1.3494 |
| | PDUV_T_(3,1) | 0.8203±0.4715 | 1.9132±1.2731 | 0.2461±0.1397 | 1.5425±1.1975 |
| | PDUV[VoS]_F_(3,1) | 1.5944±1.2864 | 2.4643±1.7766 | 1.2171±1.0416 | 2.1314±1.5666 |
| | PDUV[VoS]_T_(3,1) | 1.4401±1.0497 | 2.2995±1.5283 | 1.1764±1.0238 | 1.9241±1.3649 |
| CNO | PDUV_F_(3,1) | 0.6943±0.4235 | 2.1707±1.8099 | 0.1621±0.1032 | 1.8159±1.4794 |
| | PDUV_T_(3,1) | 0.9362±0.5384 | 66.4331±453.5010 | 0.3042±0.1905 | 2.3211±1.8495 |
| | PDUV[VoS]_F_(3,1) | 1.1775±0.8980 | 2.1291±1.4489 | 1.2240±0.8859 | 2.3550±1.6215 |
| | PDUV[VoS]_T_(3,1) | 1.3235±0.9569 | 7898e1±6549e2 | 0.9654±0.7867 | 2.6471±1.9216 |
| ResNet | PDUV_F_(3,1) | 0.5570±0.3461 | 2.3042±1.7139 | | |
| | PDUV_T_(3,1) | 1.0385±0.6198 | 3.0052±2.1978 | | |
| | PDUV[VoS]_F_(3,1) | 1.3341±1.0333 | 2.6392±1.8649 | | |
| | PDUV[VoS]_T_(3,1) | 1.6636±1.2058 | 3.2900±2.3078 | | |
| Transolver | PDUV_F_(3,1) | 4.7065±2.6888 | 4.8852±2.7289 | | |
| | PDUV_T_(3,1) | 4.7696±2.5778 | 4.6372±2.5046 | | |
| | PDUV[VoS]_F_(3,1) | 4.9803±2.8336 | 5.2035±2.9156 | | |
| | PDUV[VoS]_T_(3,1) | 4.7139±2.5053 | 4.6158±2.4665 | | |

Table 37: Effect of conditioning parameters for SIDA In-Distribution (ID) and Out-of-Distribution (OOD) datasets. Error-type 2 over KE from section 4.5 is presented.

| MODEL | TAG | 1M | | 50M | |
|---|---|---|---|---|---|
| | | ID | OOD | ID | OOD |
| UNet | PDUV_F_(3,1) | 0.5052±0.2615 | 2.5368±1.0710 | 0.1532±0.0897 | 1.4758±0.8257 |
| | PDUV_T_(3,1) | 1.3388±0.8539 | 9.7208±7.6245 | 0.7025±0.5759 | 11.4778±8.3256 |
| | PDUV[VoS]_F_(3,1) | 1.4473±1.0509 | 2.3929±1.6705 | 1.0164±0.8202 | 1.5336±0.9003 |
| | PDUV[VoS]_T_(3,1) | 1.6124±1.4219 | 13.7084±11.9033 | 3.1509±2.2207 | 19.8637±19.5043 |
| FNO | PDUV_F_(3,1) | 0.7682±0.5172 | 1.9919±1.3654 | 0.2823±0.1765 | 1.5754±1.9135 |
| | PDUV_T_(3,1) | 1.1989±1.1997 | 12.2212±8.6412 | 0.9527±1.0156 | 12.7779±10.6585 |
| | PDUV[VoS]_F_(3,1) | 8.5964±8.0247 | 7.1201±3.8520 | 4.9123±4.6565 | 14.2872±7.8378 |
| | PDUV[VoS]_T_(3,1) | 8.0715±11.6310 | 8.5574±5.6989 | 13.0682±10.7428 | 9.9693±6.6357 |
| ViT | PDUV_F_(3,1) | 6.6527±8.3450 | 32.5122±28.4652 | 0.9694±0.6501 | 3.3619±1.4849 |
| | PDUV_T_(3,1) | 5.3717±4.3670 | 10.2786±6.7211 | 1.0290±0.6755 | 5.2700±3.4986 |
| | PDUV[VoS]_F_(3,1) | 5.5548±5.2277 | 20.9420±21.4748 | 0.9345±0.7133 | 2.2616±0.7760 |
| | PDUV[VoS]_T_(3,1) | 7.3620±6.8146 | 6.6430±5.4757 | 1.6828±1.2214 | 6.3459±5.0306 |
| ScOT | PDUV_F_(3,1) | 2.6928±1.9141 | 2.5941±1.1618 | 1.0306±0.6112 | 2.1724±0.9115 |
| | PDUV_T_(3,1) | 3.9887±3.2842 | 11.9888±7.7975 | 1.8979±1.8130 | 5.5654±4.9727 |
| | PDUV[VoS]_F_(3,1) | 2.7987±2.8603 | 8.5151±4.4987 | 8.1992±5.7868 | 6.2404±3.3768 |
| | PDUV[VoS]_T_(3,1) | 6.5291±5.5513 | 5.6833±4.0009 | 9.0977±6.0278 | 22.1760±19.9296 |
| CNO | PDUV_F_(3,1) | 3.9209±2.3251 | 10.1072±9.3279 | 0.4011±0.2343 | 1.3602±0.6405 |
| | PDUV_T_(3,1) | 5.8777±6.1544 | 2.4288e8±2.9168e9 | 1.2505±0.9845 | 2.4099±2.0305 |
| | PDUV[VoS]_F_(3,1) | 4.3996±2.7430 | 24.1448±21.7419 | 6.0637±3.7637 | 6.9987±3.5410 |
| | PDUV[VoS]_T_(3,1) | 6.5098±6.0036 | — | 2.1665±2.1864 | 10.1886±11.2716 |
| ResNet | PDUV_F_(3,1) | 0.8521±0.5177 | 1.9815±0.8412 | | |
| | PDUV_T_(3,1) | 1.5262±1.4191 | 28.0582±21.1096 | | |
| | PDUV[VoS]_F_(3,1) | 1.7642±1.3131 | 2.4584±1.8013 | | |
| | PDUV[VoS]_T_(3,1) | 2.7497±2.3888 | 34.0870±30.7161 | | |
| Transolver | PDUV_F_(3,1) | 67.9430±41.9429 | 59.5887±20.7706 | | |
| | PDUV_T_(3,1) | 93.0350±110.8223 | 258.0370±171.4161 | | |
| | PDUV[VoS]_F_(3,1) | 83.0077±119.1757 | 35.9022±11.1758 | | |
| | PDUV[VoS]_T_(3,1) | 102.3762±95.8241 | 162.3676±90.5726 | | |

Table 38: Effect of conditioning parameters for SIDA In-Distribution (ID) and Out-of-Distribution (OOD) datasets. Error-type 2 over VP from section 4.5 is presented.

| MODEL | TAG | 1M | | 50M | |
|---|---|---|---|---|---|
| | | ID | OOD | ID | OOD |
| UNet | PDUV_F_(3,1) | 0.4678±0.5580 | 1.9011±1.4640 | 0.1094±0.1362 | 2.0724±1.6647 |
| | PDUV_T_(3,1) | 1.0717±1.8190 | 2.0873±2.1121 | 0.2973±0.2529 | 2.6720±2.2034 |
| | PDUV[VoS]_F_(3,1) | 1.2571±1.2591 | 1.2459±1.0752 | 0.5621±0.4353 | 2.3172±1.6960 |
| | PDUV[VoS]_T_(3,1) | 1.7996±1.7697 | 1.7940±1.4594 | 0.7514±1.5558 | 2.4527±2.0558 |
| FNO | PDUV_F_(3,1) | 0.9580±0.6336 | 2.1666±1.2573 | 0.4874±0.4576 | 1.7614±1.5075 |
| | PDUV_T_(3,1) | 1.3719±0.7673 | 2.7430±1.5973 | 0.6992±0.5029 | 1.5961±1.7109 |
| | PDUV[VoS]_F_(3,1) | 4.3073±8.5326 | 1.8055±1.3533 | 3.0148±4.1413 | 1.2411±0.9617 |
| | PDUV[VoS]_T_(3,1) | 1.4985±1.5211 | 3.7245±2.1666 | 1.4679±1.1389 | 3.9079±2.2991 |
| ViT | PDUV_F_(3,1) | 13.1278±16.1199 | 4.9536±5.5894 | 0.5882±0.9129 | 1.4598±2.5938 |
| | PDUV_T_(3,1) | 10.3206±10.3850 | 11.4467±13.3514 | 0.5533±0.6822 | 3.4015±5.2220 |
| | PDUV[VoS]_F_(3,1) | 17.3733±19.1933 | 5.2368±4.1392 | 5.3010±4.7988 | 2.2461±2.3365 |
| | PDUV[VoS]_T_(3,1) | 17.9269±18.2700 | 7.3386±7.3888 | 3.6743±3.3607 | 4.4309±5.3252 |
| ScOT | PDUV_F_(3,1) | 1.6140±2.9631 | 1.4442±0.6758 | 0.4774±0.8474 | 1.1108±0.9826 |
| | PDUV_T_(3,1) | 1.0985±1.9509 | 1.0331±0.7909 | 0.8093±1.4573 | 0.6340±0.6392 |
| | PDUV[VoS]_F_(3,1) | 2.3244±3.7987 | 1.2487±0.8233 | 2.5129±2.5532 | 0.7567±0.4354 |
| | PDUV[VoS]_T_(3,1) | 1.5664±2.8871 | 0.7565±0.4219 | 1.7867±3.6712 | 2.1391±2.2486 |
| CNO | PDUV_F_(3,1) | 1.9935±6.3802 | 0.9771±0.7020 | 0.2560±0.2127 | 2.0421±1.5173 |
| | PDUV_T_(3,1) | 3.3135±8.2291 | 8.6846e5±9.7652e6 | 0.4813±0.5887 | 1.7157±1.4923 |
| | PDUV[VoS]_F_(3,1) | 2.5010±3.3056 | 2.3219±1.6920 | 2.2471±5.3269 | 1.5615±1.2442 |
| | PDUV[VoS]_T_(3,1) | 2.3146±5.0735 | 1.3962e12±1.4557e13 | 0.9616±2.4379 | 2.9383±2.3800 |
| ResNet | PDUV_F_(3,1) | 0.5801±0.4023 | 1.6962±1.3323 | | |
| | PDUV_T_(3,1) | 0.8478±0.6680 | 3.0296±2.2831 | | |
| | PDUV[VoS]_F_(3,1) | 1.1306±0.5625 | 1.2508±1.0298 | | |
| | PDUV[VoS]_T_(3,1) | 2.4775±2.0845 | 2.1219±1.5490 | | |
| Transolver | PDUV_F_(3,1) | 42.3382±30.2457 | 23.1503±16.1705 | | |
| | PDUV_T_(3,1) | 27.5997±22.3968 | 27.7379±18.3586 | | |
| | PDUV[VoS]_F_(3,1) | 19.9930±27.6070 | 11.1969±3.1268 | | |
| | PDUV[VoS]_T_(3,1) | 27.7826±22.1833 | 16.0020±15.6628 | | |

Table 39: Effect of conditioning parameters for SIDA In-Distribution (ID) and Out-of-Distribution (OOD) datasets. Error-type 2 over COM from section 4.5 is presented.

| MODEL | TAG | 1M | | 50M | |
|---|---|---|---|---|---|
| | | ID | OOD | ID | OOD |
| UNet | PDUV_F_(3,1) | 0.0228±0.0126 | 0.5918±0.4201 | 0.0088±0.0047 | 0.7818±0.7129 |
| | PDUV_T_(3,1) | 0.0683±0.0405 | 1.8477±1.8043 | 0.0331±0.0224 | 0.5298±0.4377 |
| | PDUV[VoS]_F_(3,1) | 0.2094±0.3003 | 0.4709±0.4614 | 0.1798±0.2596 | 0.6587±0.5563 |
| | PDUV[VoS]_T_(3,1) | 0.1570±0.1729 | 1.0825±0.8026 | 0.1380±0.1897 | 0.9711±0.7603 |
| FNO | PDUV_F_(3,1) | 0.0271±0.0130 | 0.2420±0.3222 | 0.0118±0.0078 | 0.2427±0.2805 |
| | PDUV_T_(3,1) | 0.0486±0.0462 | 0.5155±0.4017 | 0.0386±0.0311 | 0.4851±0.4057 |
| | PDUV[VoS]_F_(3,1) | 0.2230±0.1315 | 0.5460±0.4013 | 0.1534±0.1235 | 0.8256±0.5094 |
| | PDUV[VoS]_T_(3,1) | 0.2803±0.1964 | 0.6526±0.5474 | 0.4330±0.2868 | 0.8936±0.5447 |
| ViT | PDUV_F_(3,1) | 0.1844±0.1821 | 1.0956±0.8733 | 0.0326±0.0211 | 0.2025±0.2268 |
| | PDUV_T_(3,1) | 0.2011±0.2260 | 1.3256±1.0355 | 0.0372±0.0299 | 0.2720±0.2121 |
| | PDUV[VoS]_F_(3,1) | 0.1523±0.1090 | 0.7958±0.7527 | 0.0826±0.0782 | 0.3130±0.3839 |
| | PDUV[VoS]_T_(3,1) | 0.1523±0.1243 | 0.4050±0.4486 | 0.0491±0.0476 | 0.2744±0.3014 |
| ScOT | PDUV_F_(3,1) | 0.0654±0.0360 | 0.3114±0.3727 | 0.0360±0.0209 | 0.2175±0.2388 |
| | PDUV_T_(3,1) | 0.1201±0.0702 | 0.6070±0.4590 | 0.0654±0.0505 | 0.3630±0.3077 |
| | PDUV[VoS]_F_(3,1) | 0.1470±0.1291 | 0.6403±0.4944 | 0.1550±0.1016 | 0.3504±0.3751 |
| | PDUV[VoS]_T_(3,1) | 0.1135±0.0753 | 0.5778±0.4467 | 0.1833±0.1173 | 0.4034±0.3748 |
| CNO | PDUV_F_(3,1) | 0.1067±0.0623 | 0.5085±0.6775 | 0.0152±0.0097 | 0.2450±0.2505 |
| | PDUV_T_(3,1) | 0.2181±0.1974 | 1.3132±0.9358 | 0.0485±0.0413 | 0.4230±0.3927 |
| | PDUV[VoS]_F_(3,1) | 0.0907±0.0534 | 0.2788±0.2190 | 0.0976±0.0675 | 0.4717±0.3578 |
| | PDUV[VoS]_T_(3,1) | 0.2610±0.1795 | 0.6138±0.5726 | 0.1189±0.1179 | 0.6371±0.6671 |
| ResNet | PDUV_F_(3,1) | 0.0387±0.0329 | 0.7315±0.6383 | | |
| | PDUV_T_(3,1) | 0.0688±0.0477 | 1.0318±0.9240 | | |
| | PDUV[VoS]_F_(3,1) | 0.1975±0.2441 | 0.6259±0.5556 | | |
| | PDUV[VoS]_T_(3,1) | 0.1777±0.1831 | 0.8637±0.6592 | | |
| Transolver | PDUV_F_(3,1) | 2.9121±1.6738 | 2.3465±0.9107 | | |
| | PDUV_T_(3,1) | 3.9408±2.1468 | 2.5241±0.9979 | | |
| | PDUV[VoS]_F_(3,1) | 3.7914±2.0321 | 2.4126±0.8926 | | |
| | PDUV[VoS]_T_(3,1) | 3.4930±1.9160 | 2.3327±0.8712 | | |

Table 40: Effect of conditioning fields for SIDA In-Distribution (ID) and Out-of-Distribution (OOD) datasets. Error-type 2 over output channels from section 4.5 is presented.

| MODEL | TAG | 1M | | 50M | |
|---|---|---|---|---|---|
| | | ID | OOD | ID | OOD |
| UNet | PDUV_F_(1,1) | 3.5661±2.4696 | 5.9914±3.2792 | 3.4983±2.5270 | 5.9438±3.2522 |
| | PDUV[VoS]_F_(1,1) | 3.9165±2.5452 | 6.1552±3.3755 | 3.8512±2.5096 | 6.0840±3.3590 |
| | PDUV_F_(3,1) | 0.3384±0.2089 | 2.2998±1.7754 | 0.1356±0.0976 | 1.9424±1.5948 |
| | PDUV[VoS]_F_(3,1) | 1.4203±1.1512 | 2.6100±1.8673 | 1.2162±1.0310 | 2.4740±1.8269 |
| | PDUV_T_(3,1) | 0.8690±0.4994 | 2.5683±1.8388 | 0.2272±0.1439 | 2.2561±1.7777 |
| | PDUV[VoS]_T_(3,1) | 1.6310±1.2358 | 2.7767±1.8269 | 1.1111±0.9569 | 2.4399±1.7482 |
| | PDUV_F_(3,2) | 0.4406±0.2437 | 2.1979±1.6703 | 0.1785±0.1080 | 1.8812±1.4977 |
| | PDUV[VoS]_F_(3,2) | 1.1622±0.8714 | 2.5568±1.8763 | 0.9378±0.8545 | 2.1892±1.6460 |
| FNO | PDUV_F_(1,1) | 3.5224±2.5529 | 6.0820±3.2896 | 3.4720±2.5577 | 5.9768±3.2572 |
| | PDUV[VoS]_F_(1,1) | 4.2901±3.0191 | 6.5170±3.5913 | 4.1024±2.7978 | 6.4133±3.4746 |
| | PDUV_F_(3,1) | 0.5333±0.2829 | 2.1215±1.5319 | 0.2498±0.1387 | 2.0963±1.5242 |
| | PDUV[VoS]_F_(3,1) | 1.8914±1.3113 | 2.6203±1.7532 | 1.7002±1.3234 | 2.6674±1.8160 |
| | PDUV_T_(3,1) | 0.7013±0.3424 | 1.8361±1.2131 | 0.4075±0.2168 | 1.7610±1.2616 |
| | PDUV[VoS]_T_(3,1) | 1.8383±1.2061 | 2.6599±1.7452 | 1.6667±1.0564 | 2.4769±1.6513 |
| | PDUV_F_(3,2) | 0.7123±0.3624 | 2.1449±1.5747 | 0.3389±0.1827 | 2.0990±1.5458 |
| | PDUV[VoS]_F_(3,2) | 1.8445±1.1729 | 2.5731±1.6426 | 1.5193±1.1838 | 2.4660±1.7004 |
| ViT | PDUV_F_(1,1) | 4.4758±2.8750 | 5.8219±3.1222 | 3.5527±2.7160 | 5.7645±3.1503 |
| | PDUV[VoS]_F_(1,1) | 4.6769±3.4301 | 5.6707±3.0938 | 3.9610±2.6804 | 5.8393±3.2097 |
| | PDUV_F_(3,1) | 1.5891±0.9548 | 3.4253±2.4145 | 0.4430±0.2336 | 1.8428±1.3876 |
| | PDUV[VoS]_F_(3,1) | 2.2152±1.5536 | 3.2082±2.2406 | 1.3497±1.0659 | 2.3674±1.7135 |
| | PDUV_T_(3,1) | 1.5949±1.0077 | 2.6376±1.7133 | 0.4249±0.2343 | 1.7359±1.3060 |
| | PDUV[VoS]_T_(3,1) | 2.0499±1.4162 | 3.0158±1.9680 | 1.2139±0.9433 | 2.0682±1.4535 |
| | PDUV_F_(3,2) | 1.3832±0.7495 | 2.4443±1.6647 | 0.5441±0.2699 | 1.8678±1.3767 |
| | PDUV[VoS]_F_(3,2) | 2.0047±1.2803 | 2.8629±1.8355 | 1.1189±0.8454 | 2.2224±1.5791 |
| ScOT | PDUV_F_(1,1) | 3.5764±2.5158 | 5.8985±3.2421 | 3.5173±2.6462 | 5.8306±3.1826 |
| | PDUV[VoS]_F_(1,1) | 4.7386±3.4158 | 5.7499±3.0977 | 2.0236±1.5584 | 4.3678±2.6511 |
| | PDUV_F_(3,1) | 0.6884±0.4222 | 1.9005±1.3770 | 0.2059±0.1199 | 1.6318±1.3494 |
| | PDUV[VoS]_F_(3,1) | 1.5944±1.2864 | 2.4643±1.7766 | 1.2171±1.0416 | 2.1314±1.5666 |
| | PDUV_T_(3,1) | 0.8203±0.4715 | 1.9132±1.2731 | 0.2461±0.1397 | 1.5425±1.1975 |
| | PDUV[VoS]_T_(3,1) | 1.4401±1.0497 | 2.2995±1.5283 | 1.1764±1.0238 | 1.9241±1.3649 |
| | PDUV_F_(3,2) | 0.7995±0.4265 | 1.9084±1.3867 | 0.2639±0.1459 | 1.5164±1.2414 |
| | PDUV[VoS]_F_(3,2) | 1.3294±0.9609 | 2.1893±1.5154 | 1.0655±0.9041 | 1.9709±1.4616 |
| CNO | PDUV_F_(1,1) | 3.9955±2.6786 | 5.8508±3.1777 | 3.5025±2.5455 | 5.9444±3.2344 |
| | PDUV[VoS]_F_(1,1) | 4.3438±2.7533 | 5.9778±3.2569 | 3.8389±2.5887 | 6.1782±3.4032 |
| | PDUV_F_(3,1) | 0.6943±0.4235 | 2.1707±1.8099 | 0.1621±0.1032 | 1.8159±1.4794 |
| | PDUV[VoS]_F_(3,1) | 1.1775±0.8980 | 2.1291±1.4489 | 1.2240±0.8859 | 2.3550±1.6215 |
| | PDUV_T_(3,1) | 0.9362±0.5384 | 66.43±453.50 | 0.3042±0.1905 | 2.3211±1.8495 |
| | PDUV[VoS]_T_(3,1) | 1.3235±0.9569 | 7898e1±6549e2 | 0.9654±0.7867 | 2.6471±1.9216 |
| | PDUV_F_(3,2) | 0.8584±0.4459 | 1.8360±1.3758 | 0.2210±0.1248 | 1.7634±1.3875 |
| | PDUV[VoS]_F_(3,2) | 1.2663±0.8642 | 1.9694±1.2874 | 1.0921±0.7494 | 2.3248±1.6283 |
| ResNet | PDUV_F_(1,1) | 3.7260±2.4574 | 6.0844±3.3406 | | |
| | PDUV[VoS]_F_(1,1) | 4.0975±2.5778 | 5.9925±3.3210 | | |
| | PDUV_F_(3,1) | 0.5570±0.3461 | 2.3042±1.7139 | | |
| | PDUV[VoS]_F_(3,1) | 1.3341±1.0333 | 2.6392±1.8649 | | |
| | PDUV_T_(3,1) | 1.0385±0.6198 | 3.0052±2.1978 | | |
| | PDUV[VoS]_T_(3,1) | 1.6636±1.2058 | 3.2900±2.3078 | | |
| | PDUV_F_(3,2) | 0.6356±0.3669 | 2.3012±1.6856 | | |
| | PDUV[VoS]_F_(3,2) | 1.2118±0.9203 | 2.4939±1.7801 | | |
| Transolver | PDUV_F_(1,1) | 5.7995±3.4502 | 6.9499±3.7154 | | |
| | PDUV[VoS]_F_(1,1) | 5.6838±3.1902 | 6.6967±3.4709 | | |
| | PDUV_F_(3,1) | 4.7065±2.6888 | 4.8852±2.7289 | | |
| | PDUV[VoS]_F_(3,1) | 4.9803±2.8336 | 5.2035±2.9156 | | |
| | PDUV_T_(3,1) | 4.7696±2.5778 | 4.6372±2.5046 | | |
| | PDUV[VoS]_T_(3,1) | 4.7139±2.5053 | 4.6158±2.4665 | | |
| | PDUV_F_(3,2) | 5.2026±2.7770 | 5.1580±2.7698 | | |
| | PDUV[VoS]_F_(3,2) | 5.0919±2.7361 | 5.1995±2.8432 | | |

Table 41: Effect of conditioning fields for SIDA In-Distribution (ID) and Out-of-Distribution (OOD) datasets. Error-type 2 over KE from section 4.5 is presented.

| MODEL | TAG | 1M | | 50M | |
|---|---|---|---|---|---|
| | | ID | OOD | ID | OOD |
| UNet | PDUV_F_(1,1) | 11.2017±29.5656 | 5.6983±3.0230 | 11.7372±31.0932 | 5.6613±2.9967 |
| | PDUV[VoS]_F_(1,1) | 13.3619±35.6175 | 5.5863±2.9830 | 12.0621±32.6814 | 5.6066±2.9599 |
| | PDUV_F_(3,1) | 0.5052±0.2615 | 2.5368±1.0710 | 0.1532±0.0897 | 1.4758±0.8257 |
| | PDUV[VoS]_F_(3,1) | 1.4473±1.0509 | 2.3929±1.6705 | 1.0164±0.8202 | 1.5336±0.9003 |
| | PDUV_T_(3,1) | 1.3388±0.8539 | 9.7208±7.6245 | 0.7025±0.5759 | 11.4778±8.3256 |
| | PDUV[VoS]_T_(3,1) | 1.6124±1.4219 | 13.7084±11.9033 | 3.1509±2.2207 | 19.8637±19.5043 |
| | PDUV_F_(3,2) | 0.9620±0.6433 | 9.6801±5.2064 | 0.4389±0.2677 | 2.2230±1.1932 |
| | PDUV[VoS]_F_(3,2) | 2.4488±1.9719 | 18.6904±11.8640 | 0.8829±0.7575 | 3.2323±1.6620 |
| FNO | PDUV_F_(1,1) | 10.8110±27.7074 | 5.7480±3.0231 | 10.7965±28.7023 | 5.6957±2.9859 |
| | PDUV[VoS]_F_(1,1) | 11.5359±30.3539 | 5.6379±3.0019 | 9.4723±24.4498 | 5.7736±3.0065 |
| | PDUV_F_(3,1) | 0.7682±0.5172 | 1.9919±1.3654 | 0.2823±0.1765 | 1.5754±1.9135 |
| | PDUV[VoS]_F_(3,1) | 8.5964±8.0247 | 7.1201±3.8520 | 4.9123±4.6565 | 14.2872±7.8378 |
| | PDUV_T_(3,1) | 1.1989±1.1997 | 12.2212±8.6412 | 0.9527±1.0156 | 12.7779±10.6585 |
| | PDUV[VoS]_T_(3,1) | 8.0715±11.6310 | 8.5574±5.6989 | 13.0682±10.7428 | 9.9693±6.6357 |
| | PDUV_F_(3,2) | 1.2933±0.6185 | 1.5242±0.8227 | 0.5000±0.3997 | 2.7004±1.5806 |
| | PDUV[VoS]_F_(3,2) | 9.4691±11.8198 | 5.7364±3.8886 | 2.9789±2.6236 | 9.1980±5.1366 |
| ViT | PDUV_F_(1,1) | 31.9704±87.7762 | 4.0911±2.0809 | 15.7706±44.1921 | 5.3135±2.7717 |
| | PDUV[VoS]_F_(1,1) | 45.4819±122.3350 | 3.2056±1.7203 | 19.0189±54.0257 | 4.9858±2.5512 |
| | PDUV_F_(3,1) | 6.6527±8.3450 | 32.5122±28.4652 | 0.9694±0.6501 | 3.3619±1.4849 |
| | PDUV[VoS]_F_(3,1) | 5.5548±5.2277 | 20.9420±21.4748 | 0.9345±0.7133 | 2.2616±0.7760 |
| | PDUV_T_(3,1) | 5.3717±4.3670 | 10.2786±6.7211 | 1.0290±0.6755 | 5.2700±3.4986 |
| | PDUV[VoS]_T_(3,1) | 7.3620±6.8146 | 6.6430±5.4757 | 1.6828±1.2214 | 6.3459±5.0306 |
| | PDUV_F_(3,2) | 5.3889±3.2322 | 5.4872±2.9669 | 1.9649±1.1585 | 2.7149±1.9905 |
| | PDUV[VoS]_F_(3,2) | 8.3093±6.0799 | 19.7742±12.6920 | 2.3528±1.3365 | 5.9363±2.9190 |
| ScOT | PDUV_F_(1,1) | 16.2807±44.8844 | 5.3495±2.8561 | 15.2877±42.6846 | 5.3579±2.8067 |
| | PDUV[VoS]_F_(1,1) | 32.4834±90.6835 | 3.9399±1.8714 | 6.1295±7.5593 | 1.8328±1.4427 |
| | PDUV_F_(3,1) | 2.6928±1.9141 | 2.5941±1.1618 | 1.0306±0.6112 | 2.1724±0.9115 |
| | PDUV[VoS]_F_(3,1) | 2.7987±2.8603 | 8.5151±4.4987 | 8.1992±5.7868 | 6.2404±3.3768 |
| | PDUV_T_(3,1) | 3.9887±3.2842 | 11.9888±7.7975 | 1.8979±1.8130 | 5.5654±4.9727 |
| | PDUV[VoS]_T_(3,1) | 6.5291±5.5513 | 5.6833±4.0009 | 9.0977±6.0278 | 22.1760±19.9296 |
| | PDUV_F_(3,2) | 4.2225±2.2977 | 9.6165±5.0281 | 1.7483±0.9528 | 5.3631±2.7718 |
| | PDUV[VoS]_F_(3,2) | 3.7081±2.5449 | 6.3008±2.9780 | 2.3370±1.6511 | 4.1953±1.7958 |
| CNO | PDUV_F_(1,1) | 17.6490±49.1966 | 5.0935±2.5799 | 12.7633±35.2539 | 5.5010±2.8628 |
| | PDUV[VoS]_F_(1,1) | 34.8635±91.4468 | 4.3743±2.4571 | 11.2465±26.6401 | 5.8412±3.1259 |
| | PDUV_F_(3,1) | 3.9209±2.3251 | 10.1072±9.3279 | 0.4011±0.2343 | 1.3602±0.6405 |
| | PDUV[VoS]_F_(3,1) | 4.3996±2.7430 | 24.1448±21.7419 | 6.0637±3.7637 | 6.9987±3.5410 |
| | PDUV_T_(3,1) | 5.8777±6.1544 | 2428e5±2916e5 | 1.2505±0.9845 | 2.4099±2.0305 |
| | PDUV[VoS]_T_(3,1) | 6.5098±6.0036 | – | 2.1665±2.1864 | 10.1886±11.2716 |
| | PDUV_F_(3,2) | 4.9192±3.6170 | 20.5139±13.7615 | 0.8113±0.5196 | 1.3482±0.7874 |
| | PDUV[VoS]_F_(3,2) | 8.8186±4.5825 | 15.8403±6.7297 | 7.7326±4.5056 | 7.9407±4.5767 |
| ResNet | PDUV_F_(1,1) | 10.9262±28.4670 | 5.7403±3.0356 | | |
| | PDUV[VoS]_F_(1,1) | 27.4023±76.4492 | 4.4619±2.2558 | | |
| | PDUV_F_(3,1) | 0.8521±0.5177 | 1.9815±0.8412 | | |
| | PDUV[VoS]_F_(3,1) | 1.7642±1.3131 | 2.4584±1.8013 | | |
| | PDUV_T_(3,1) | 1.5262±1.4191 | 28.0582±21.1096 | | |
| | PDUV[VoS]_T_(3,1) | 2.7497±2.3888 | 34.0870±30.7161 | | |
| | PDUV_F_(3,2) | 1.4069±0.6998 | 5.6830±3.2346 | | |
| | PDUV[VoS]_F_(3,2) | 3.2122±2.2595 | 9.1524±3.6021 | | |
| Transolver | PDUV_F_(1,1) | 21.6155±58.9766 | 5.0188±2.6784 | | |
| | PDUV[VoS]_F_(1,1) | 69.0841±195.2310 | 4.1611±2.9672 | | |
| | PDUV_F_(3,1) | 67.9430±41.9429 | 59.5887±20.7706 | | |
| | PDUV[VoS]_F_(3,1) | 83.0077±119.1757 | 35.9022±11.1758 | | |
| | PDUV_T_(3,1) | 93.0350±110.8223 | 258.0369±171.4161 | | |
| | PDUV[VoS]_T_(3,1) | 102.3762±95.8241 | 162.3676±90.5726 | | |
| | PDUV_F_(3,2) | 181.1406±151.2849 | 96.0656±48.8018 | | |
| | PDUV[VoS]_F_(3,2) | 113.5771±105.2289 | 77.0457±28.0248 | | |

Table 42: Effect of conditioning fields for SIDA In-Distribution (ID) and Out-of-Distribution (OOD) datasets. Error-type 2 over VP from section 4.5 is presented.

| MODEL | TAG | 1M | | 50M | |
|---|---|---|---|---|---|
| | | ID | OOD | ID | OOD |
| UNet | PDUV_F_(1,1) | 4.8548±9.7684 | 5.1872±2.8746 | 5.1272±10.2187 | 5.1860±2.8710 |
| | PDUV[VoS]_F_(1,1) | 4.7080±9.0211 | 5.3153±2.9026 | 4.6359±5.0888 | 5.5856±3.1400 |
| | PDUV_F_(3,1) | 0.4678±0.5580 | 1.9011±1.4640 | 0.1094±0.1362 | 2.0724±1.6647 |
| | PDUV[VoS]_F_(3,1) | 1.2571±1.2591 | 1.2459±1.0752 | 0.5621±0.4353 | 2.3172±1.6960 |
| | PDUV_T_(3,1) | 1.0717±1.8190 | 2.0873±2.1121 | 0.2973±0.2529 | 2.6720±2.2034 |
| | PDUV[VoS]_T_(3,1) | 1.7996±1.7697 | 1.7940±1.4594 | 0.7514±1.5558 | 2.4527±2.0558 |
| | PDUV_F_(3,2) | 0.7461±1.0477 | 2.4576±1.7732 | 0.3118±0.3532 | 2.1101±1.7997 |
| | PDUV[VoS]_F_(3,2) | 2.0485±2.9920 | 1.5854±1.0366 | 0.6229±0.4991 | 2.2803±1.6937 |
| FNO | PDUV_F_(1,1) | 4.6886±8.4082 | 5.4445±2.9026 | 4.7255±9.1404 | 5.3154±2.8825 |
| | PDUV[VoS]_F_(1,1) | 4.7356±3.4140 | 5.9386±3.2272 | 5.5056±12.1875 | 5.2449±2.7262 |
| | PDUV_F_(3,1) | 0.9580±0.6336 | 2.1666±1.2573 | 0.4874±0.4576 | 1.7614±1.5075 |
| | PDUV[VoS]_F_(3,1) | 4.3073±8.5326 | 1.8055±1.3533 | 3.0148±4.1413 | 1.2411±0.9617 |
| | PDUV_T_(3,1) | 1.3719±0.7673 | 2.7430±1.5973 | 0.6992±0.5029 | 1.5961±1.7109 |
| | PDUV[VoS]_T_(3,1) | 1.4985±1.5211 | 3.7245±2.1666 | 1.4679±1.1389 | 3.9079±2.2991 |
| | PDUV_F_(3,2) | 1.1520±0.7034 | 2.7981±1.6515 | 0.6015±0.4159 | 1.3866±0.9442 |
| | PDUV[VoS]_F_(3,2) | 6.4856±9.7679 | 2.6203±1.6912 | 1.9468±2.0439 | 0.9703±0.8280 |
| ViT | PDUV_F_(1,1) | 28.3750±59.3870 | 2.6103±1.1959 | 5.4192±11.6520 | 5.0160±2.7830 |
| | PDUV[VoS]_F_(1,1) | 78.3286±188.0569 | 6.6896±6.4194 | 10.8812±26.3883 | 4.0604±2.1094 |
| | PDUV_F_(3,1) | 13.1278±16.1199 | 4.9536±5.5894 | 0.5882±0.9129 | 1.4598±2.5938 |
| | PDUV[VoS]_F_(3,1) | 17.3733±19.1933 | 5.2368±4.1392 | 5.3010±4.7988 | 2.2461±2.3365 |
| | PDUV_T_(3,1) | 10.3206±10.3850 | 11.4467±13.3514 | 0.5533±0.6822 | 3.4015±5.2220 |
| | PDUV[VoS]_T_(3,1) | 17.9269±18.2700 | 7.3386±7.3888 | 3.6743±3.3607 | 4.4309±5.3252 |
| | PDUV_F_(3,2) | 4.1804±4.5308 | 2.1003±1.7241 | 0.8756±1.1214 | 1.7605±2.6311 |
| | PDUV[VoS]_F_(3,2) | 10.2143±10.9571 | 3.9340±3.2964 | 2.1356±2.0161 | 1.3421±1.2843 |
| ScOT | PDUV_F_(1,1) | 5.0246±10.1023 | 5.1421±2.8596 | 5.4551±12.0375 | 4.9934±2.7667 |
| | PDUV[VoS]_F_(1,1) | 9.3970±23.3038 | 4.3577±2.2503 | 1.6456±3.2854 | 3.6278±2.1986 |
| | PDUV_F_(3,1) | 1.6140±2.9631 | 1.4442±0.6758 | 0.4774±0.8474 | 1.1108±0.9826 |
| | PDUV[VoS]_F_(3,1) | 2.3244±3.7987 | 1.2487±0.8233 | 2.5129±2.5532 | 0.7567±0.4354 |
| | PDUV_T_(3,1) | 1.0985±1.9509 | 1.0331±0.7909 | 0.8093±1.4573 | 0.6340±0.6392 |
| | PDUV[VoS]_T_(3,1) | 1.5664±2.8871 | 0.7565±0.4219 | 1.7867±3.6712 | 2.1391±2.2486 |
| | PDUV_F_(3,2) | 1.3736±2.0323 | 0.7779±0.5431 | 0.8046±1.0890 | 0.7494±0.6484 |
| | PDUV[VoS]_F_(3,2) | 1.1559±1.8917 | 1.0690±0.8675 | 2.3292±3.2330 | 1.0448±0.4318 |
| CNO | PDUV_F_(1,1) | 5.0839±10.2831 | 5.0351±2.8432 | 5.0165±10.3994 | 5.0809±2.8365 |
| | PDUV[VoS]_F_(1,1) | 4.8944±10.2847 | 5.1612±2.8534 | 4.4769±6.0270 | 5.4406±3.0542 |
| | PDUV_F_(3,1) | 1.9935±6.3802 | 0.9771±0.7020 | 0.2560±0.2127 | 2.0421±1.5173 |
| | PDUV[VoS]_F_(3,1) | 2.5010±3.3056 | 2.3219±1.6920 | 2.2471±5.3269 | 1.5615±1.2442 |
| | PDUV_T_(3,1) | 3.3135±8.2291 | 8684e2±9765e3 | 0.4813±0.5887 | 1.7157±1.4923 |
| | PDUV[VoS]_T_(3,1) | 2.3146±5.0735 | 1396e9±1455e10 | 0.9616±2.4379 | 2.9383±2.3800 |
| | PDUV_F_(3,2) | 3.4611±6.6855 | 2.8181±1.6026 | 0.5898±0.4428 | 2.1937±1.6745 |
| | PDUV[VoS]_F_(3,2) | 3.0088±6.6812 | 1.0182±0.8092 | 3.0996±5.7459 | 1.5902±1.2229 |
| ResNet | PDUV_F_(1,1) | 5.1266±10.6196 | 5.1755±2.8330 | | |
| | PDUV[VoS]_F_(1,1) | 6.0033±14.0618 | 4.9715±2.6616 | | |
| | PDUV_F_(3,1) | 0.5801±0.4023 | 1.6962±1.3323 | | |
| | PDUV[VoS]_F_(3,1) | 1.1306±0.5625 | 1.2508±1.0298 | | |
| | PDUV_T_(3,1) | 0.8478±0.6680 | 3.0296±2.2831 | | |
| | PDUV[VoS]_T_(3,1) | 2.4775±2.0845 | 2.1219±1.5490 | | |
| | PDUV_F_(3,2) | 0.6297±0.4655 | 2.2821±1.5949 | | |
| | PDUV[VoS]_F_(3,2) | 0.7945±0.6044 | 1.8356±1.2950 | | |
| Transolver | PDUV_F_(1,1) | 6.9632±15.3887 | 5.0256±2.5608 | | |
| | PDUV[VoS]_F_(1,1) | 9.8138±26.7480 | 4.8703±2.2493 | | |
| | PDUV_F_(3,1) | 42.3382±30.2457 | 23.1503±16.1705 | | |
| | PDUV[VoS]_F_(3,1) | 19.9930±27.6070 | 11.1969±3.1268 | | |
| | PDUV_T_(3,1) | 27.5997±22.3968 | 27.7379±18.3586 | | |
| | PDUV[VoS]_T_(3,1) | 27.7826±22.1833 | 16.0020±15.6628 | | |
| | PDUV_F_(3,2) | 11.3000±14.7913 | 1.8346±1.1888 | | |
| | PDUV[VoS]_F_(3,2) | 9.2376±12.2430 | 2.5793±1.7156 | | |

Table 43: Effect of conditioning fields for SIDA In-Distribution (ID) and Out-of-Distribution (OOD) datasets. Error-type 2 over COM from section 4.5 is presented.

| MODEL | TAG | 1M | | 50M | |
|---|---|---|---|---|---|
| | | ID | OOD | ID | OOD |
| UNet | PDUV_F_(1,1) | 0.2325±0.2905 | 0.9394±0.9670 | 0.2235±0.3098 | 0.9910±0.9754 |
| | PDUV[VoS]_F_(1,1) | 0.2669±0.3680 | 1.0355±1.0419 | 0.2547±0.3429 | 1.0036±1.0209 |
| | PDUV_F_(3,1) | 0.0228±0.0126 | 0.5918±0.4201 | 0.0088±0.0047 | 0.7818±0.7129 |
| | PDUV[VoS]_F_(3,1) | 0.2094±0.3003 | 0.4709±0.4614 | 0.1798±0.2596 | 0.6587±0.5563 |
| | PDUV_T_(3,1) | 0.0683±0.0405 | 1.8477±1.8043 | 0.0331±0.0224 | 0.5298±0.4377 |
| | PDUV[VoS]_T_(3,1) | 0.1570±0.1729 | 1.0825±0.8026 | 0.1380±0.1897 | 0.9711±0.7603 |
| | PDUV_F_(3,2) | 0.0566±0.0265 | 0.4628±0.4387 | 0.0251±0.0148 | 0.5248±0.4867 |
| | PDUV[VoS]_F_(3,2) | 0.1600±0.1434 | 1.0387±0.6696 | 0.1552±0.1965 | 0.7248±0.6708 |
| FNO | PDUV_F_(1,1) | 0.2377±0.3125 | 1.0179±0.9760 | 0.2188±0.2944 | 0.9605±0.9621 |
| | PDUV[VoS]_F_(1,1) | 0.2766±0.3603 | 1.0277±1.0291 | 0.2747±0.3434 | 0.9904±1.0263 |
| | PDUV_F_(3,1) | 0.0271±0.0130 | 0.2420±0.3222 | 0.0118±0.0078 | 0.2427±0.2805 |
| | PDUV[VoS]_F_(3,1) | 0.2230±0.1315 | 0.5460±0.4013 | 0.1534±0.1235 | 0.8256±0.5094 |
| | PDUV_T_(3,1) | 0.0486±0.0462 | 0.5155±0.4017 | 0.0386±0.0311 | 0.4851±0.4057 |
| | PDUV[VoS]_T_(3,1) | 0.2803±0.1964 | 0.6526±0.5474 | 0.4330±0.2868 | 0.8936±0.5447 |
| | PDUV_F_(3,2) | 0.0561±0.0319 | 0.2477±0.3338 | 0.0267±0.0141 | 0.2091±0.2934 |
| | PDUV[VoS]_F_(3,2) | 0.4183±0.1414 | 0.5852±0.4017 | 0.1546±0.1802 | 0.7524±0.5978 |
| ViT | PDUV_F_(1,1) | 0.3214±0.2879 | 0.8025±0.8487 | 0.2269±0.2774 | 0.9255±0.9327 |
| | PDUV[VoS]_F_(1,1) | 0.3038±0.2661 | 0.8288±0.8959 | 0.2590±0.3339 | 0.9899±1.0103 |
| | PDUV_F_(3,1) | 0.1844±0.1821 | 1.0956±0.8733 | 0.0326±0.0211 | 0.2025±0.2268 |
| | PDUV[VoS]_F_(3,1) | 0.1523±0.1090 | 0.7958±0.7527 | 0.0826±0.0782 | 0.3130±0.3839 |
| | PDUV_T_(3,1) | 0.2011±0.2260 | 1.3256±1.0355 | 0.0372±0.0299 | 0.2720±0.2121 |
| | PDUV[VoS]_T_(3,1) | 0.1523±0.1243 | 0.4050±0.4486 | 0.0491±0.0476 | 0.2744±0.3014 |
| | PDUV_F_(3,2) | 0.2007±0.1252 | 0.4172±0.3822 | 0.0751±0.0418 | 0.1544±0.1388 |
| | PDUV[VoS]_F_(3,2) | 0.1785±0.1410 | 0.9977±0.6339 | 0.0679±0.0424 | 0.3529±0.3592 |
| ScOT | PDUV_F_(1,1) | 0.2393±0.3107 | 0.9747±0.9872 | 0.2238±0.2732 | 0.9072±0.9265 |
| | PDUV[VoS]_F_(1,1) | 0.2696±0.2834 | 0.7985±0.8553 | 0.1273±0.1445 | 0.6410±0.7542 |
| | PDUV_F_(3,1) | 0.0654±0.0359 | 0.3114±0.3727 | 0.0360±0.0209 | 0.2175±0.2388 |
| | PDUV[VoS]_F_(3,1) | 0.1470±0.1291 | 0.6403±0.4944 | 0.1550±0.1016 | 0.3504±0.3751 |
| | PDUV_T_(3,1) | 0.1201±0.0702 | 0.6070±0.4590 | 0.0654±0.0505 | 0.3630±0.3077 |
| | PDUV[VoS]_T_(3,1) | 0.1135±0.0753 | 0.5778±0.4467 | 0.1833±0.1173 | 0.4034±0.3748 |
| | PDUV_F_(3,2) | 0.1993±0.0869 | 0.3583±0.2814 | 0.0623±0.0293 | 0.2439±0.3005 |
| | PDUV[VoS]_F_(3,2) | 0.1407±0.1018 | 0.4966±0.3738 | 0.1141±0.1122 | 0.4051±0.3803 |
| CNO | PDUV_F_(1,1) | 0.4844±0.6753 | 0.6387±0.6294 | 0.2202±0.2753 | 0.9190±0.9273 |
| | PDUV[VoS]_F_(1,1) | 0.3139±0.3389 | 0.7298±0.8015 | 0.2666±0.3817 | 1.0782±1.0407 |
| | PDUV_F_(3,1) | 0.1067±0.0623 | 0.5085±0.6775 | 0.0152±0.0097 | 0.2450±0.2505 |
| | PDUV[VoS]_F_(3,1) | 0.0907±0.0534 | 0.2788±0.2190 | 0.0976±0.0675 | 0.4717±0.3578 |
| | PDUV_T_(3,1) | 0.2181±0.1974 | 1.3132±0.9358 | 0.0485±0.0413 | 0.4230±0.3927 |
| | PDUV[VoS]_T_(3,1) | 0.2610±0.1795 | 0.6138±0.5726 | 0.1189±0.1179 | 0.6371±0.6671 |
| | PDUV_F_(3,2) | 0.1791±0.1004 | 0.3669±0.2622 | 0.0337±0.0189 | 0.2031±0.2339 |
| | PDUV[VoS]_F_(3,2) | 0.3016±0.1480 | 0.6484±0.3145 | 0.2154±0.0912 | 0.6700±0.4355 |
| ResNet | PDUV_F_(1,1) | 0.2284±0.3093 | 0.9806±0.9846 | | |
| | PDUV[VoS]_F_(1,1) | 0.2574±0.2885 | 0.9271±0.9708 | | |
| | PDUV_F_(3,1) | 0.0387±0.0329 | 0.7315±0.6383 | | |
| | PDUV[VoS]_F_(3,1) | 0.1975±0.2441 | 0.6259±0.5556 | | |
| | PDUV_T_(3,1) | 0.0688±0.0477 | 1.0318±0.9240 | | |
| | PDUV[VoS]_T_(3,1) | 0.1777±0.1831 | 0.8637±0.6592 | | |
| | PDUV_F_(3,2) | 0.0729±0.0535 | 0.4976±0.3697 | | |
| | PDUV[VoS]_F_(3,2) | 0.1063±0.0724 | 0.8453±0.5704 | | |
| Transolver | PDUV_F_(1,1) | 0.6920±0.4419 | 0.7175±0.7466 | | |
| | PDUV[VoS]_F_(1,1) | 1.4407±1.0233 | 0.7835±0.5373 | | |
| | PDUV_F_(3,1) | 2.9121±1.6738 | 2.3465±0.9107 | | |
| | PDUV[VoS]_F_(3,1) | 3.7914±2.0321 | 2.4126±0.8926 | | |
| | PDUV_T_(3,1) | 3.9408±2.1468 | 2.5241±0.9979 | | |
| | PDUV[VoS]_T_(3,1) | 3.4930±1.9160 | 2.3327±0.8712 | | |
| | PDUV_F_(3,2) | 4.0355±2.0120 | 2.7159±1.0297 | | |
| | PDUV[VoS]_F_(3,2) | 4.1446±2.1033 | 2.6625±1.0178 | | |

# D  ERROR ROLLOUT OVER TIMESTEPS

## D.1  ERROR ROLLOUT OVER TIMESTEPS FOR THE LIDE DATASET

The evolution of cumulative nRMSE is studied in a cumulative fashion over timesteps for 1M and 50M models on the test trajectories of both In-Distribution (ID) and Out-of-Distribution (OOD) LIDE datasets.

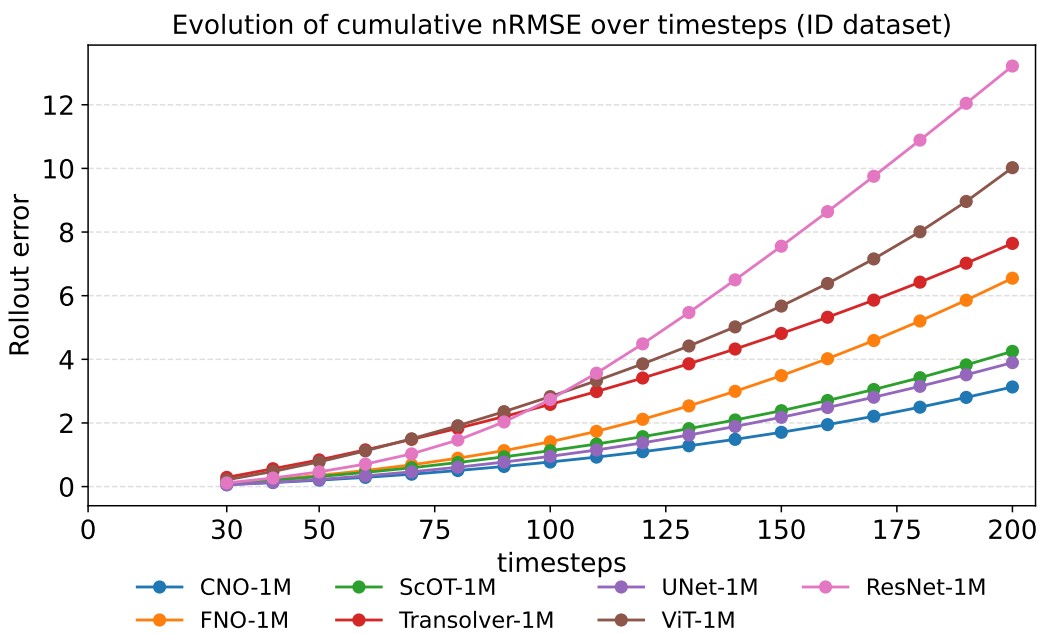

Figure 9: Temporal rollout of cumulative nRMSE over output channels for the LIDE-ID-Experiment PDUV_F_(3,1) for all 1M models.

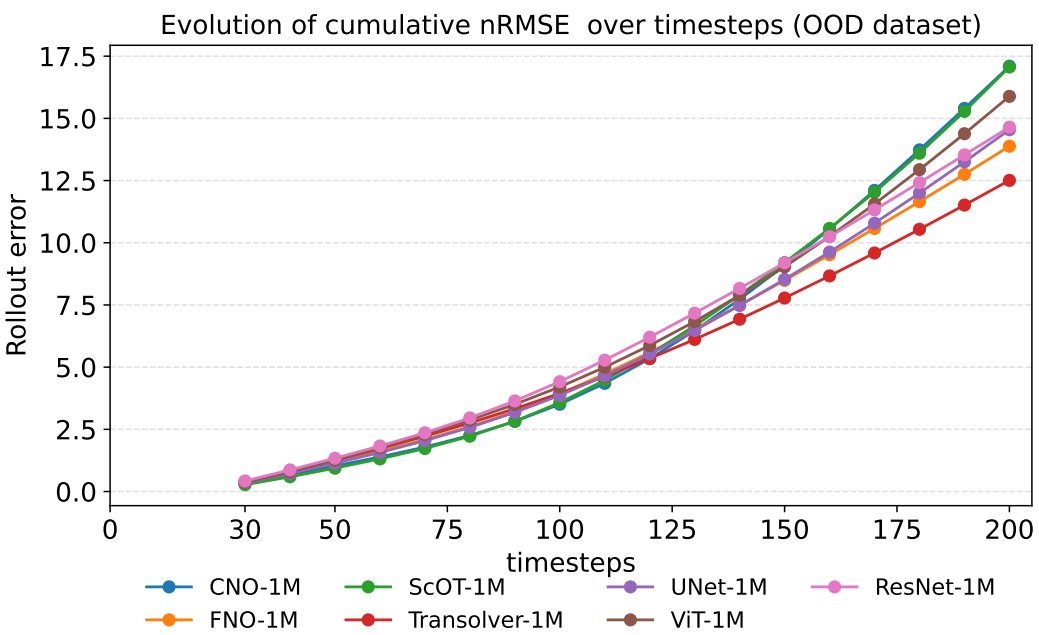

Figure 10: Temporal rollout of cumulative nRMSE over output channels for the LIDE-OOD-Experiment PDUV_F_(3,1) for all 1M models.

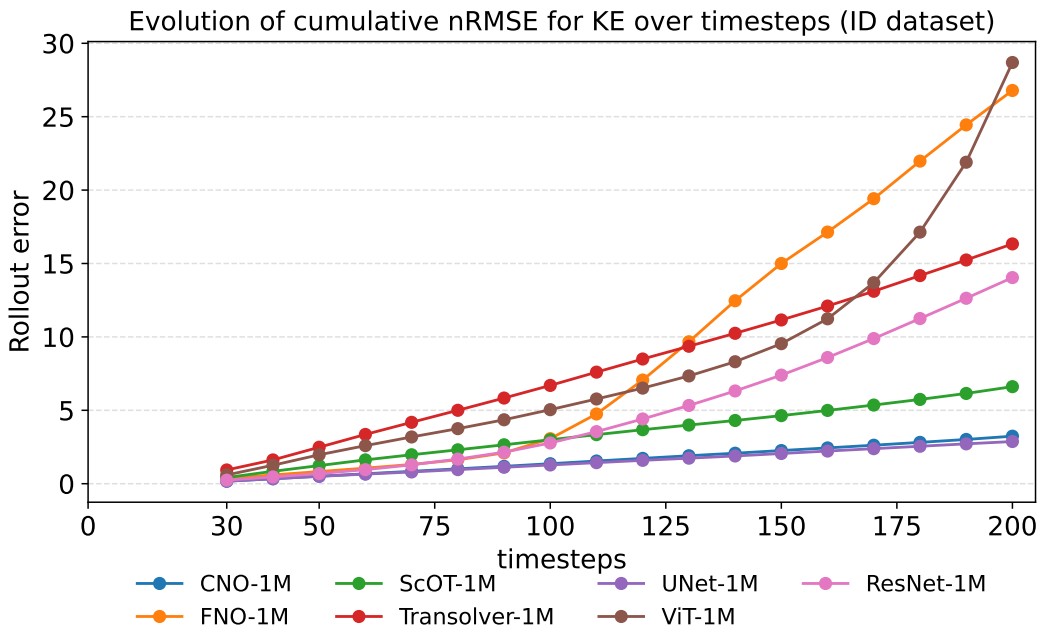

Figure 11: Temporal rollout of cumulative nRMSE over KE for the LIDE-ID-Experiment PDUV_F_(3,1) for all 1M models.

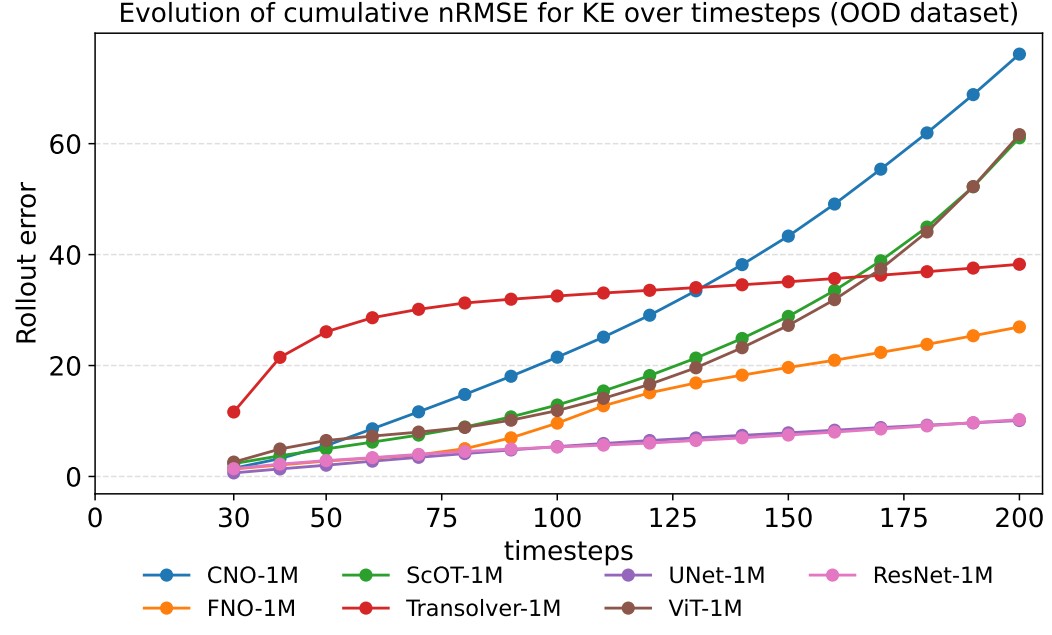

Figure 12: Temporal rollout of cumulative nRMSE over KE for the LIDE-OOD-Experiment PDUV_F_(3,1) for all 1M models.

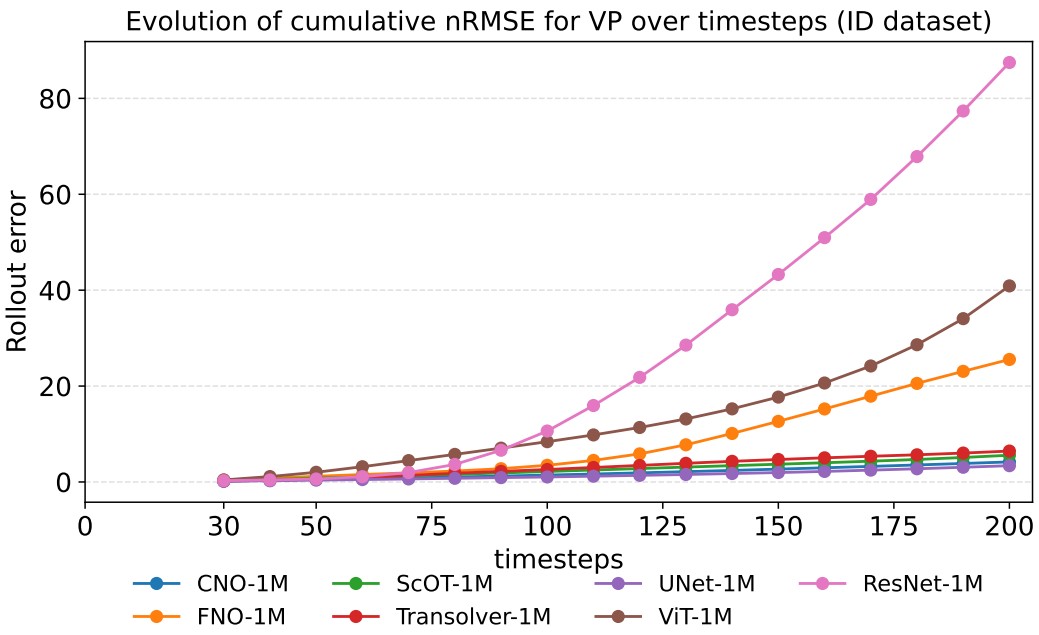

Figure 13: Temporal rollout of cumulative nRMSE over VP for the LIDE-ID-Experiment PDUV_F_(3,1) for all 1M models.

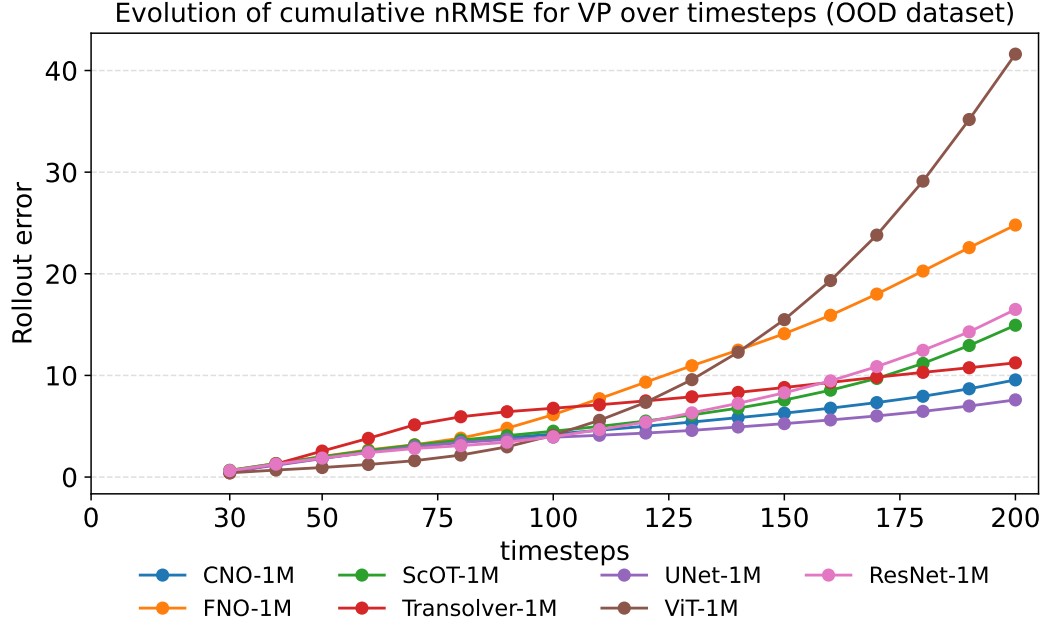

Figure 14: Temporal rollout of cumulative nRMSE over VP for the LIDE-OOD-Experiment PDUV_F_(3,1) for all 1M models.

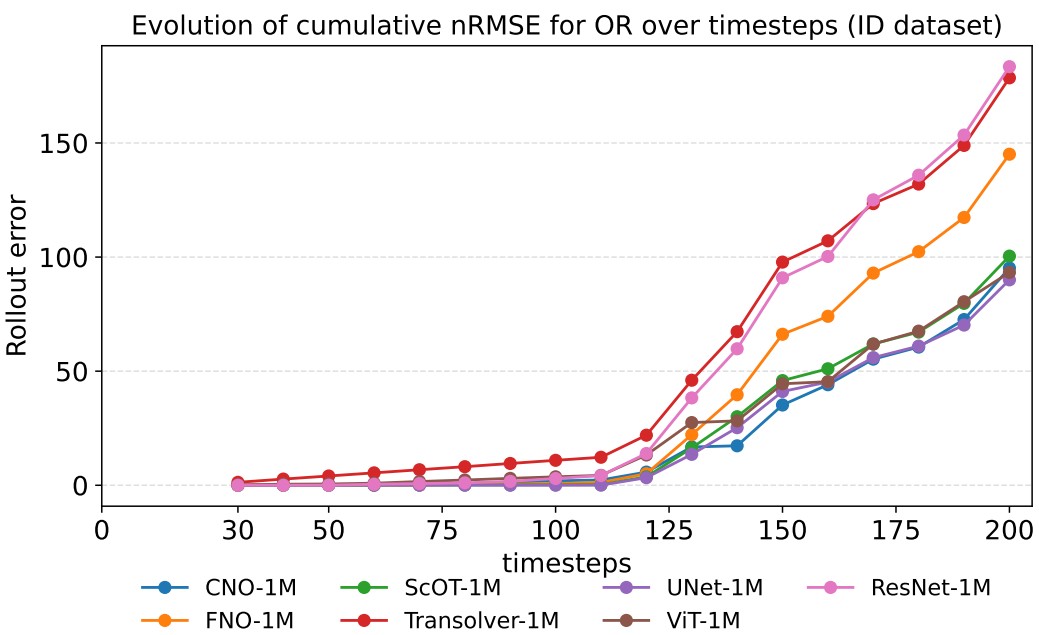

Figure 15: Temporal rollout of cumulative nRMSE over OR for the LIDE-ID-Experiment PDUV_F_(3,1) for all 1M models.

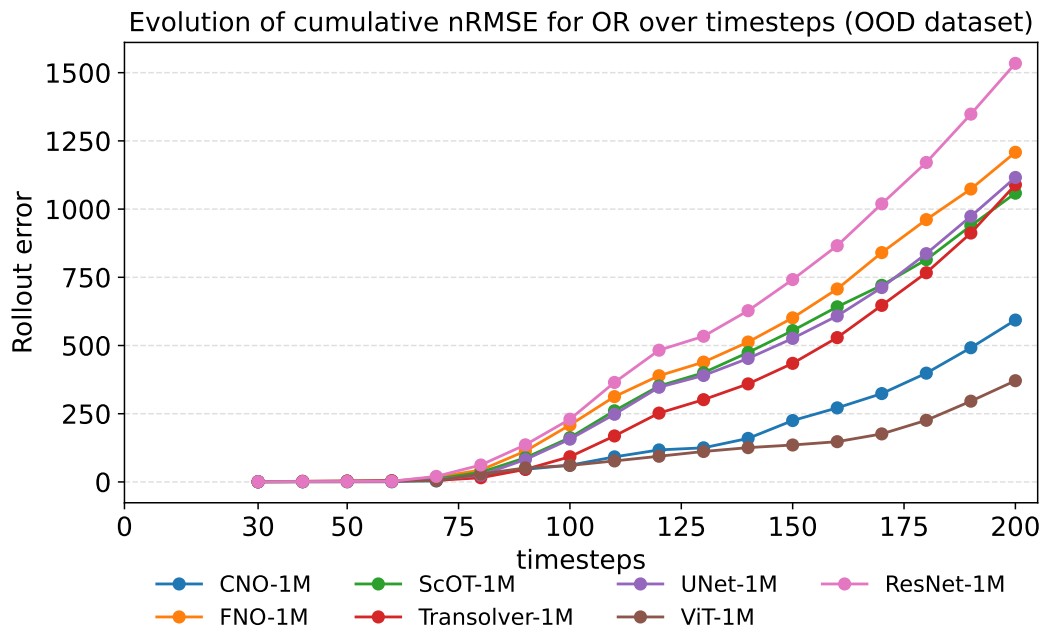

Figure 16: Temporal rollout of cumulative nRMSE over OR for the LIDE-OOD-Experiment PDUV_F_(3,1) for all 1M models.

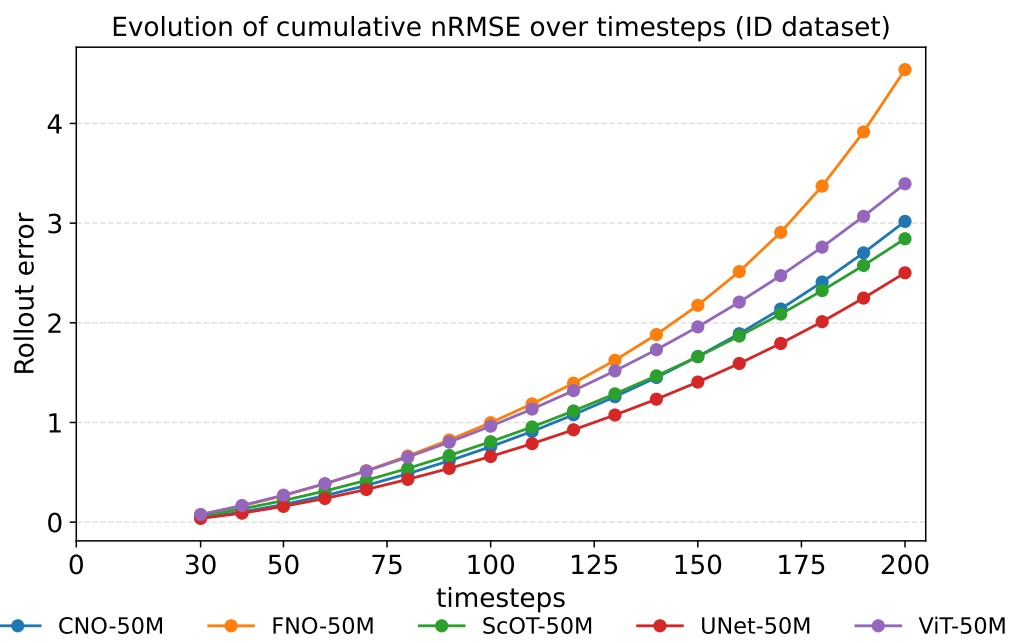

Figure 17: Temporal rollout of cumulative nRMSE over output channels for the LIDE-ID-Experiment PDUV_F_(3,1) for all 50M models.

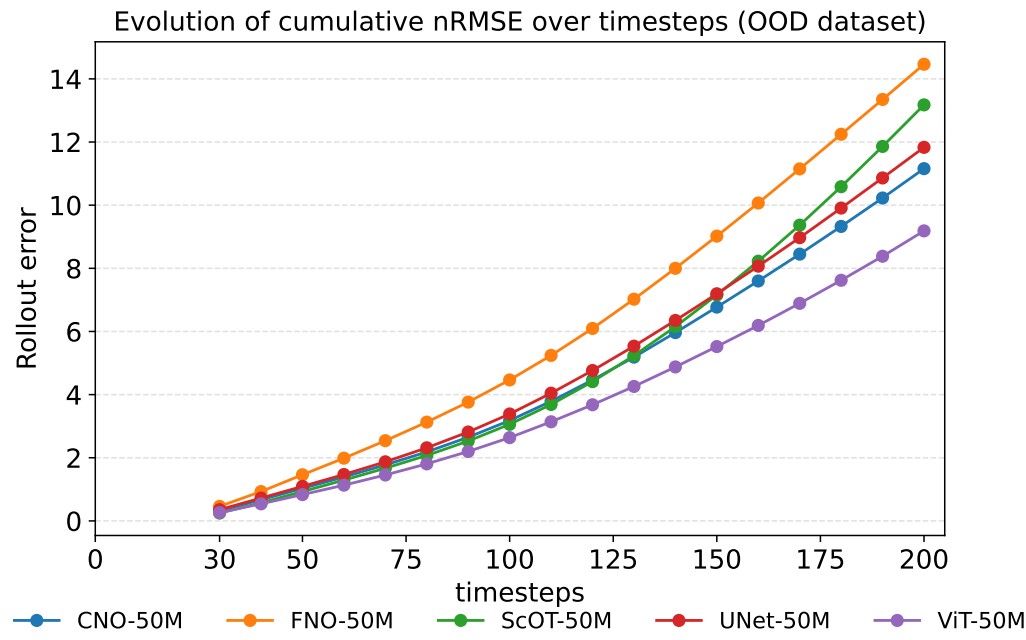

Figure 18: Temporal rollout of cumulative nRMSE over output channels for the LIDE-OOD-Experiment PDUV_F_(3,1) for all 50M models.

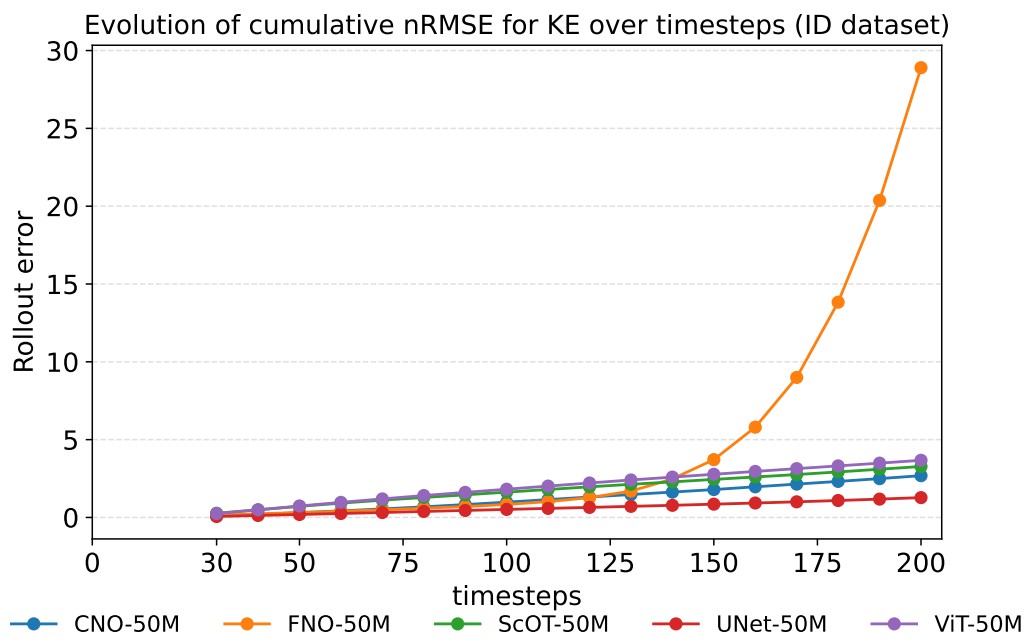

Figure 19: Temporal rollout of cumulative nRMSE over KE for the LIDE-ID-Experiment PDUV_F_(3,1) for all 50M models.

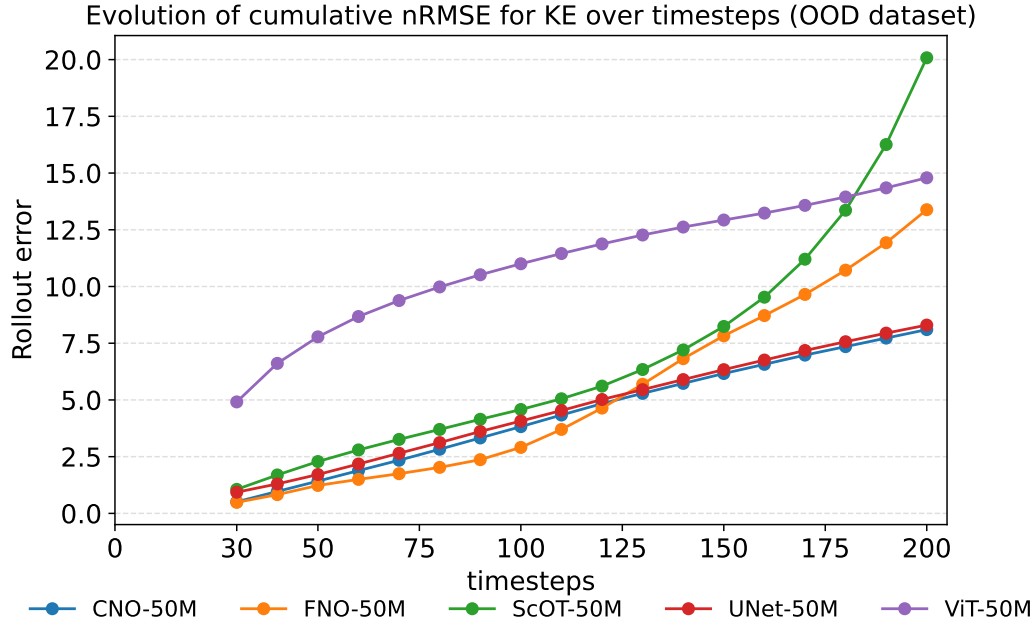

Figure 20: Temporal rollout of cumulative nRMSE over KE for the LIDE-OOD-Experiment PDUV_F_(3,1) for all 50M models.

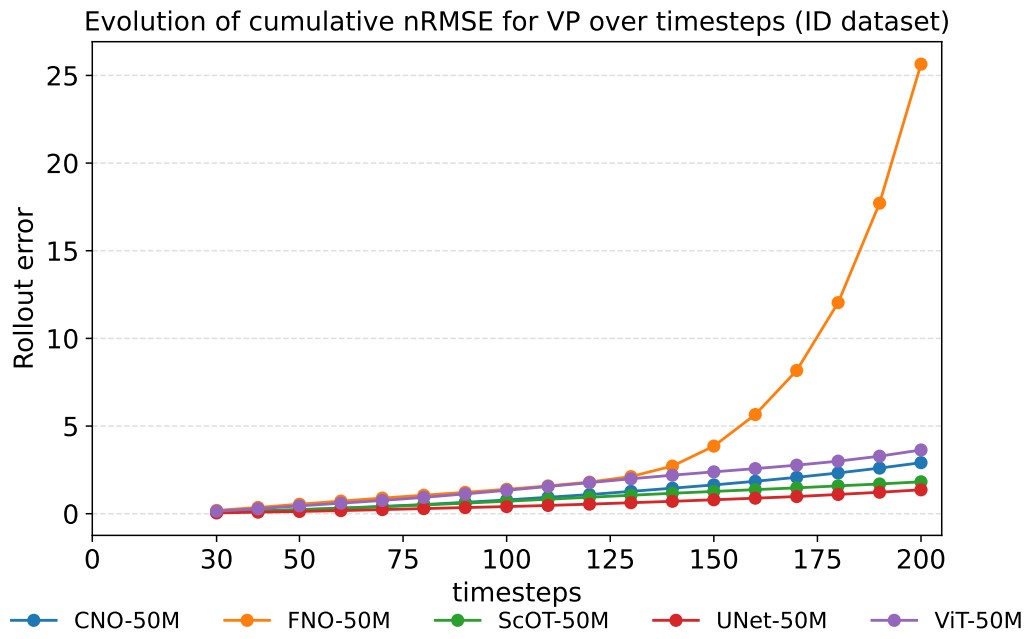

Figure 21: Temporal rollout of cumulative nRMSE over VP for the LIDE-ID-Experiment PDUV_F_(3,1) for all 50M models.

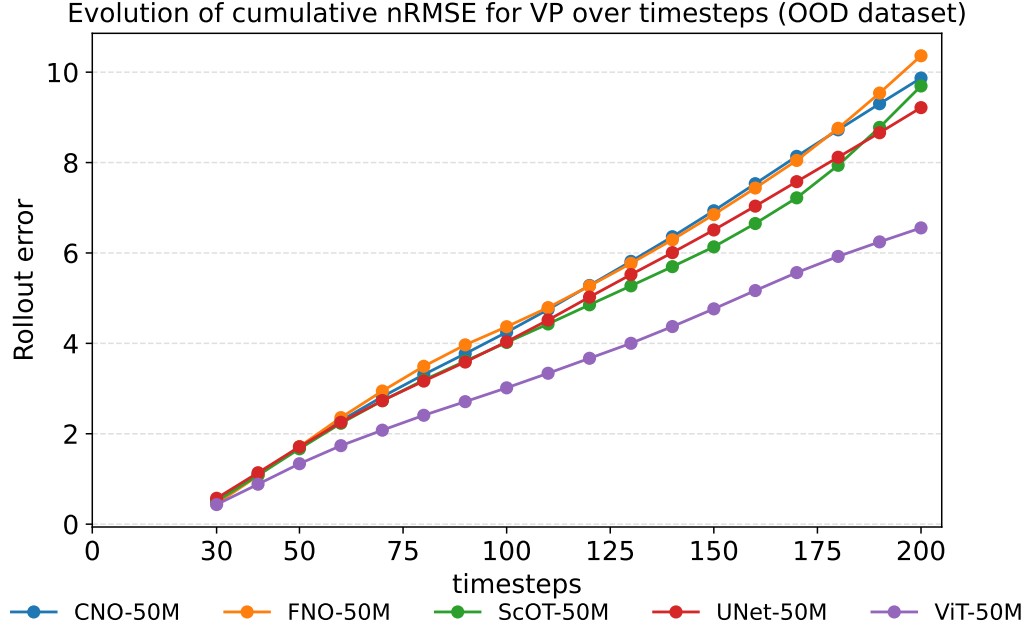

Figure 22: Temporal rollout of cumulative nRMSE over VP for the LIDE-OOD-Experiment PDUV_F_(3,1) for all 50M models.

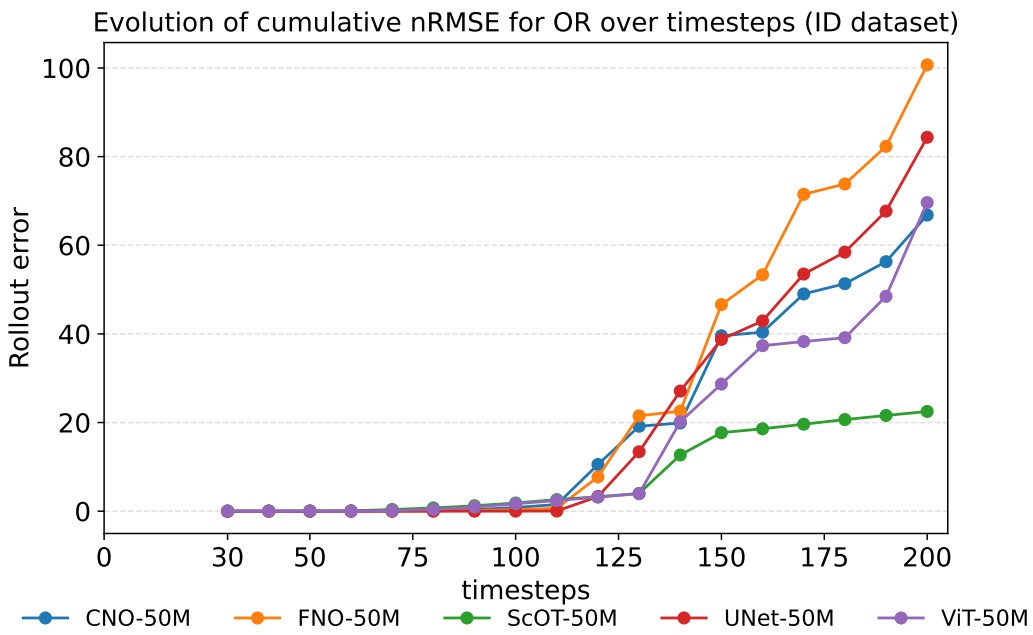

Figure 23: Temporal rollout of cumulative nRMSE over OR for the LIDE-ID-Experiment PDUV_F_(3,1) for all 50M models.

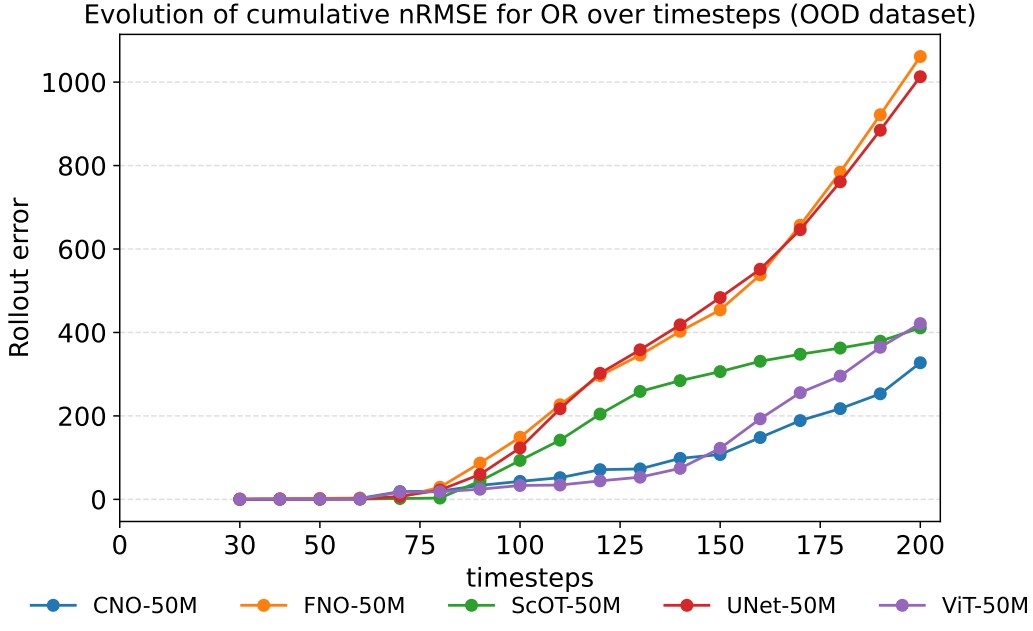

Figure 24: Temporal rollout of cumulative nRMSE over OR for the LIDE-OOD-Experiment PDUV_F_(3,1) for all 50M models.

## D.2 Error rollout over timesteps for the SIDA dataset

The evolution of nRMSE is studied in a cumulative fashion over timesteps for 1M and 50M models on the test trajectories of both In-Distribution (ID) and Out-of-Distribution (OOD) SIDA datasets.

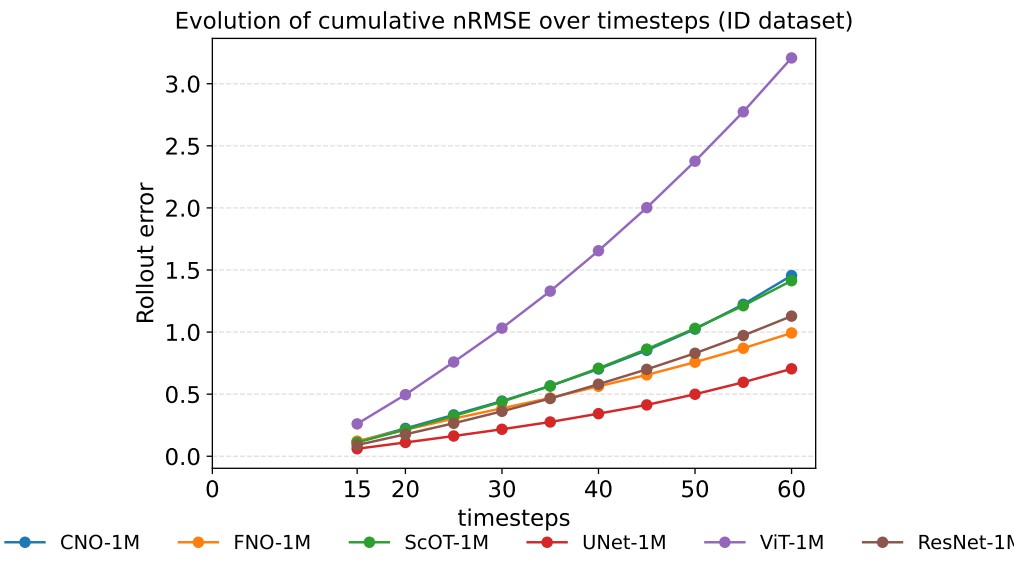

Figure 25: Temporal rollout of cumulative nRMSE over output channels for the SIDA-ID-Experiment PDUV_F_(3,1) for all 1M models.

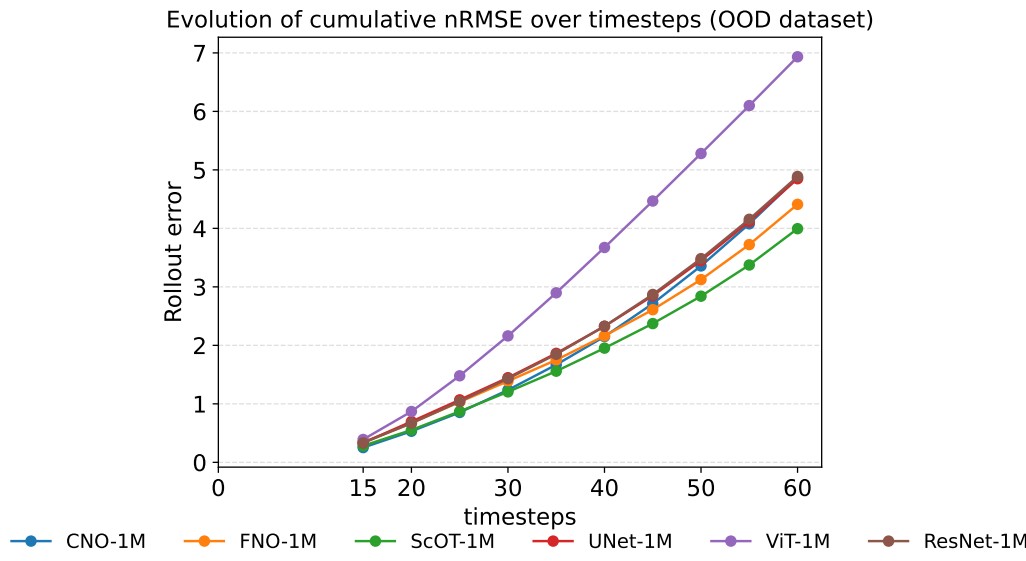

Figure 26: Temporal rollout of cumulative nRMSE over output channels for the SIDA-OOD-Experiment PDUV_F_(3,1) for all 1M models.

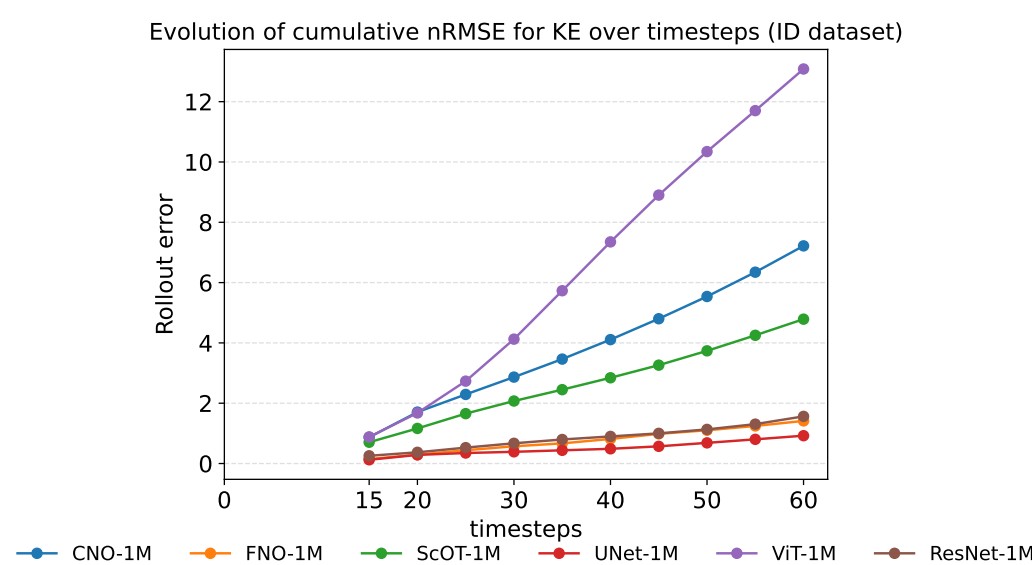

Figure 27: Temporal rollout of cumulative nRMSE over KE for the SIDA-ID-Experiment PDUV_F_(3,1) for all 1M models.

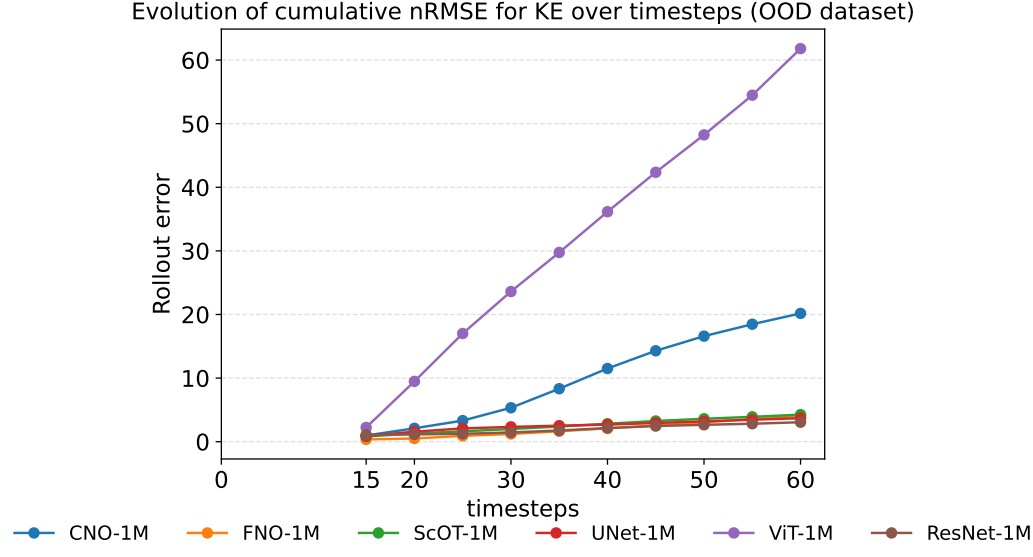

Figure 28: Temporal rollout of cumulative nRMSE over KE for the SIDA-OOD-Experiment PDUV_F_(3,1) for all 1M models.

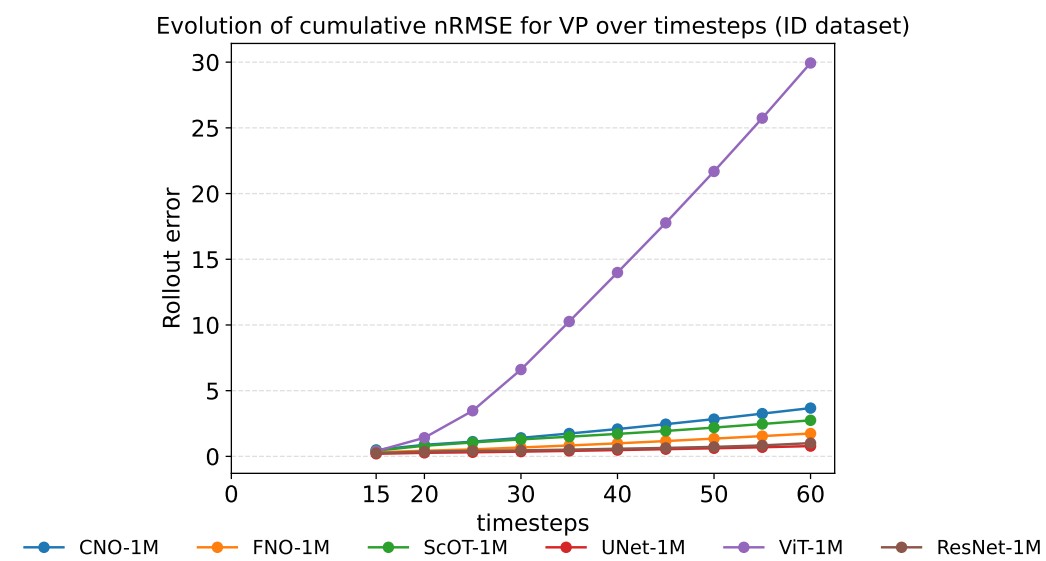

Figure 29: Temporal rollout of cumulative nRMSE over VP for the SIDA-ID-Experiment PDUV_F_(3,1) for all 1M models.

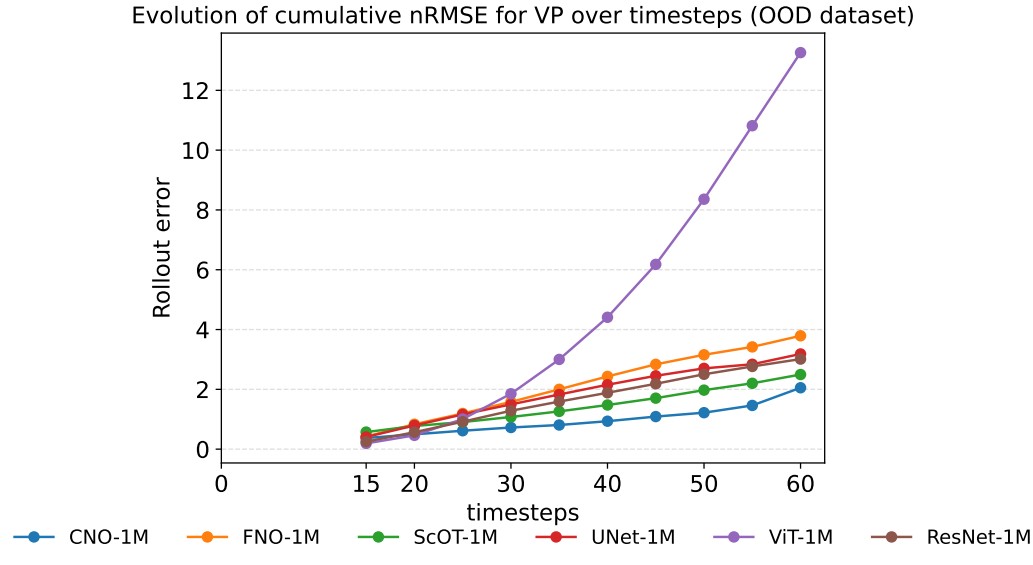

Figure 30: Temporal rollout of cumulative nRMSE over VP for the SIDA-OOD-Experiment PDUV_F_(3,1) for all 1M models.

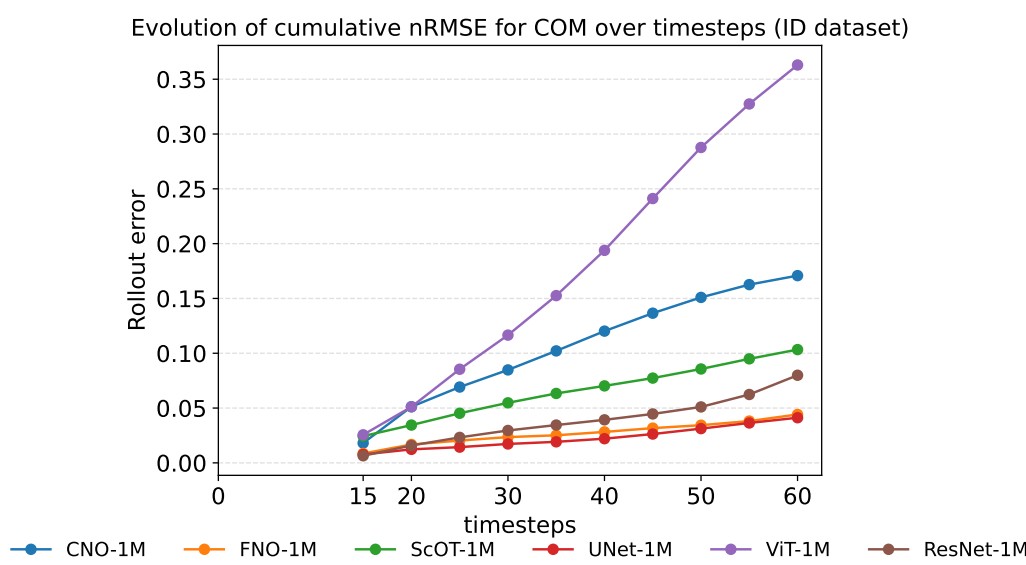

Figure 31: Temporal rollout of cumulative nRMSE over COM for the SIDA-ID-Experiment PDUV_F_(3,1) for all 1M models.

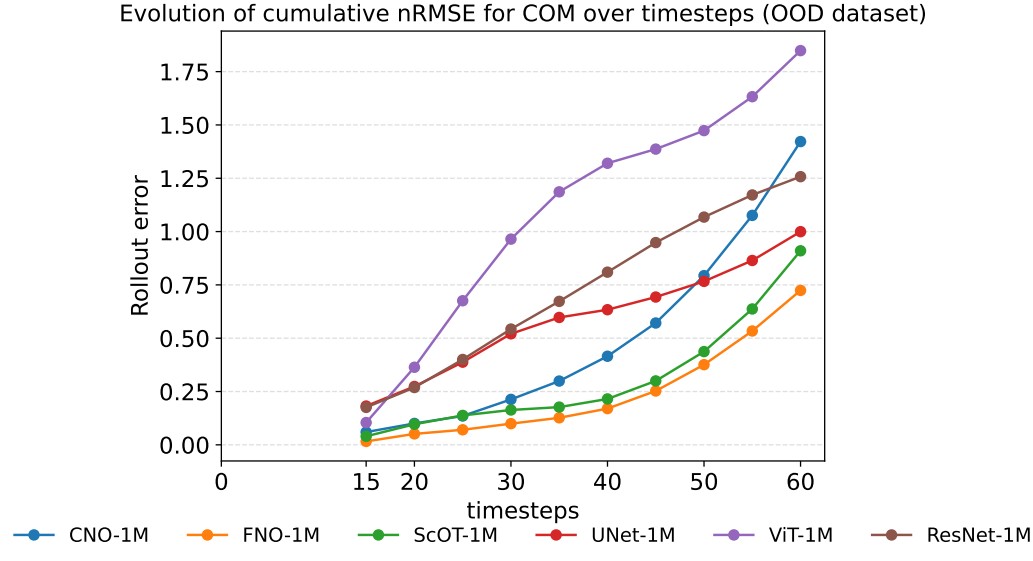

Figure 32: Temporal rollout of cumulative nRMSE over COM for the SIDA-OOD-Experiment PDUV_F_(3,1) for all 1M models.

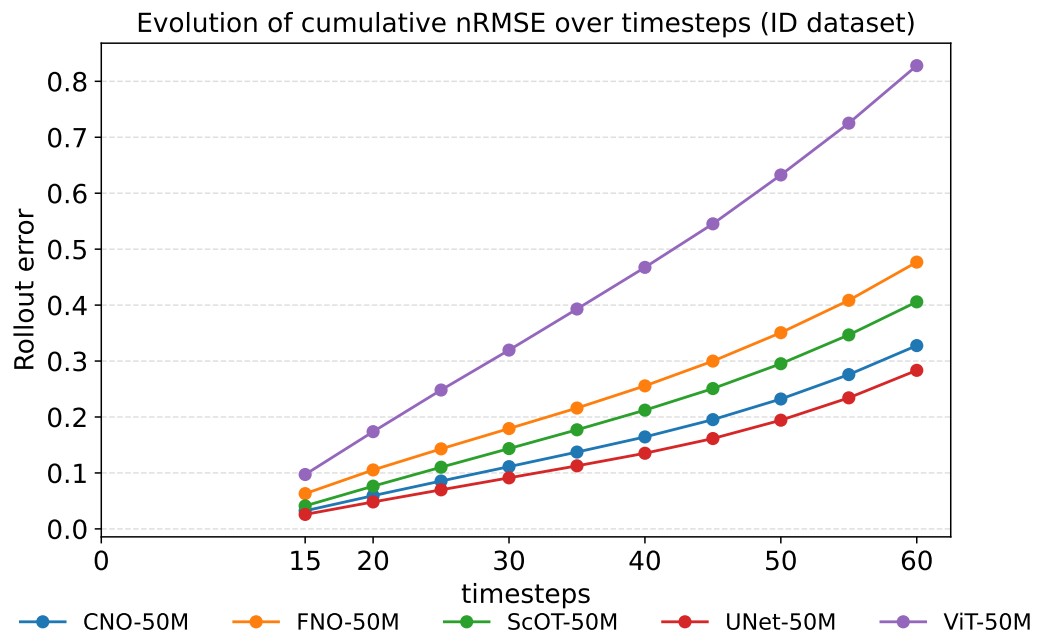

Figure 33: Temporal rollout of cumulative nRMSE over output channels for the SIDA-ID-Experiment PDUV_F_(3,1) for all 50M models.

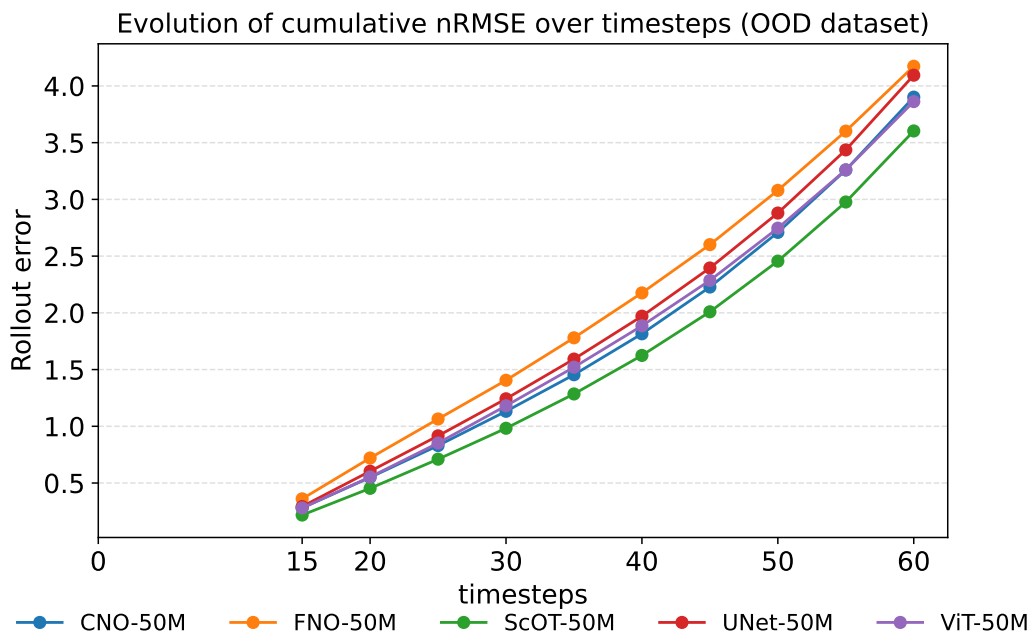

Figure 34: Temporal rollout of cumulative nRMSE over output channels for the SIDA-OOD-Experiment PDUV_F_(3,1) for all 50M models.

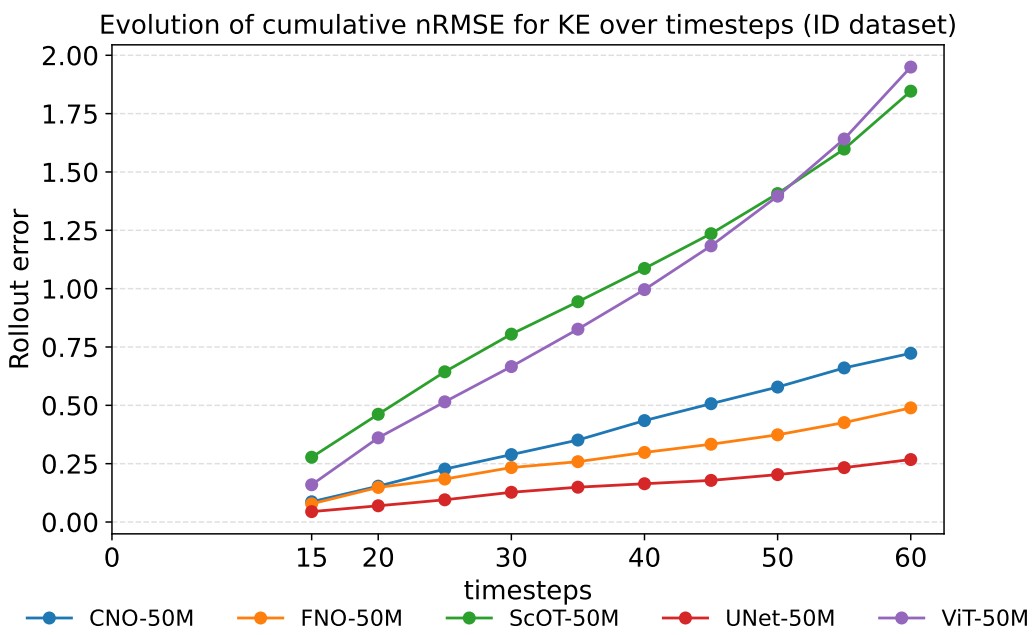

Figure 35: Temporal rollout of cumulative nRMSE over KE for the SIDA-ID-Experiment PDUV_F_(3,1) for all 50M models.

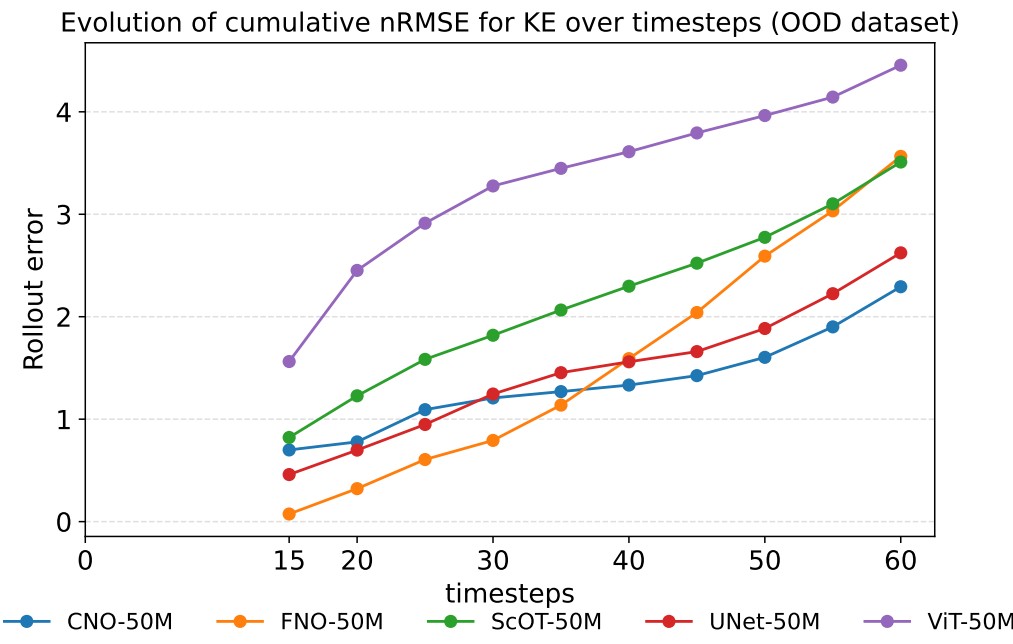

Figure 36: Temporal rollout of cumulative nRMSE over KE for the SIDA-OOD-Experiment PDUV_F_(3,1) for all 50M models.

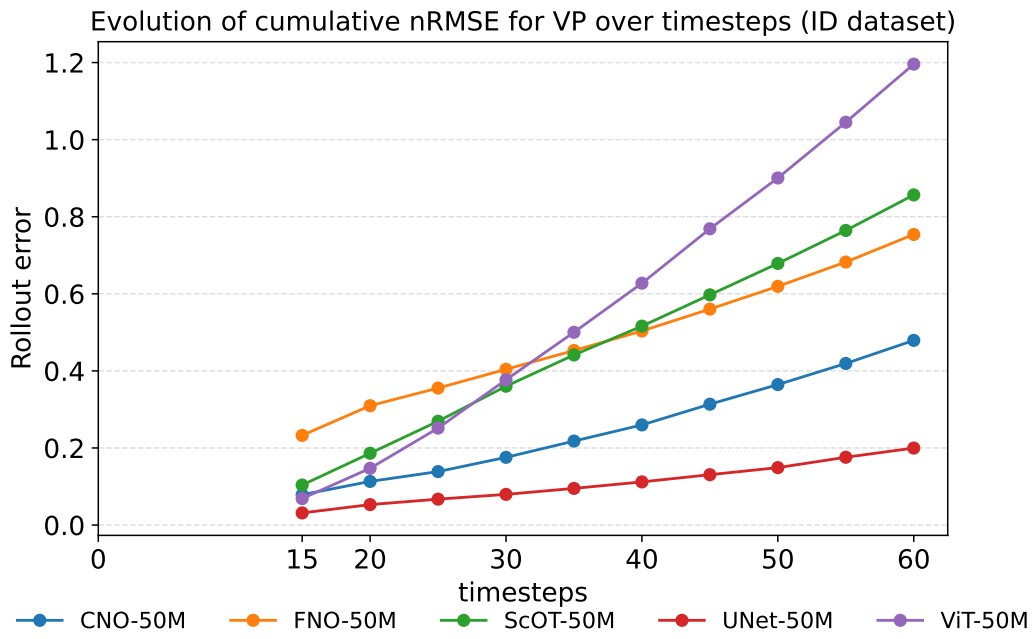

Figure 37: Temporal rollout of cumulative nRMSE over VP for the SIDA-ID-Experiment PDUV_F_(3,1) for all 50M models.

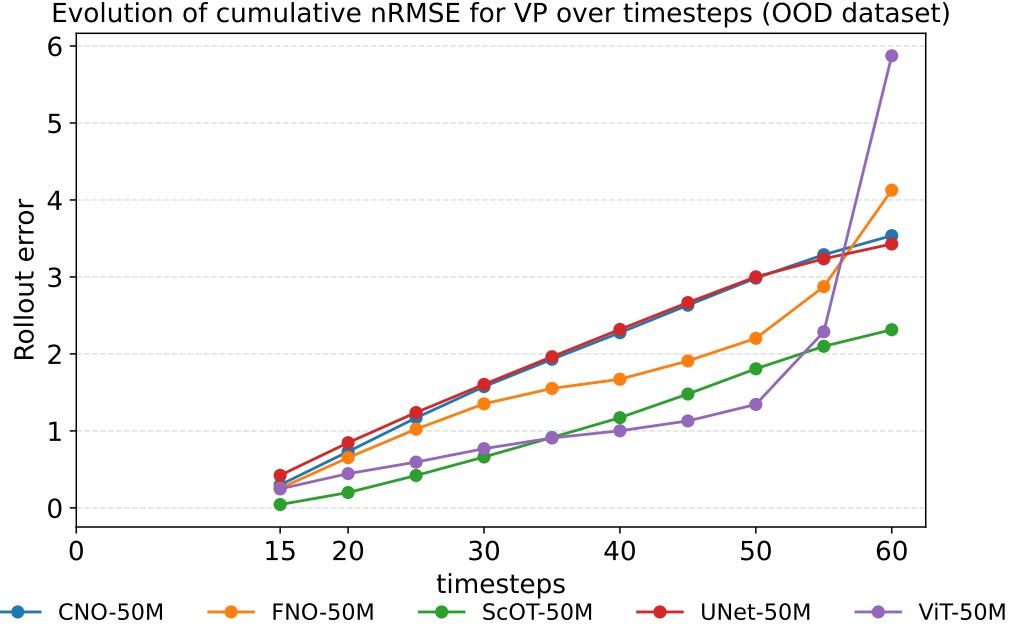

Figure 38: Temporal rollout of cumulative nRMSE over VP for the SIDA-OOD-Experiment PDUV_F_(3,1) for all 50M models.

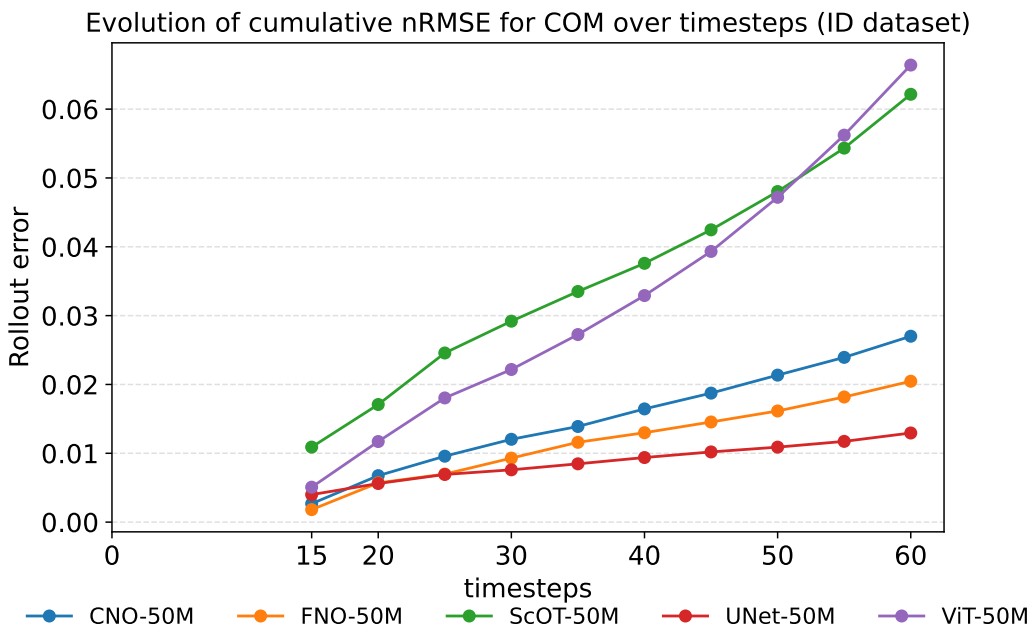

Figure 39: Temporal rollout of cumulative nRMSE over COM for the SIDA-ID-Experiment PDUV_F_(3,1) for all 50M models.

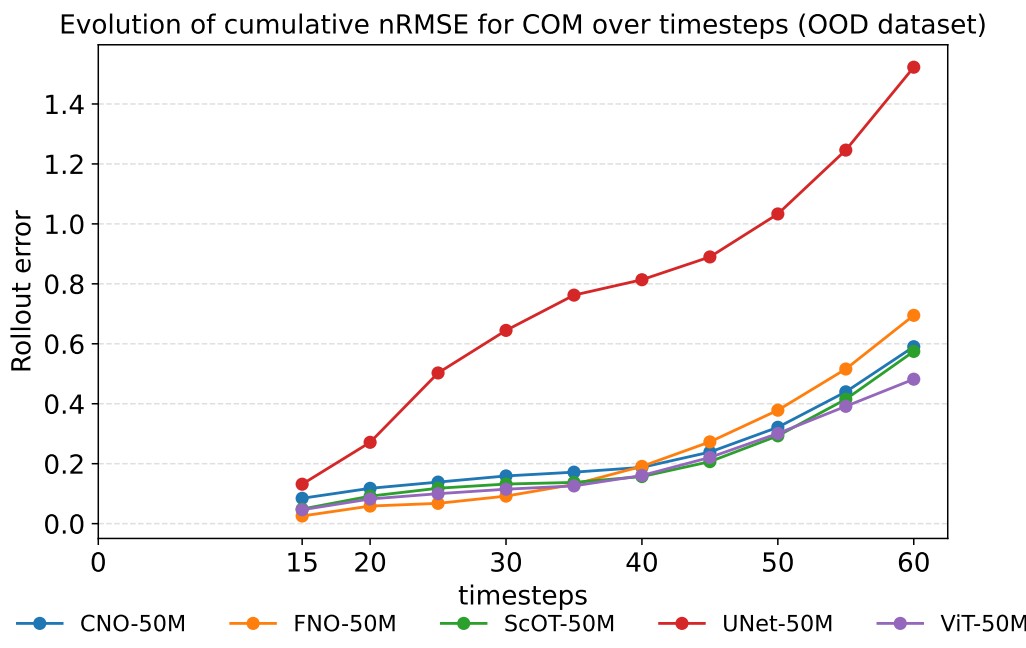

Figure 40: Temporal rollout of cumulative nRMSE over COM for the SIDA-OOD-Experiment PDUV_F_(3,1) for all 50M models.

# E  TABLES

## E.1  ERROR-TYPE 1 METRICS FOR THE LIDE DATASET

In this section, we present the nRMSE of Type 1 **from section 4.5** for all experiments on the LIDE In- and Out of Distribution dataset. The error is represented over four quantities of interest. The output channels column represents the relative RMSE across all output channels and test trajectories, considering the maximum allowable rollout steps. OR represents the error over the outer radius of the droplet. KE and VP represent the error over the kinetic energy and vorticity production of the whole system.

Table 44: Error-type 1 for experiment P_F_(1,1) for the LIDE dataset across all 1M models.

| MODEL | ID | | | | OOD | | | |
|---|---|---|---|---|---|---|---|---|
| | output channels | OR | KE | VP | output channels | OR | KE | VP |
| CNO | 0.5036 | - | - | - | 1.4540 | - | - | - |
| FNO | 0.4514 | - | - | - | 0.8434 | - | - | - |
| ResNet | 0.8425 | - | - | - | 0.9781 | - | - | - |
| ScOT | 0.6704 | - | - | - | 0.8929 | - | - | - |
| Transolver | 0.5548 | - | - | - | 0.8193 | - | - | - |
| UNet | 0.6122 | - | - | - | 0.9712 | - | - | - |
| ViT | 0.6313 | - | - | - | 0.8377 | - | - | - |

Table 45: Error-type 1 for experiment P_F_(1,1) for the LIDE dataset across all 50M models.

| MODEL | ID | | | | OOD | | | |
|---|---|---|---|---|---|---|---|---|
| | output channels | OR | KE | VP | output channels | OR | KE | VP |
| CNO | 0.4558 | - | - | - | 1.6325 | - | - | - |
| FNO | 0.4377 | - | - | - | 0.8540 | - | - | - |
| ScOT | 0.4835 | - | - | - | 0.8752 | - | - | - |
| UNet | 0.5040 | - | - | - | 0.9437 | - | - | - |
| ViT | 0.5079 | - | - | - | 0.8232 | - | - | - |

Table 46: Error-type 1 for experiment P_F_(3,1) for the LIDE dataset across all 1M models.

| MODEL | ID | | | | OOD | | | |
|---|---|---|---|---|---|---|---|---|
| | output channels | OR | KE | VP | output channels | OR | KE | VP |
| CNO | 0.2824 | - | - | - | 0.9684 | - | - | - |
| FNO | 0.6141 | - | - | - | 0.9833 | - | - | - |
| ResNet | 0.8883 | - | - | - | 1.0743 | - | - | - |
| ScOT | 0.6511 | - | - | - | 0.7819 | - | - | - |
| Transolver | 0.4400 | - | - | - | 0.7585 | - | - | - |
| UNet | 0.3964 | - | - | - | 0.9759 | - | - | - |
| ViT | 0.8650 | - | - | - | 1.0220 | - | - | - |

Table 47: Error-type 1 for experiment P_F_(3,1) for the LIDE dataset across all 50M models.

| MODEL | ID | | | | OOD | | | |
| | output channels | OR | KE | VP | output channels | OR | KE | VP |
|---|---|---|---|---|---|---|---|---|
| CNO | 0.3665 | - | - | - | 0.8253 | - | - | - |
| FNO | 0.4943 | - | - | - | 0.9420 | - | - | - |
| ScOT | 0.4988 | - | - | - | 0.7273 | - | - | - |
| UNet | 0.3230 | - | - | - | 0.9112 | - | - | - |
| ViT | 0.5398 | - | - | - | 0.8904 | - | - | - |

Table 48: Error-type 1 for experiment P_T_(1,1) for the LIDE dataset across all 1M models.

| MODEL | ID | | | | OOD | | | |
| | output channels | OR | KE | VP | output channels | OR | KE | VP |
|---|---|---|---|---|---|---|---|---|
| CNO | 0.3746 | - | - | - | 1.0632 | - | - | - |
| FNO | 0.4191 | - | - | - | 0.8097 | - | - | - |
| ResNet | 0.5210 | - | - | - | 1.1980 | - | - | - |
| ScOT | 0.5417 | - | - | - | 3.2936 | - | - | - |
| Transolver | 0.2742 | - | - | - | 0.7994 | - | - | - |
| UNet | 0.3959 | - | - | - | 1.0804 | - | - | - |
| ViT | 0.4802 | - | - | - | 0.8444 | - | - | - |

Table 49: Error-type 1 for experiment P_T_(1,1) for the LIDE dataset across all 50M models.

| MODEL | ID | | | | OOD | | | |
| | output channels | OR | KE | VP | output channels | OR | KE | VP |
|---|---|---|---|---|---|---|---|---|
| CNO | 0.3518 | - | - | - | 0.9494 | - | - | - |
| FNO | 0.3365 | - | - | - | 1.0315 | - | - | - |
| ScOT | 0.1841 | - | - | - | 0.8564 | - | - | - |
| UNet | 0.1689 | - | - | - | 0.8466 | - | - | - |
| ViT | 0.1382 | - | - | - | 0.6708 | - | - | - |

Table 50: Error-type 1 for experiment P_T_(3,1) for the LIDE dataset across all 1M models.

| MODEL | ID | | | | OOD | | | |
| | output channels | OR | KE | VP | output channels | OR | KE | VP |
|---|---|---|---|---|---|---|---|---|
| CNO | 0.4647 | - | - | - | 1.0615 | - | - | - |
| FNO | 0.4439 | - | - | - | 0.9360 | - | - | - |
| ResNet | 0.7218 | - | - | - | 1.8583 | - | - | - |
| ScOT | 0.5154 | - | - | - | 3.9418 | - | - | - |
| Transolver | 0.3271 | - | - | - | 0.8181 | - | - | - |
| UNet | 0.4858 | - | - | - | 1.2237 | - | - | - |
| ViT | 0.5381 | - | - | - | 1.0488 | - | - | - |

Table 51: Error-type 1 for experiment P_T_(3,1) for the LIDE dataset across all 50M models.

| MODEL | ID | | | | OOD | | | |
|---|---|---|---|---|---|---|---|---|
| | output channels | OR | KE | VP | output channels | OR | KE | VP |
| CNO | 0.3652 | - | - | - | 0.9529 | - | - | - |
| FNO | 0.3219 | - | - | - | 0.9391 | - | - | - |
| ScOT | 0.2252 | - | - | - | 2.5944 | - | - | - |
| UNet | 0.1979 | - | - | - | 0.8472 | - | - | - |
| ViT | 0.1912 | - | - | - | 0.8504 | - | - | - |

Table 52: Error-type 1 for experiment PDUV_F_(1,1) for the LIDE dataset across all 1M models.

| MODEL | ID | | | | OOD | | | |
|---|---|---|---|---|---|---|---|---|
| | output channels | OR | KE | VP | output channels | OR | KE | VP |
| CNO | 0.2629 | 0.3114 | 0.2102 | 0.2316 | 0.8049 | 1.9745 | 0.7422 | 0.7015 |
| FNO | 0.4939 | 0.7593 | 1.4189 | 1.6107 | 0.8035 | 1.7849 | 0.8039 | 0.8758 |
| ResNet | 0.8566 | 1.1304 | 2.5096 | 1.1816 | 0.9591 | 2.0483 | 1.7894 | 0.7860 |
| ScOT | 0.3322 | 0.3044 | 0.3134 | 0.3627 | 0.7746 | 0.9406 | 0.5202 | 0.8277 |
| Transolver | 0.5692 | 1.2263 | 0.7104 | 0.5402 | 0.8377 | 2.2660 | 0.7232 | 0.7507 |
| UNet | 0.3817 | 0.4428 | 0.2472 | 0.2280 | 0.9356 | 1.3699 | 0.7342 | 0.9043 |
| ViT | 0.7592 | 0.4418 | 1.1000 | 2.5903 | 0.8513 | 1.3258 | 0.6202 | 0.6930 |

Table 53: Error-type 1 for experiment PDUV_F_(1,1) for the LIDE dataset across all 50M models.

| MODEL | ID | | | | OOD | | | |
|---|---|---|---|---|---|---|---|---|
| | output channels | OR | KE | VP | output channels | OR | KE | VP |
| CNO | 0.2647 | 0.3075 | 0.1199 | 0.1540 | 0.6902 | 1.2532 | 0.5474 | 0.7395 |
| FNO | 0.3472 | 0.6953 | 0.7417 | 1.1417 | 0.7403 | 1.8706 | 1.3258 | 1.0421 |
| ScOT | 0.2806 | 0.6725 | 0.1422 | 0.1706 | 0.7444 | 1.7365 | 0.5360 | 0.7541 |
| UNet | 0.2673 | 0.4714 | 0.0647 | 0.1123 | 0.8562 | 1.5476 | 0.9008 | 0.8546 |
| ViT | 0.2625 | 0.7581 | 0.1653 | 0.2395 | 0.6853 | 1.1653 | 0.4153 | 0.7482 |

Table 54: Error-type 1 for experiment PDUV_F_(3,1) for the LIDE dataset across all 1M models.

| MODEL | ID | | | | OOD | | | |
|---|---|---|---|---|---|---|---|---|
| | output channels | OR | KE | VP | output channels | OR | KE | VP |
| CNO | 0.1641 | 0.7208 | 0.1182 | 0.2360 | 0.7965 | 1.3917 | 2.6376 | 0.7217 |
| FNO | 0.3814 | 0.7405 | 1.5689 | 2.0840 | 0.6766 | 1.9568 | 1.2789 | 1.0530 |
| ResNet | 0.7244 | 1.0878 | 0.2947 | 1.4805 | 0.6969 | 2.2930 | 0.6900 | 0.7133 |
| ScOT | 0.2234 | 0.4091 | 0.3476 | 0.3303 | 0.9286 | 1.7081 | 4.6177 | 0.8089 |
| Transolver | 0.4091 | 1.2833 | 0.7547 | 0.4055 | 0.6324 | 2.0543 | 14.7765 | 1.3631 |
| UNet | 0.2140 | 0.3373 | 0.1471 | 0.1766 | 0.6814 | 1.6791 | 0.5000 | 0.6830 |
| ViT | 0.5237 | 0.8853 | 1.5604 | 1.3256 | 0.7154 | 1.3003 | 2.6848 | 1.6023 |

Table 55: Error-type 1 for experiment PDUV_F_(3,1) for the LIDE dataset across all 50M models.

| MODEL | ID | | | | OOD | | | |
|---|---|---|---|---|---|---|---|---|
| | output channels | OR | KE | VP | output channels | OR | KE | VP |
| CNO | 0.1658 | 0.7696 | 0.0995 | 0.1413 | 0.5502 | 1.2733 | 0.5196 | 0.7047 |
| FNO | 0.2538 | 0.7069 | 0.7270 | 0.9354 | 0.7913 | 1.8454 | 0.7272 | 0.6834 |
| ScOT | 0.1570 | 0.8316 | 0.1831 | 0.1294 | 0.7089 | 1.3724 | 1.9937 | 0.6623 |
| UNet | 0.1383 | 0.3408 | 0.0601 | 0.0677 | 0.5854 | 1.5623 | 0.5511 | 0.7349 |
| ViT | 0.1792 | 0.8206 | 0.2179 | 0.1896 | 0.4485 | 1.2850 | 2.7728 | 0.4836 |

Table 56: Error-type 1 for experiment P[ES]_F_(1,1) for the LIDE dataset across all 1M models.

| MODEL | ID | | | | OOD | | | |
|---|---|---|---|---|---|---|---|---|
| | output channels | OR | KE | VP | output channels | OR | KE | VP |
| CNO | 1.6851 | - | - | - | 1.6036 | - | - | - |
| FNO | 1.6562 | - | - | - | 1.1943 | - | - | - |
| ResNet | 1.3139 | - | - | - | 1.4039 | - | - | - |
| ScOT | 1.3833 | - | - | - | 1.1345 | - | - | - |
| Transolver | 1.0443 | - | - | - | 0.9856 | - | - | - |
| UNet | 1.2778 | - | - | - | 1.0259 | - | - | - |
| ViT | 1.3596 | - | - | - | 1.0713 | - | - | - |

Table 57: Error-type 1 for experiment P[ES]_F_(1,1) for the LIDE dataset across all 50M models.

| MODEL | ID | | | | OOD | | | |
|---|---|---|---|---|---|---|---|---|
| | output channels | OR | KE | VP | output channels | OR | KE | VP |
| CNO | 1.7389 | - | - | - | 1.4028 | - | - | - |
| FNO | 1.8858 | - | - | - | 1.2497 | - | - | - |
| ScOT | 0.9373 | - | - | - | 0.9780 | - | - | - |
| UNet | 1.8593 | - | - | - | 1.5409 | - | - | - |
| ViT | 0.9180 | - | - | - | 0.9606 | - | - | - |

Table 58: Error-type 1 for experiment P[ES]_F_(3,1) for the LIDE dataset across all 1M models.

| MODEL | ID | | | | OOD | | | |
|---|---|---|---|---|---|---|---|---|
| | output channels | OR | KE | VP | output channels | OR | KE | VP |
| CNO | 1.0090 | - | - | - | 1.0005 | - | - | - |
| FNO | 1.1216 | - | - | - | 1.2849 | - | - | - |
| ResNet | 1.0200 | - | - | - | 1.3084 | - | - | - |
| ScOT | 0.7419 | - | - | - | 0.8235 | - | - | - |
| Transolver | 1.0213 | - | - | - | 1.0578 | - | - | - |
| UNet | 1.6746 | - | - | - | 1.7769 | - | - | - |
| ViT | 1.0429 | - | - | - | 1.1444 | - | - | - |

Table 59: Error-type 1 for experiment P[ES]_F_(3,1) for the LIDE dataset across all 50M models.

| MODEL | ID | | | | OOD | | | |
|---|---|---|---|---|---|---|---|---|
| | output channels | OR | KE | VP | output channels | OR | KE | VP |
| CNO | 0.9660 | - | - | - | 1.4007 | - | - | - |
| FNO | 1.4601 | - | - | - | 1.3018 | - | - | - |
| ScOT | 0.6965 | - | - | - | 0.8378 | - | - | - |
| UNet | 1.3407 | - | - | - | 1.6829 | - | - | - |
| ViT | 0.8213 | - | - | - | 1.0365 | - | - | - |

Table 60: Error-type 1 for experiment PDUV[ES]_F_(1,1) for the LIDE dataset across all 1M models.

| MODEL | ID | | | | OOD | | | |
|---|---|---|---|---|---|---|---|---|
| | output channels | OR | KE | VP | output channels | OR | KE | VP |
| CNO | 1.2229 | 0.6435 | 4.6978 | 0.7810 | 1.2592 | 1.5198 | 7.5844 | 0.6541 |
| FNO | 1.0630 | 0.9357 | 2.0852 | 0.6448 | 1.1524 | 2.1351 | 1.1364 | 0.8400 |
| ResNet | 0.7373 | 0.9818 | 2.1267 | 4.0487 | 0.9972 | 0.9069 | 3.2353 | 1.4468 |
| ScOT | 1.2051 | 0.7292 | 3.7019 | 0.6542 | 0.9178 | 1.5549 | 1.7931 | 0.8171 |
| Transolver | 1.1740 | 1.0733 | 3.4391 | 6.1363 | 0.9761 | 1.8676 | 1.2208 | 1.4010 |
| UNet | 0.9590 | 0.9442 | 1.0926 | 0.5010 | 1.0156 | 0.9468 | 0.7576 | 0.7147 |
| ViT | 1.6079 | 0.7157 | 84.3898 | 19.0520 | 1.2845 | 1.3337 | 25.3722 | 3.6711 |

Table 61: Error-type 1 for experiment PDUV[ES]_F_(1,1) for the LIDE dataset across all 50M models.

| MODEL | ID | | | | OOD | | | |
|---|---|---|---|---|---|---|---|---|
| | output channels | OR | KE | VP | output channels | OR | KE | VP |
| CNO | 1.0375 | 0.3462 | 2.2243 | 0.6907 | 1.1061 | 0.7104 | 4.1761 | 0.6623 |
| FNO | 1.1017 | 0.3970 | 1.1027 | 0.8632 | 1.0473 | 0.8639 | 0.6317 | 0.7118 |
| ScOT | 1.0842 | 0.3138 | 5.1516 | 1.2275 | 1.0623 | 0.9669 | 2.0632 | 0.8507 |
| UNet | 1.2581 | 0.3212 | 10.1976 | 0.9566 | 1.0858 | 0.8798 | 8.4109 | 1.0080 |
| ViT | 0.5288 | 0.3013 | 1.1427 | 0.4881 | 0.7361 | 0.9410 | 0.4628 | 0.6720 |

Table 62: Error-type 1 for experiment PDUV[ES]_F_(3,1) for the LIDE dataset across all 1M models.

| MODEL | ID | | | | OOD | | | |
|---|---|---|---|---|---|---|---|---|
| | output channels | OR | KE | VP | output channels | OR | KE | VP |
| CNO | 0.8496 | 0.8761 | 2.3398 | 0.3785 | 0.8293 | 1.0048 | 1.9510 | 0.6839 |
| FNO | 0.9314 | 0.5236 | 1.5661 | 0.5155 | 1.0127 | 1.1182 | 0.5089 | 0.6728 |
| ResNet | 0.8098 | 0.8470 | 2.2236 | 1.9128 | 0.9036 | 0.9456 | 1.2876 | 0.8280 |
| ScOT | 0.7853 | 0.5273 | 2.4910 | 0.6505 | 1.3457 | 1.6916 | 6.7820 | 0.8291 |
| Transolver | 0.7752 | 1.1435 | 8.5083 | 1.1835 | 0.8184 | 1.9053 | 12.2341 | 1.2247 |
| UNet | 0.8315 | 0.3880 | 1.5762 | 0.5594 | 1.0015 | 0.8827 | 9.4459 | 1.4276 |
| ViT | 0.7199 | 0.7572 | 1.7756 | 3.1967 | 0.8519 | 1.0887 | 2.0591 | 2.0040 |

Table 63: Error-type 1 for experiment PDUV[ES]_F_(3,1) for the LIDE dataset across all 50M models.

| MODEL | ID | | | | OOD | | | |
| | output channels | OR | KE | VP | output channels | OR | KE | VP |
|---|---|---|---|---|---|---|---|---|
| CNO | 0.7444 | 0.3423 | 0.4881 | 0.4047 | 0.8804 | 0.8859 | 0.5464 | 0.7090 |
| FNO | 0.7419 | 0.3347 | 0.6132 | 0.3902 | 0.9423 | 1.3608 | 1.0011 | 1.4644 |
| ScOT | 0.7998 | 0.3195 | 3.1428 | 0.5878 | 1.1444 | 1.1325 | 3.7367 | 0.6780 |
| UNet | 0.6749 | 0.3143 | 1.3913 | 0.4287 | 0.9349 | 0.9718 | 2.6840 | 0.8234 |
| ViT | 0.5741 | 0.8446 | 0.6115 | 0.5772 | 0.6063 | 1.1230 | 2.0634 | 0.4701 |

Table 64: Error-type 1 for experiment P_F_(3,2) for the LIDE dataset across all 1M models.

| MODEL | ID | | | | OOD | | | |
| | output channels | OR | KE | VP | output channels | OR | KE | VP |
|---|---|---|---|---|---|---|---|---|
| CNO | 0.3370 | - | - | - | 0.8082 | - | - | - |
| FNO | 0.5785 | - | - | - | 0.9104 | - | - | - |
| ResNet | 0.7244 | - | - | - | 0.9998 | - | - | - |
| ScOT | 0.5407 | - | - | - | 0.7937 | - | - | - |
| Transolver | 0.5719 | - | - | - | 0.8974 | - | - | - |
| UNet | 0.4697 | - | - | - | 0.9480 | - | - | - |
| ViT | 0.8045 | - | - | - | 1.0514 | - | - | - |

Table 65: Error-type 1 for experiment P_F_(3,2) for the LIDE dataset across all 50M models.

| MODEL | ID | | | | OOD | | | |
| | output channels | OR | KE | VP | output channels | OR | KE | VP |
|---|---|---|---|---|---|---|---|---|
| CNO | 0.3623 | - | - | - | 1.0601 | - | - | - |
| FNO | 0.4481 | - | - | - | 0.8793 | - | - | - |
| ScOT | 0.5479 | - | - | - | 0.7460 | - | - | - |
| UNet | 0.3347 | - | - | - | 0.8520 | - | - | - |
| ViT | 0.5914 | - | - | - | 0.8713 | - | - | - |

Table 66: Error-type 1 for experiment PDUV_F_(3,2) for the LIDE dataset across all 1M models.

| MODEL | ID | | | | OOD | | | |
| | output channels | OR | KE | VP | output channels | OR | KE | VP |
|---|---|---|---|---|---|---|---|---|
| CNO | 0.1681 | 0.5961 | 0.2427 | 0.3172 | 0.4908 | 1.3585 | 0.5875 | 0.7162 |
| FNO | 0.2659 | 0.6912 | 0.4239 | 0.5619 | 0.6846 | 1.8140 | 0.7110 | 0.8241 |
| ResNet | 0.3439 | 0.7605 | 0.2512 | 1.1572 | 0.6110 | 1.4525 | 0.9998 | 0.7292 |
| ScOT | 0.1991 | 0.4183 | 0.5548 | 0.4106 | 0.5261 | 1.6937 | 2.0553 | 0.7480 |
| Transolver | 0.6848 | 0.9282 | 3.3758 | 1.9064 | 0.7317 | 1.4803 | 5.4101 | 2.4985 |
| UNet | 0.2602 | 0.3964 | 0.1902 | 0.3175 | 0.5980 | 1.4825 | 0.6453 | 0.6684 |
| ViT | 0.4009 | 0.7294 | 0.6847 | 0.2659 | 0.8038 | 1.9366 | 17.3108 | 0.7356 |

Table 67: Error-type 1 for experiment PDUV_F_(3,2) for the LIDE dataset across all 50M models.

| MODEL | ID | | | | OOD | | | |
| | output channels | OR | KE | VP | output channels | OR | KE | VP |
|---|---|---|---|---|---|---|---|---|
| CNO | 0.1772 | 0.7030 | 0.1803 | 0.1979 | 0.5377 | 1.9170 | 0.4089 | 0.6949 |
| FNO | 0.2015 | 0.5992 | 0.1149 | 0.2680 | 0.7814 | 1.7143 | 0.6559 | 0.6455 |
| ScOT | 0.1596 | 0.6556 | 0.2521 | 0.1761 | 0.5054 | 1.5690 | 0.6536 | 0.6506 |
| UNet | 0.1398 | 0.5207 | 0.1052 | 0.1055 | 0.6313 | 1.2962 | 0.4841 | 0.6589 |
| ViT | 0.1584 | 0.8388 | 0.3299 | 0.2562 | 0.4713 | 1.8939 | 3.1967 | 0.5349 |

Table 68: Error-type 1 for experiment PDUV_T_(1,1) for the LIDE dataset across all 1M models.

| MODEL | ID | | | | OOD | | | |
| | output channels | OR | KE | VP | output channels | OR | KE | VP |
|---|---|---|---|---|---|---|---|---|
| CNO | 0.3308 | 0.3259 | 1.2956 | 0.3905 | 1.0250 | 1.3658 | 15.1482 | 0.8278 |
| FNO | 0.3758 | 0.7462 | 0.2703 | 0.7034 | 0.9118 | 2.0678 | 0.6794 | 1.4235 |
| ResNet | 0.5823 | 0.8530 | 0.7443 | 3.0600 | 0.8987 | 1.2044 | 2.1359 | 0.9952 |
| ScOT | 0.2423 | 0.8981 | 0.6786 | 0.4264 | 1.6251 | 1.2974 | 6.4974 | 0.7815 |
| Transolver | 0.8041 | 1.0787 | 2.6660 | 3.2419 | 0.8448 | 1.6026 | 0.8981 | 1.5772 |
| UNet | 0.3366 | 0.3787 | 0.2565 | 0.2655 | 0.9788 | 1.1754 | 2.3093 | 0.7987 |
| ViT | 0.5406 | 0.5891 | 1.7829 | 1.0861 | 1.3901 | 1.0735 | 27.0802 | 5.5687 |

Table 69: Error-type 1 for experiment PDUV_T_(1,1) for the LIDE dataset across all 50M models.

| MODEL | ID | | | | OOD | | | |
| | output channels | OR | KE | VP | output channels | OR | KE | VP |
|---|---|---|---|---|---|---|---|---|
| CNO | 0.2563 | 0.5913 | 0.3306 | 0.2356 | 0.9424 | 0.9781 | 2.7716 | 0.9026 |
| FNO | 0.2186 | 0.5734 | 0.1571 | 0.3101 | 0.8777 | 1.9468 | 2.2576 | 2.1369 |
| ScOT | 0.1154 | 0.5328 | 0.1918 | 0.1369 | 1.1404 | 2.2448 | 46.2113 | 2.1308 |
| UNet | 0.1541 | 0.4554 | 0.1905 | 0.1161 | 0.8473 | 1.1420 | 0.9216 | 0.9274 |
| ViT | 0.1190 | 0.6870 | 0.1765 | 0.1727 | 0.6476 | 1.8988 | 0.8740 | 0.6958 |

Table 70: Error-type 1 for experiment PDUV_T_(3,1) for the LIDE dataset across all 1M models.

| MODEL | ID | | | | OOD | | | |
| | output channels | OR | KE | VP | output channels | OR | KE | VP |
|---|---|---|---|---|---|---|---|---|
| CNO | 0.3545 | 0.3782 | 0.5048 | 0.3388 | 0.9964 | 1.3516 | 1.8233 | 0.7798 |
| FNO | 0.3364 | 0.7212 | 0.6183 | 1.0469 | 0.7403 | 1.9948 | 2.2181 | 1.0348 |
| ResNet | 0.4386 | 0.6767 | 0.6628 | 1.5547 | 0.9144 | 1.0341 | 0.6918 | 0.9492 |
| ScOT | 0.2259 | 0.4166 | 0.5126 | 0.3882 | 0.9113 | 1.3358 | 19.9497 | 0.7177 |
| Transolver | 0.5447 | 1.0129 | 2.8613 | 1.0453 | 0.9236 | 0.9998 | 3.6309 | 0.9388 |
| UNet | 0.3723 | 0.4363 | 0.5094 | 0.2309 | 1.0128 | 1.0576 | 10.6509 | 0.8182 |
| ViT | 0.4118 | 0.6382 | 0.7999 | 0.9934 | 1.3428 | 1.9946 | 7.8630 | 9.3536 |

Table 71: Error-type 1 for experiment PDUV_T_(3,1) for the LIDE dataset across all 50M models.

| MODEL | ID | | | | OOD | | | |
|---|---|---|---|---|---|---|---|---|
| | output channels | OR | KE | VP | output channels | OR | KE | VP |
| CNO | 0.2463 | 0.7879 | 0.2353 | 0.2650 | 0.8835 | 1.4789 | 2.2330 | 0.7969 |
| FNO | 0.1969 | 0.6082 | 0.0841 | 0.2513 | 0.8324 | 1.5054 | 1.0485 | 1.0591 |
| ScOT | 0.1111 | 0.8634 | 0.2411 | 0.1482 | 0.9309 | 1.4528 | 6.9306 | 0.8574 |
| UNet | 0.1407 | 0.5266 | 0.1017 | 0.1425 | 0.7480 | 0.8901 | 3.6310 | 0.8814 |
| ViT | 0.1293 | 0.6679 | 0.2070 | 0.1606 | 0.5036 | 1.4336 | 1.1170 | 0.3788 |

## E.2 ERROR-TYPE 1 METRICS FOR THE SIDA DATASET

In this section, we present the nRMSE **as defined in section 4.5** of Type 1 for all experiments on the SIDA In-and Out of Distribution dataset. The error is represented over four quantities of interest. The output channels column represents the relative RMSE across all output channels and test trajectories, considering the maximum allowable rollout steps. COM represents the error over displacement of the droplet's Center of Mass. KE and VP represent the error over the kinetic energy and vorticity production of the whole system, respectively.

Table 72: Error-type 1 for experiment PDUV_F_(1,1) for the SIDA dataset across all 1M models.

| MODEL | ID | | | | OOD | | | |
| | output channels | COM | KE | VP | output channels | COM | KE | VP |
|---|---|---|---|---|---|---|---|---|
| CNO | 0.7350 | 0.2510 | 0.6356 | 0.7034 | 0.9400 | 0.2304 | 0.8165 | 0.8690 |
| FNO | 0.6937 | 0.1068 | 0.7125 | 0.7587 | 0.9734 | 0.4068 | 0.9016 | 0.8879 |
| ResNet | 0.7199 | 0.1097 | 0.7132 | 0.6845 | 0.9816 | 0.4103 | 0.9046 | 0.8655 |
| ScOT | 0.6814 | 0.1124 | 0.6331 | 0.6905 | 0.9582 | 0.4134 | 0.8645 | 0.8727 |
| Transolver | 1.0046 | 0.1246 | 0.6115 | 0.7339 | 1.0749 | 0.3686 | 0.8160 | 0.8206 |
| UNet | 0.6883 | 0.1082 | 0.7101 | 0.6913 | 0.9673 | 0.4082 | 0.9035 | 0.8727 |
| ViT | 0.7152 | 0.0942 | 0.9125 | 1.7978 | 0.9308 | 0.3674 | 0.6793 | 0.5404 |

Table 73: Error-type 1 for experiment PDUV_F_(1,1) for the SIDA dataset across all 50M models.

| MODEL | ID | | | | OOD | | | |
| | output channels | COM | KE | VP | output channels | COM | KE | VP |
|---|---|---|---|---|---|---|---|---|
| CNO | 0.6693 | 0.0987 | 0.6471 | 0.6852 | 0.9580 | 0.3944 | 0.8728 | 0.8692 |
| FNO | 0.6768 | 0.1045 | 0.6949 | 0.7280 | 0.9636 | 0.4038 | 0.8974 | 0.8805 |
| ScOT | 0.6544 | 0.0965 | 0.6185 | 0.6560 | 0.9448 | 0.3904 | 0.8513 | 0.8509 |
| UNet | 0.6760 | 0.1068 | 0.7001 | 0.6774 | 0.9608 | 0.4068 | 0.8997 | 0.8687 |
| ViT | 0.6479 | 0.0972 | 0.6126 | 0.6634 | 0.9373 | 0.3921 | 0.8444 | 0.8565 |

Table 74: Error-type 1 for experiment PDUV_F_(3,1) for the SIDA dataset across all 1M models.

| MODEL | ID | | | | OOD | | | |
| | output channels | COM | KE | VP | output channels | COM | KE | VP |
|---|---|---|---|---|---|---|---|---|
| CNO | 0.1589 | 0.0208 | 0.7489 | 0.2001 | 0.6232 | 0.3265 | 2.7838 | 0.2841 |
| FNO | 0.1081 | 0.0058 | 0.1871 | 0.2550 | 0.5309 | 0.1788 | 0.4071 | 0.4372 |
| ResNet | 0.1366 | 0.0160 | 0.2594 | 0.1918 | 0.5838 | 0.1536 | 0.4090 | 0.4080 |
| ScOT | 0.1479 | 0.0125 | 0.4926 | 0.2315 | 0.4771 | 0.2262 | 0.4276 | 0.3090 |
| Transolver | 0.9153 | 0.5274 | 10.8694 | 8.2858 | 0.9343 | 0.3303 | 8.9317 | 3.5904 |
| UNet | 0.0855 | 0.0051 | 0.1208 | 0.1086 | 0.5933 | 0.1555 | 0.4710 | 0.4239 |
| ViT | 0.3440 | 0.0522 | 1.0397 | 3.3843 | 0.7788 | 0.2326 | 7.8675 | 1.7382 |

Table 75: Error-type 1 for experiment PDUV_F_(3,1) for the SIDA dataset across all 50M models.

| MODEL | ID | | | | OOD | | | |
|---|---|---|---|---|---|---|---|---|
| | output channels | COM | KE | VP | output channels | COM | KE | VP |
| CNO | 0.0432 | 0.0036 | 0.1054 | 0.0694 | 0.4912 | 0.1246 | 0.4211 | 0.4616 |
| FNO | 0.0580 | 0.0028 | 0.0483 | 0.1251 | 0.5014 | 0.1479 | 0.6771 | 0.6162 |
| ScOT | 0.0486 | 0.0076 | 0.2135 | 0.0509 | 0.4607 | 0.1329 | 0.4149 | 0.3405 |
| UNet | 0.0427 | 0.0020 | 0.0375 | 0.0290 | 0.5261 | 0.2744 | 0.3688 | 0.4780 |
| ViT | 0.0910 | 0.0081 | 0.2439 | 0.1242 | 0.4739 | 0.0995 | 0.7688 | 1.2988 |

Table 76: Error-type 1 for experiment PDUV_T_(3,1) for the SIDA dataset across all 1M models.

| MODEL | ID | | | | OOD | | | |
|---|---|---|---|---|---|---|---|---|
| | output channels | COM | KE | VP | output channels | COM | KE | VP |
| CNO | 0.1897 | 0.0560 | 1.2485 | 0.4213 | 329.3752 | 0.2660 | inf | inf |
| FNO | 0.1344 | 0.0115 | 0.3637 | 0.3582 | 0.4198 | 0.1680 | 2.5206 | 0.5613 |
| ResNet | 0.2266 | 0.0148 | 0.4769 | 0.3163 | 0.7322 | 0.2278 | 6.4481 | 0.7256 |
| ScOT | 0.1694 | 0.0241 | 0.7422 | 0.1666 | 0.4334 | 0.1210 | 2.3970 | 0.3141 |
| Transolver | 0.8795 | 0.6770 | 17.3106 | 6.8722 | 0.8577 | 0.3341 | 54.9306 | 4.7863 |
| UNet | 0.1842 | 0.0170 | 0.3507 | 0.1606 | 0.6009 | 0.3724 | 2.0907 | 0.6366 |
| ViT | 0.4162 | 0.0823 | 1.2424 | 3.5928 | 0.5955 | 0.2493 | 2.0887 | 5.7141 |

Table 77: Error-type 1 for experiment PDUV_T_(3,1) for the SIDA dataset across all 50M models.

| MODEL | ID | | | | OOD | | | |
|---|---|---|---|---|---|---|---|---|
| | output channels | COM | KE | VP | output channels | COM | KE | VP |
| CNO | 0.0780 | 0.0125 | 0.2857 | 0.1107 | 0.5859 | 0.1692 | 0.5093 | 0.4618 |
| FNO | 0.0864 | 0.0089 | 0.1515 | 0.1579 | 0.4269 | 0.1612 | 3.0881 | 0.7062 |
| ScOT | 0.0577 | 0.0137 | 0.3477 | 0.0736 | 0.3840 | 0.0814 | 1.4028 | 0.2192 |
| UNet | 0.0628 | 0.0075 | 0.1949 | 0.0766 | 0.5529 | 0.1964 | 2.2313 | 0.7016 |
| ViT | 0.0928 | 0.0103 | 0.2557 | 0.1125 | 0.4577 | 0.0799 | 1.0613 | 2.0527 |

Table 78: Error-type 1 for experiment PDUV[VoS]_F_(1,1) for the SIDA dataset across all 1M models.

| MODEL | ID | | | | OOD | | | |
|---|---|---|---|---|---|---|---|---|
| | output channels | COM | KE | VP | output channels | COM | KE | VP |
| CNO | 0.7918 | 0.0992 | 0.6924 | 0.6835 | 0.9571 | 0.3585 | 0.7551 | 0.8626 |
| FNO | 0.9003 | 0.1372 | 0.7275 | 0.8660 | 1.0429 | 0.4379 | 0.9090 | 0.9489 |
| ResNet | 0.7620 | 0.1160 | 0.7873 | 0.6619 | 0.9709 | 0.4164 | 0.7139 | 0.8288 |
| ScOT | 0.8163 | 0.0923 | 1.2445 | 0.7049 | 0.9180 | 0.3643 | 0.6217 | 0.7423 |
| Transolver | 0.9367 | 0.2609 | 5.0196 | 1.0873 | 1.0140 | 0.2702 | 0.8002 | 0.7441 |
| UNet | 0.7622 | 0.1353 | 0.6859 | 0.7230 | 0.9877 | 0.4363 | 0.8919 | 0.8820 |
| ViT | 0.7847 | 0.0958 | 1.4725 | 6.4234 | 0.9272 | 0.3799 | 0.5464 | 1.6108 |

Table 79: Error-type 1 for experiment PDUV[VoS]_F_(1,1) for the SIDA dataset across all 50M models.

| MODEL | ID | | | | OOD | | | |
|---|---|---|---|---|---|---|---|---|
| | output channels | COM | KE | VP | output channels | COM | KE | VP |
| CNO | 0.7827 | 0.1344 | 0.7758 | 0.7839 | 0.9983 | 0.4357 | 0.9271 | 0.9184 |
| FNO | 0.8233 | 0.1310 | 0.7085 | 0.7276 | 1.0158 | 0.4319 | 0.9013 | 0.8486 |
| ScOT | 0.4879 | 0.0623 | 0.6491 | 0.3016 | 0.7900 | 0.3329 | 0.4133 | 0.6805 |
| UNet | 0.7603 | 0.1319 | 0.6798 | 0.7842 | 0.9836 | 0.4330 | 0.8903 | 0.9261 |
| ViT | 0.7295 | 0.1276 | 0.6112 | 0.7436 | 0.9490 | 0.4290 | 0.7976 | 0.7233 |

Table 80: Error-type 1 for experiment PDUV[VoS]_F_(3,1) for the SIDA dataset across all 1M models.

| MODEL | ID | | | | OOD | | | |
|---|---|---|---|---|---|---|---|---|
| | output channels | COM | KE | VP | output channels | COM | KE | VP |
| CNO | 0.3491 | 0.0248 | 0.9291 | 0.2643 | 0.4978 | 0.1225 | 6.7419 | 0.5641 |
| FNO | 0.4655 | 0.0601 | 1.8175 | 0.4452 | 0.6241 | 0.2448 | 1.1234 | 0.4679 |
| ResNet | 0.3957 | 0.1040 | 0.4972 | 0.2219 | 0.6470 | 0.2972 | 0.4464 | 0.3036 |
| ScOT | 0.5083 | 0.0586 | 1.1911 | 0.4072 | 0.6315 | 0.2591 | 1.4353 | 0.2242 |
| Transolver | 0.9821 | 0.6493 | 10.1858 | 2.2680 | 0.9961 | 0.3362 | 6.9883 | 1.6341 |
| UNet | 0.4399 | 0.1227 | 0.3030 | 0.1796 | 0.6513 | 0.2596 | 0.4430 | 0.3453 |
| ViT | 0.5564 | 0.0612 | 0.8397 | 3.2743 | 0.7827 | 0.3364 | 5.5856 | 1.5554 |

Table 81: Error-type 1 for experiment PDUV[VoS]_F_(3,1) for the SIDA dataset across all 50M models.

| MODEL | ID | | | | OOD | | | |
|---|---|---|---|---|---|---|---|---|
| | output channels | COM | KE | VP | output channels | COM | KE | VP |
| CNO | 0.3364 | 0.0352 | 1.2159 | 0.2618 | 0.5647 | 0.2030 | 1.1491 | 0.4548 |
| FNO | 0.4698 | 0.0548 | 2.0420 | 0.3363 | 0.6415 | 0.2331 | 2.4405 | 0.3442 |
| ScOT | 0.4151 | 0.0514 | 1.6202 | 0.3146 | 0.5552 | 0.2223 | 1.0314 | 0.1890 |
| UNet | 0.4064 | 0.1086 | 0.2991 | 0.1905 | 0.6389 | 0.2908 | 0.3951 | 0.5494 |
| ViT | 0.4062 | 0.0389 | 0.2410 | 1.5262 | 0.6021 | 0.2244 | 0.4503 | 0.9361 |

Table 82: Error-type 1 for experiment PDUV[VoS]_T_(3,1) for the SIDA dataset across all 1M models.

| MODEL | ID | | | | OOD | | | |
|---|---|---|---|---|---|---|---|---|
| | output channels | COM | KE | VP | output channels | COM | KE | VP |
| CNO | 0.3917 | 0.0611 | 1.1840 | 0.3466 | 64.7e4 | 0.2348 | inf | inf |
| FNO | 0.4649 | 0.0819 | 1.7223 | 0.4595 | 0.6151 | 0.3043 | 1.3698 | 0.7637 |
| ResNet | 0.4587 | 0.0798 | 0.9467 | 0.5002 | 0.7801 | 0.2378 | 9.0898 | 0.4642 |
| ScOT | 0.3974 | 0.0290 | 1.7939 | 0.1907 | 0.5321 | 0.1693 | 1.1423 | 0.1948 |
| Transolver | 0.8631 | 0.5943 | 16.2865 | 7.1293 | 0.8508 | 0.3161 | 29.8996 | 2.7680 |
| UNet | 0.4721 | 0.0715 | 0.6354 | 0.4156 | 0.6236 | 0.2490 | 3.3039 | 0.4502 |
| ViT | 0.5181 | 0.0432 | 1.6658 | 3.9481 | 0.6729 | 0.2653 | 1.5476 | 2.4982 |

Table 83: Error-type 1 for experiment PDUV[VoS]_T_(3,1) for the SIDA dataset across all 50M models.

| MODEL | ID | | | | OOD | | | |
|---|---|---|---|---|---|---|---|---|
| | output channels | COM | KE | VP | output channels | COM | KE | VP |
| CNO | 0.3125 | 0.0498 | 0.6993 | 0.1445 | 0.6209 | 0.2689 | 3.0228 | 0.7119 |
| FNO | 0.4128 | 0.0980 | 2.5765 | 0.4931 | 0.5936 | 0.2335 | 1.5842 | 0.8104 |
| ScOT | 0.4188 | 0.0556 | 1.9811 | 0.1534 | 0.4735 | 0.2147 | 6.3644 | 0.6552 |
| UNet | 0.3896 | 0.0904 | 0.6386 | 0.1623 | 0.5941 | 0.2198 | 5.5796 | 0.6366 |
| ViT | 0.3560 | 0.0232 | 0.4724 | 1.1887 | 0.5107 | 0.1515 | 1.6572 | 2.1826 |

Table 84: Error-type 1 for experiment PDUV_F_(3,2) for the SIDA dataset across all 1M models.

| MODEL | ID | | | | OOD | | | |
|---|---|---|---|---|---|---|---|---|
| | output channels | COM | KE | VP | output channels | COM | KE | VP |
| CNO | 0.1575 | 0.0388 | 1.4388 | 0.4546 | 0.5050 | 0.1514 | 5.1212 | 0.6247 |
| FNO | 0.1338 | 0.0107 | 0.2739 | 0.3349 | 0.5554 | 0.1943 | 0.3353 | 0.5985 |
| ResNet | 0.1472 | 0.0233 | 0.2932 | 0.1528 | 0.5761 | 0.1077 | 1.1428 | 0.5209 |
| ScOT | 0.1532 | 0.0393 | 0.9001 | 0.1925 | 0.4968 | 0.2138 | 1.7578 | 0.2268 |
| Transolver | 0.9577 | 0.6721 | 28.9895 | 1.3517 | 0.9609 | 0.3576 | 13.6453 | 0.3941 |
| UNet | 0.0955 | 0.0120 | 0.3596 | 0.1350 | 0.5663 | 0.1878 | 1.8528 | 0.5861 |
| ViT | 0.2780 | 0.0439 | 1.0290 | 0.6201 | 0.5849 | 0.1939 | 1.0244 | 0.6553 |

Table 85: Error-type 1 for experiment PDUV_F_(3,2) for the SIDA dataset across all 50M models.

| MODEL | ID | | | | OOD | | | |
|---|---|---|---|---|---|---|---|---|
| | output channels | COM | KE | VP | output channels | COM | KE | VP |
| CNO | 0.0513 | 0.0070 | 0.2164 | 0.1403 | 0.4681 | 0.1316 | 0.2591 | 0.5476 |
| FNO | 0.0737 | 0.0063 | 0.1264 | 0.1556 | 0.5289 | 0.1629 | 0.5149 | 0.3155 |
| ScOT | 0.0571 | 0.0113 | 0.3775 | 0.0879 | 0.4410 | 0.1673 | 0.9820 | 0.2451 |
| UNet | 0.0463 | 0.0054 | 0.1301 | 0.0561 | 0.4909 | 0.1693 | 0.3850 | 0.5484 |
| ViT | 0.1081 | 0.0172 | 0.4534 | 0.1510 | 0.4738 | 0.0835 | 0.8493 | 1.2346 |

Table 86: Error-type 1 for experiment PDUV[VoS]_F_(3,2) for the SIDA dataset across all 1M models.

| MODEL | ID | | | | OOD | | | |
|---|---|---|---|---|---|---|---|---|
| | output channels | COM | KE | VP | output channels | COM | KE | VP |
| CNO | 0.3466 | 0.0682 | 1.6896 | 0.3208 | 0.4629 | 0.1822 | 2.9526 | 0.3537 |
| FNO | 0.4198 | 0.0780 | 1.5123 | 0.5962 | 0.5853 | 0.2031 | 1.1691 | 0.5967 |
| ResNet | 0.3530 | 0.0344 | 0.9827 | 0.1911 | 0.6241 | 0.1701 | 1.7610 | 0.4161 |
| ScOT | 0.3796 | 0.0512 | 0.7868 | 0.1669 | 0.5444 | 0.1718 | 1.1167 | 0.3058 |
| Transolver | 0.9507 | 0.6911 | 18.3170 | 1.1671 | 0.9841 | 0.3493 | 12.6450 | 0.5056 |
| UNet | 0.3325 | 0.0638 | 0.8743 | 0.2758 | 0.6513 | 0.2507 | 3.5701 | 0.3652 |
| ViT | 0.4556 | 0.0487 | 1.2767 | 1.7015 | 0.6301 | 0.2489 | 3.8598 | 0.9421 |

Table 87: Error-type 1 for experiment PDUV[VoS]_F_(3,2) for the SIDA dataset across all 50M models.

| MODEL | ID | | | | OOD | | | |
|---|---|---|---|---|---|---|---|---|
| | output channels | COM | KE | VP | output channels | COM | KE | VP |
| CNO | 0.2647 | 0.0412 | 1.8487 | 0.3474 | 0.5649 | 0.1785 | 1.2554 | 0.4279 |
| FNO | 0.4295 | 0.0724 | 1.3556 | 0.2476 | 0.6063 | 0.2797 | 1.6246 | 0.2670 |
| ScOT | 0.3694 | 0.0557 | 0.6053 | 0.2316 | 0.5262 | 0.2207 | 0.6769 | 0.2067 |
| UNet | 0.3423 | 0.0871 | 0.3185 | 0.2050 | 0.5732 | 0.3279 | 0.7159 | 0.5444 |
| ViT | 0.3239 | 0.0240 | 0.4701 | 0.4947 | 0.5605 | 0.1965 | 1.2653 | 0.5427 |

# F  INFERNCE ROLLOUT PLOTS

## F.1  ROLLOUT PREDICTIONS FROM INITIAL CONDITIONS FOR THE LIDE IN-DISTRIBUTION (ID) DATASET

In the following, we present rollout predictions for various models—each with 50M parameters, except for ResNet and Transolver, which have only 1M parameter count. The trajectories are shown in the Figures 41, 42, 43, 44, 45, 46, and 47. It corresponds to the following simulation parameters: filament pressure $9.3886 \times 10^9$ [Pa], ambient pressure $1.0382 \times 10^5$ [Pa], laser half-width $1.1727 \times 10^{-6}$ [m], and droplet radii $1.5966 \times 10^{-5}$ [m] and $1.2139 \times 10^{-5}$ [m] along z- and r-axis, respectively. In all figures, the time frames are presented in order from top to bottom.

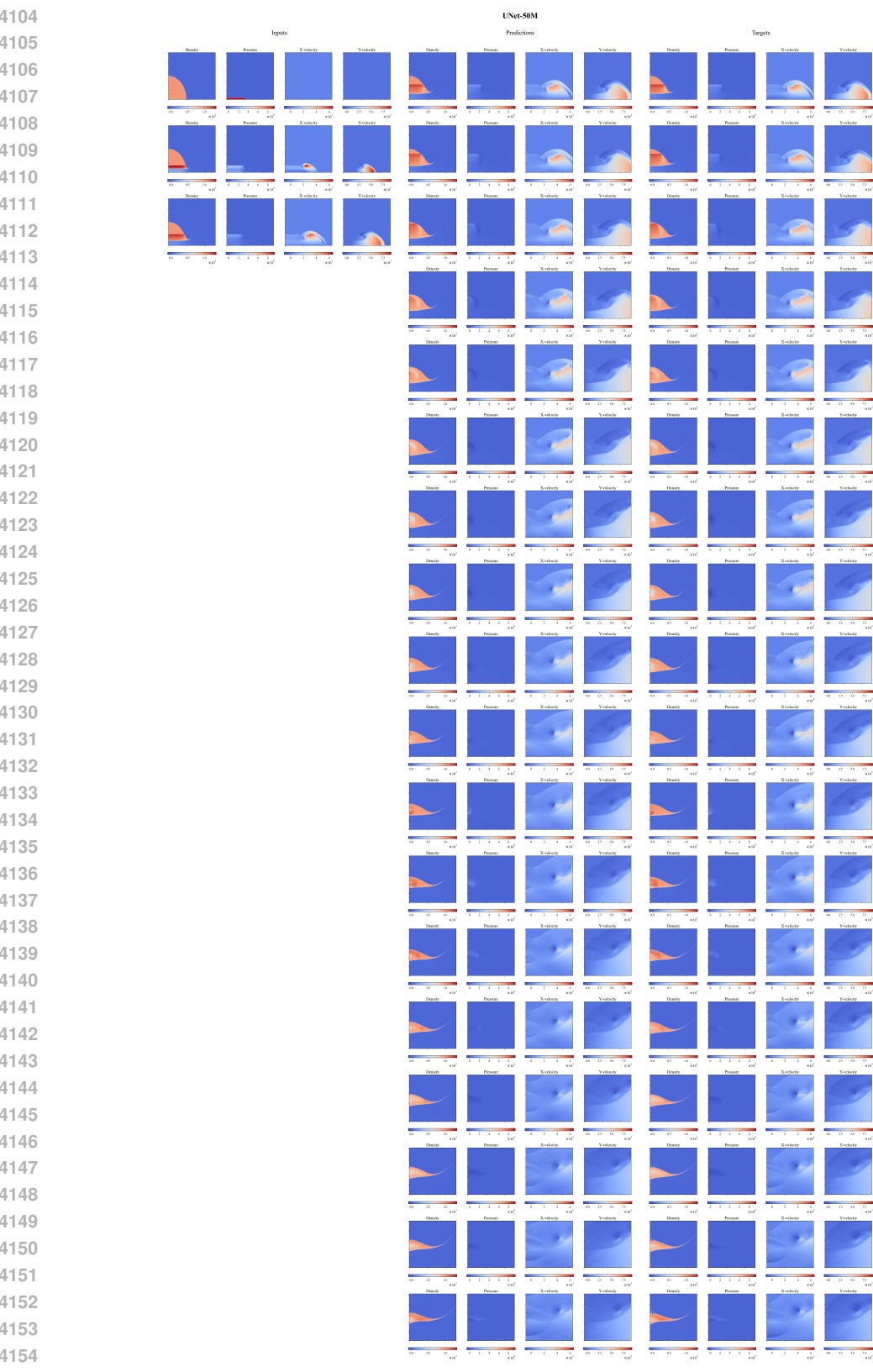

Figure 41: Rollout predictions for the LIDE-ID-Experiment PDUV_F_(3,1) with UNet-50M.

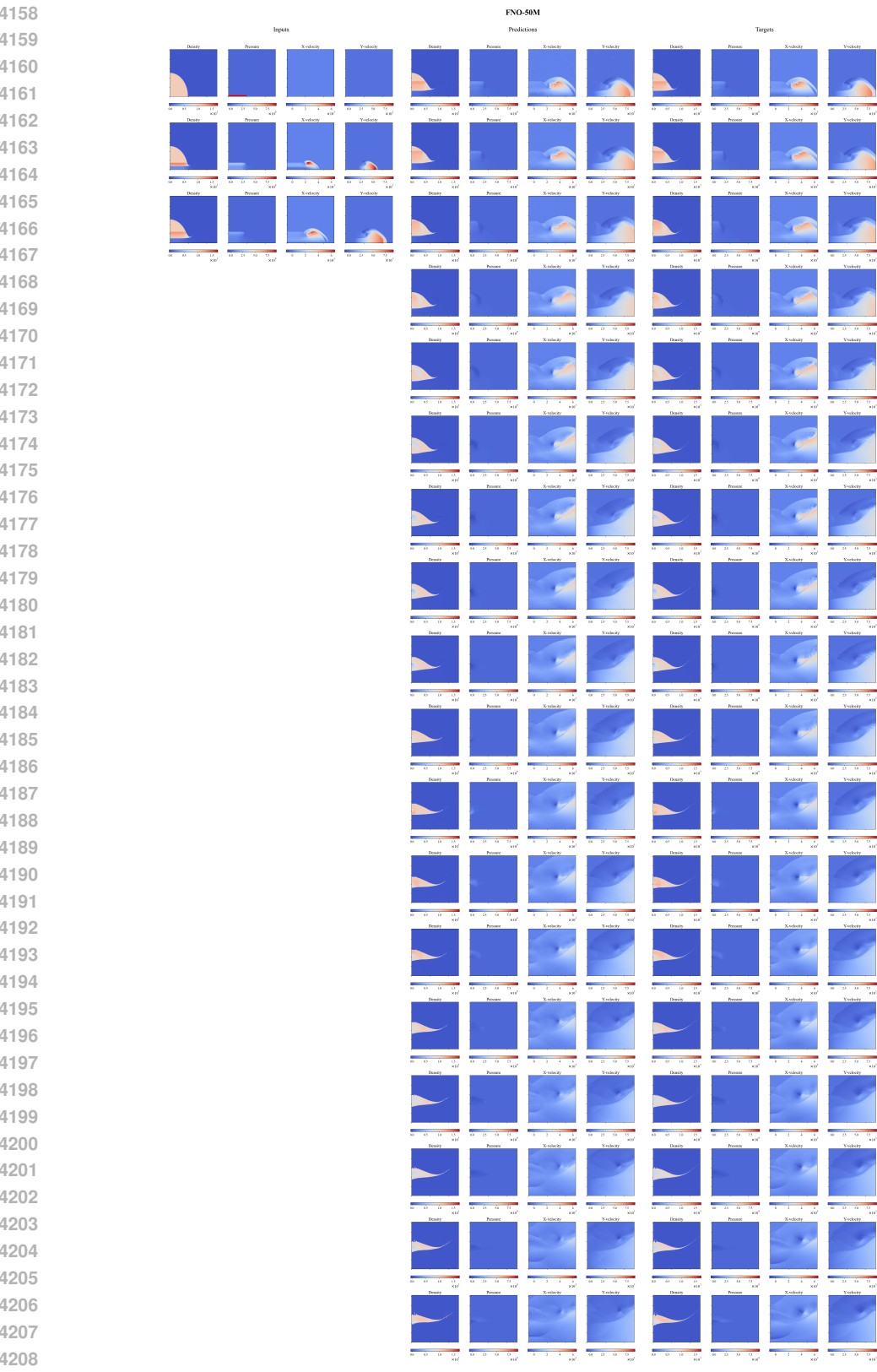

Figure 42: Rollout predictions for the LIDE-ID-Experiment PDUV_F_(3,1) with FNO-50M.

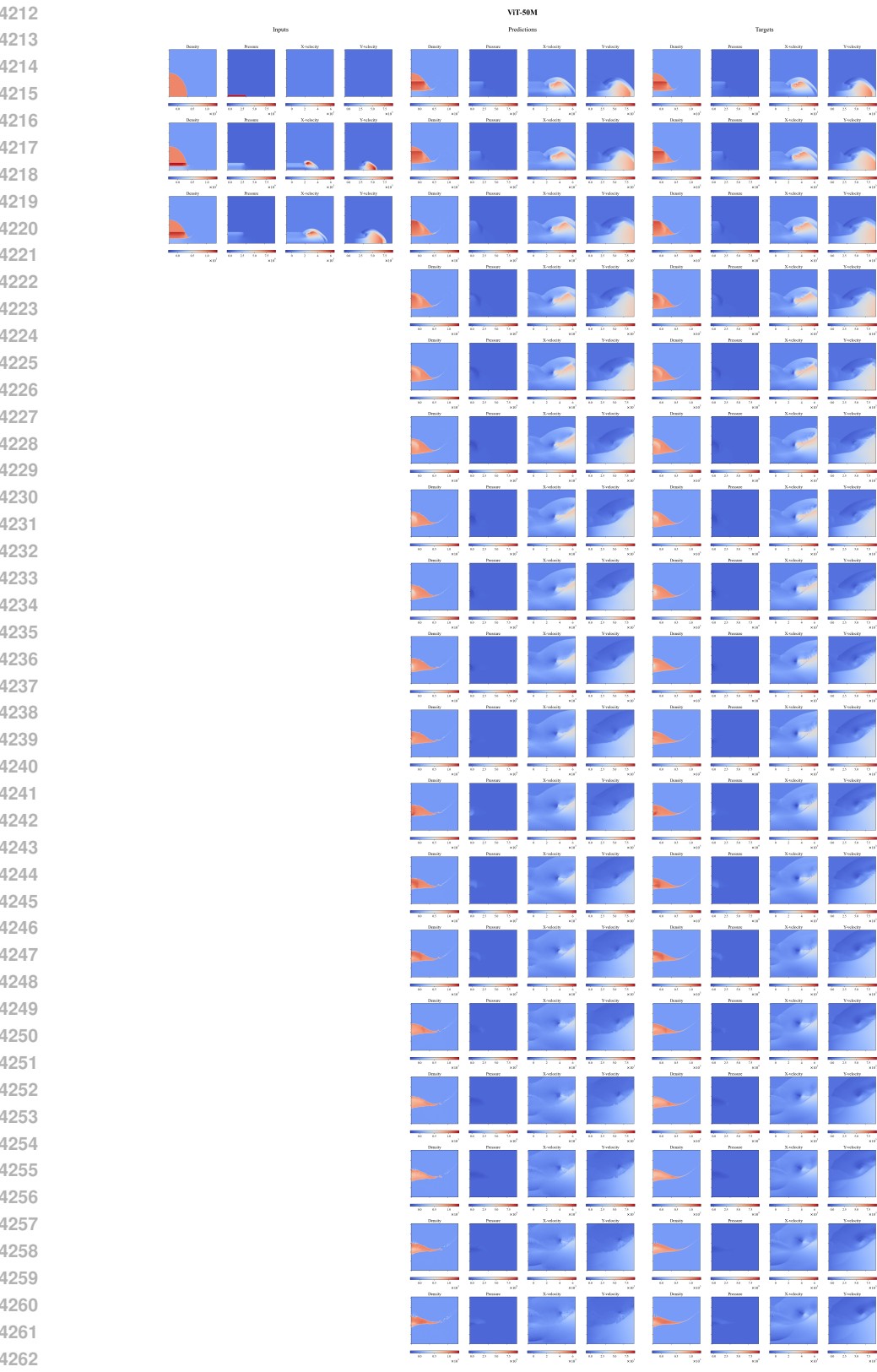

Figure 43: Rollout predictions for the LIDE-ID-Experiment PDUV_F_(3,1) with ViT-50M.

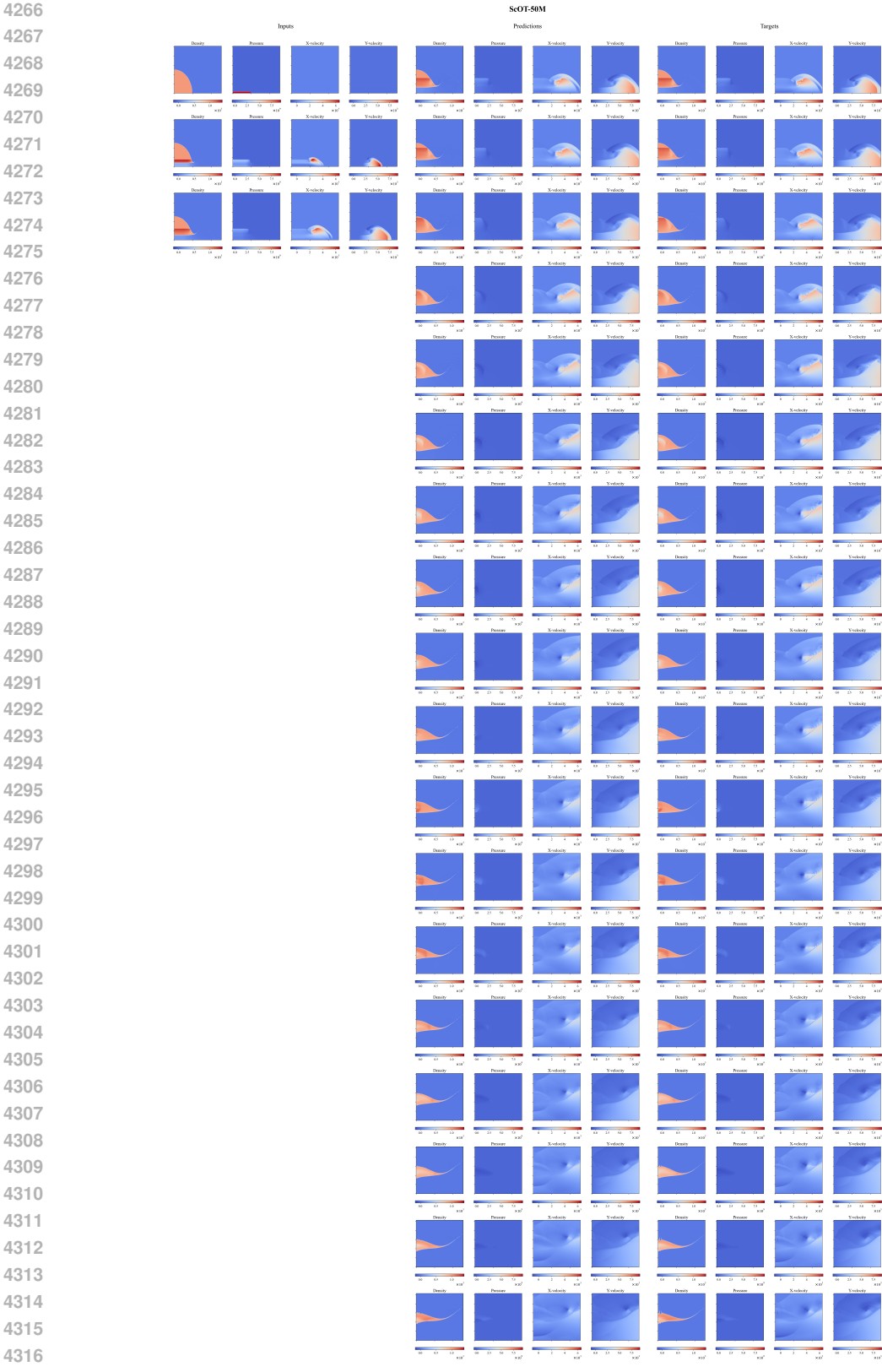

Figure 44: Rollout predictions for the LIDE-ID-Experiment PDUV_F_(3,1) with ScOT-50M.

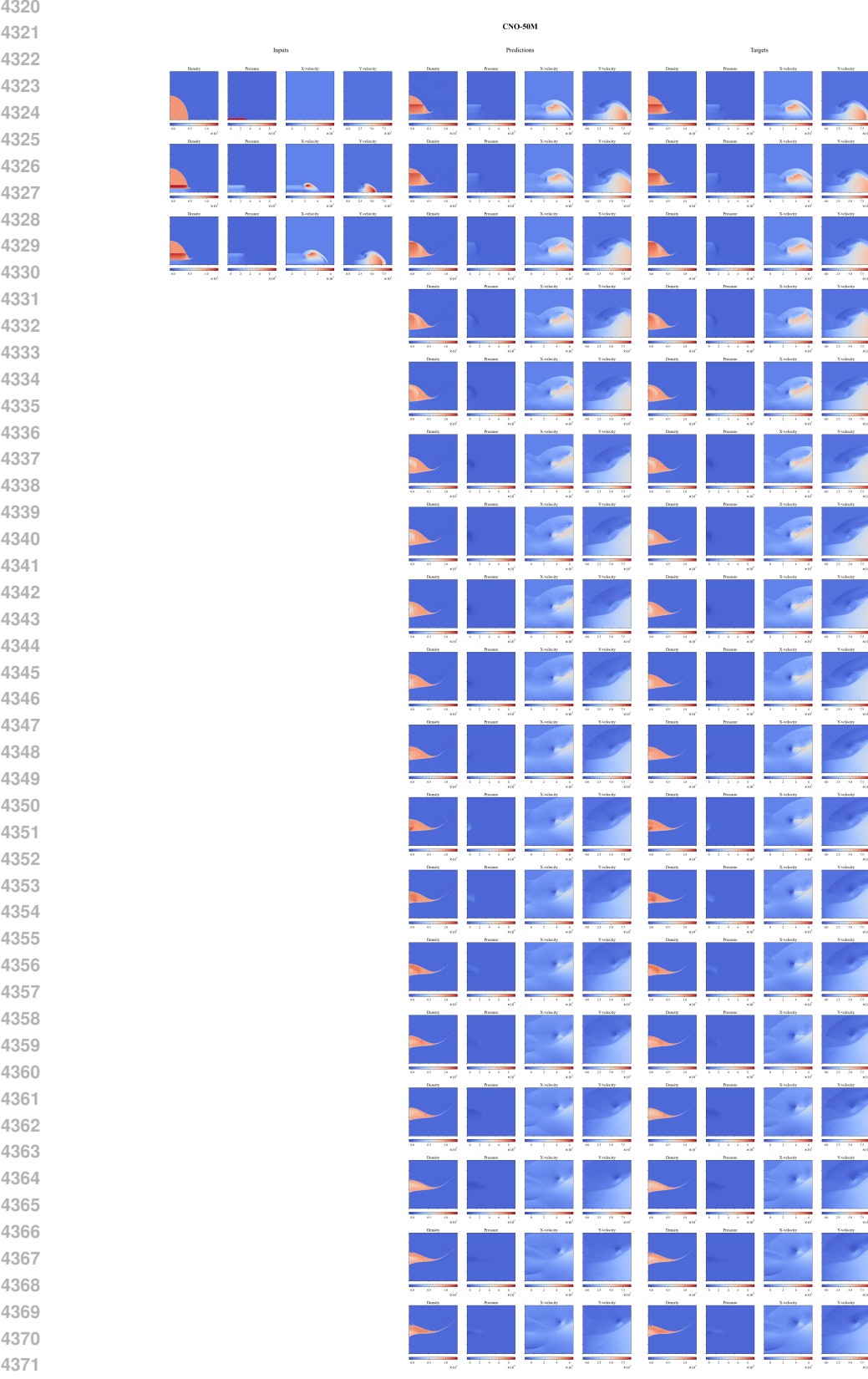

Figure 45: Rollout predictions for the LIDE-ID-Experiment PDUV_F_(3,1) with CNO-50M.

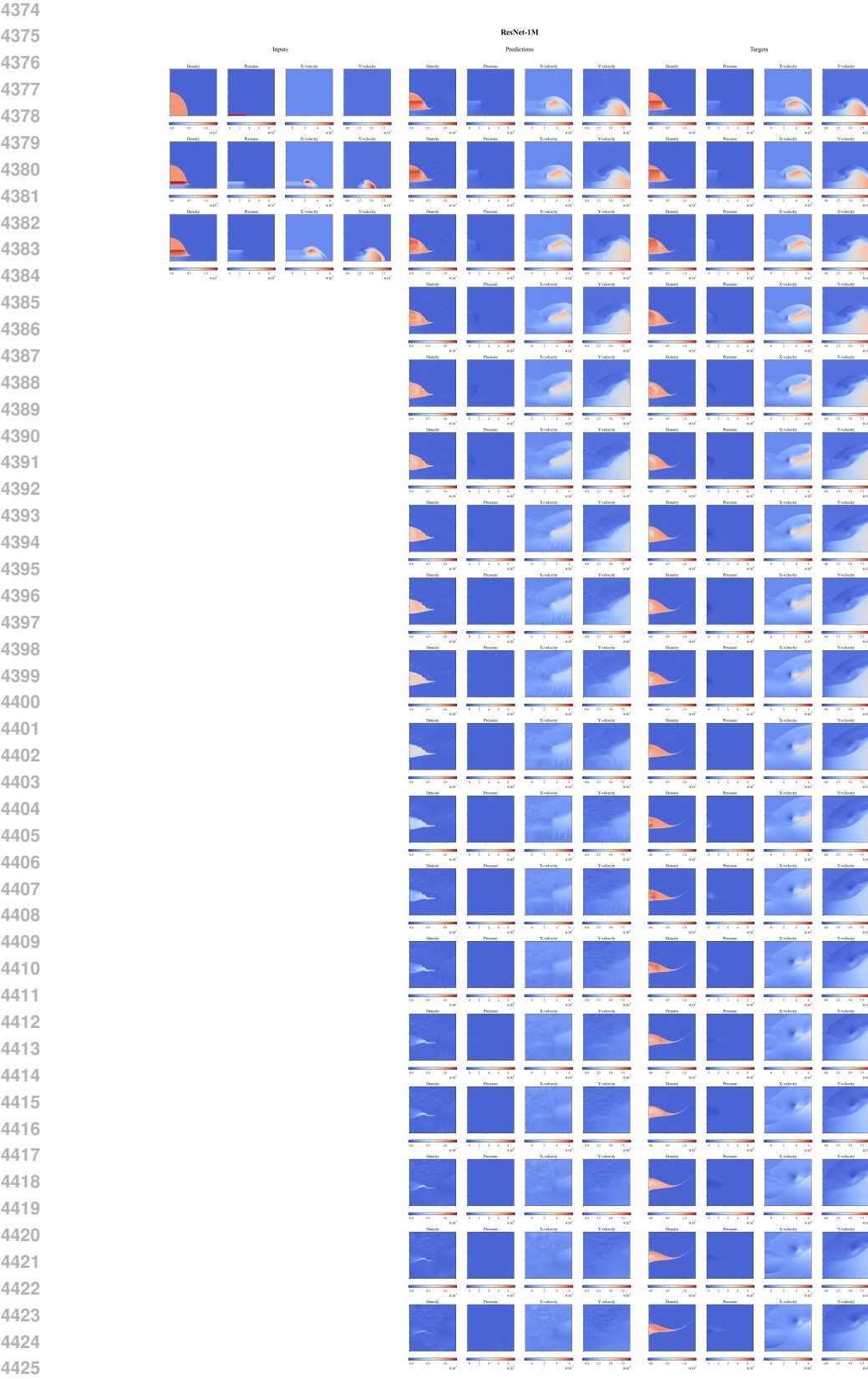

Figure 46: Rollout predictions for the LIDE-ID-Experiment PDUV_F_(3,1) with ResNet-1M.

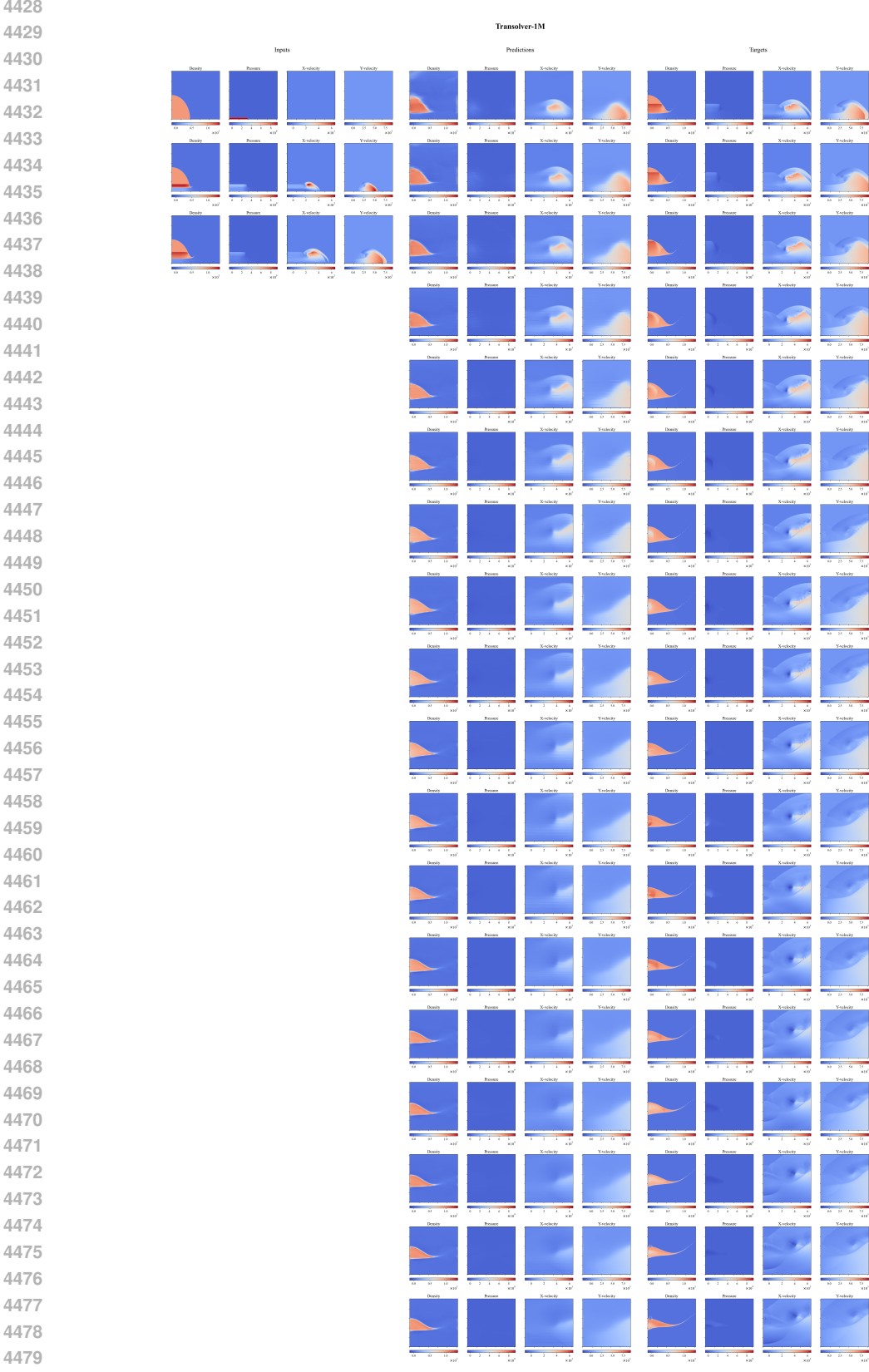

Figure 47: Rollout predictions for the LIDE-ID-Experiment PDUV_F_(3,1) with Transolver-1M.

## F.2    ROLLOUT PREDICTIONS FROM INITIAL CONDITIONS FOR THE SIDA IN-DISTRIBUTION (ID) DATASET

Here, we present rollout predictions for various models—each with 50M parameters, except for ResNet, which has only 1M parameter count. The trajectory shown in the Figures 48, 49, 50, 51, 52, and 53 corresponds to the following simulation parameters: The shock Mach number 3.26, the flow Mach number 1.42, and the Weber number 13820. In all figures, the time frames are presented in order from top to bottom.

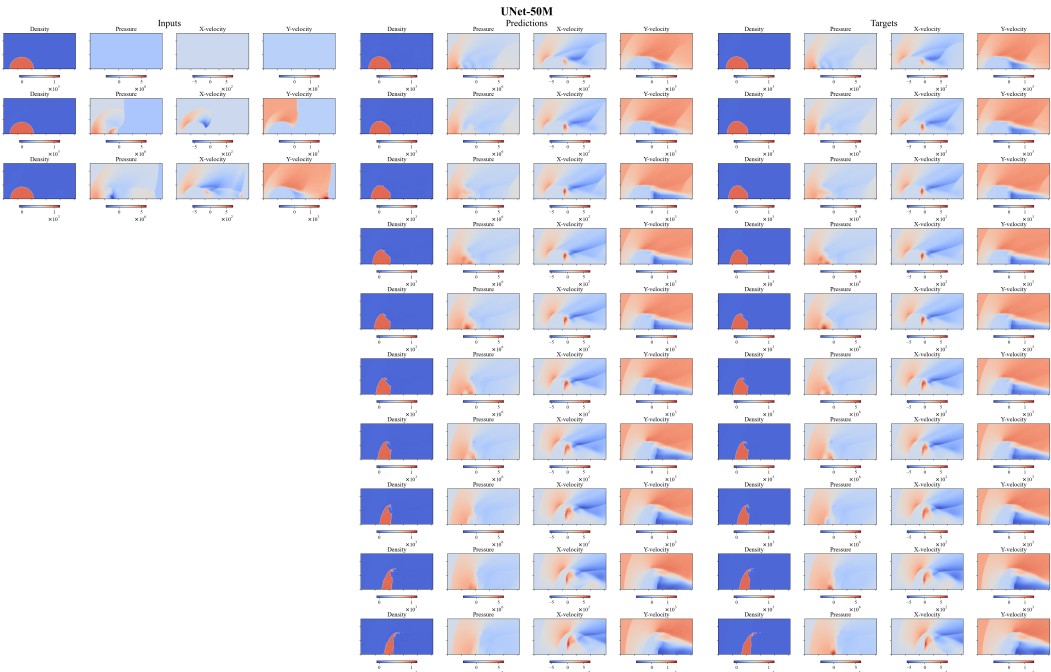

Figure 48: Rollout predictions for the SIDA-ID-Experiment PDUV_F_(3,1) with UNet-50M.

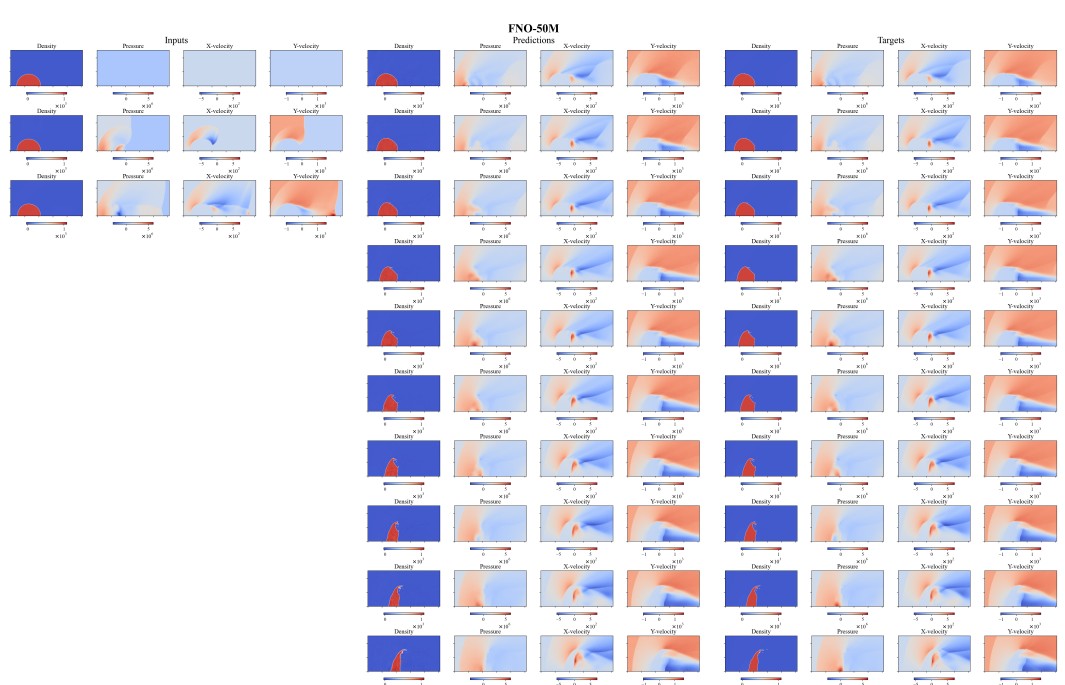

Figure 49: Rollout predictions for the SIDA-ID-Experiment PDUV_F_(3,1) with FNO-50M.

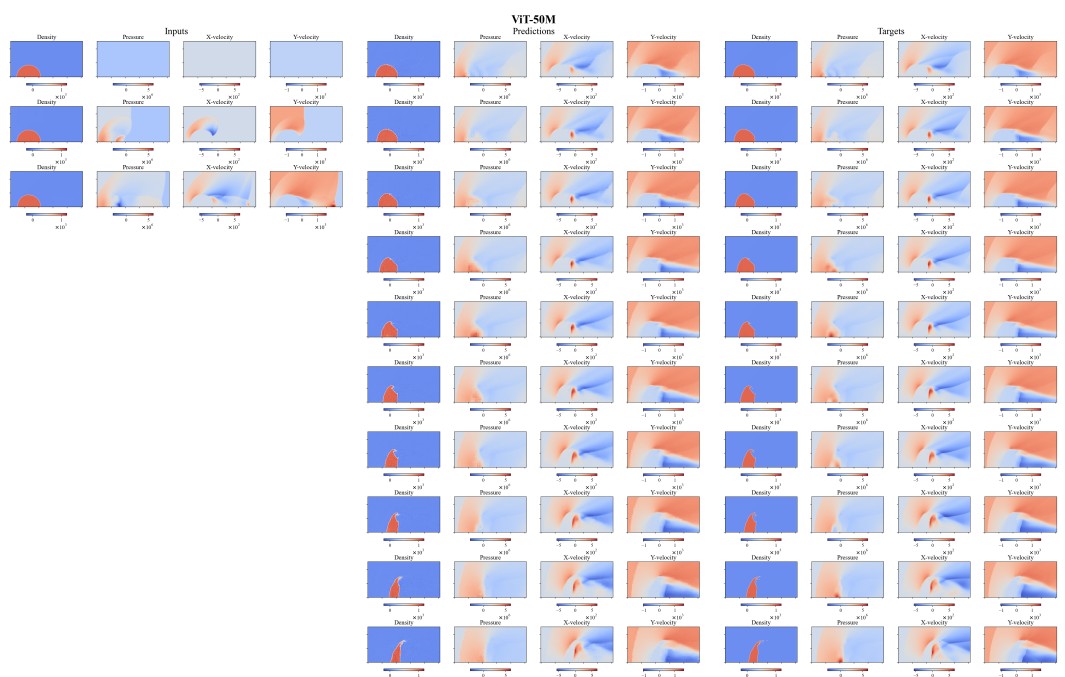

Figure 50: Rollout predictions for the SIDA-ID-Experiment PDUV_F_(3,1) with ViT-50M.

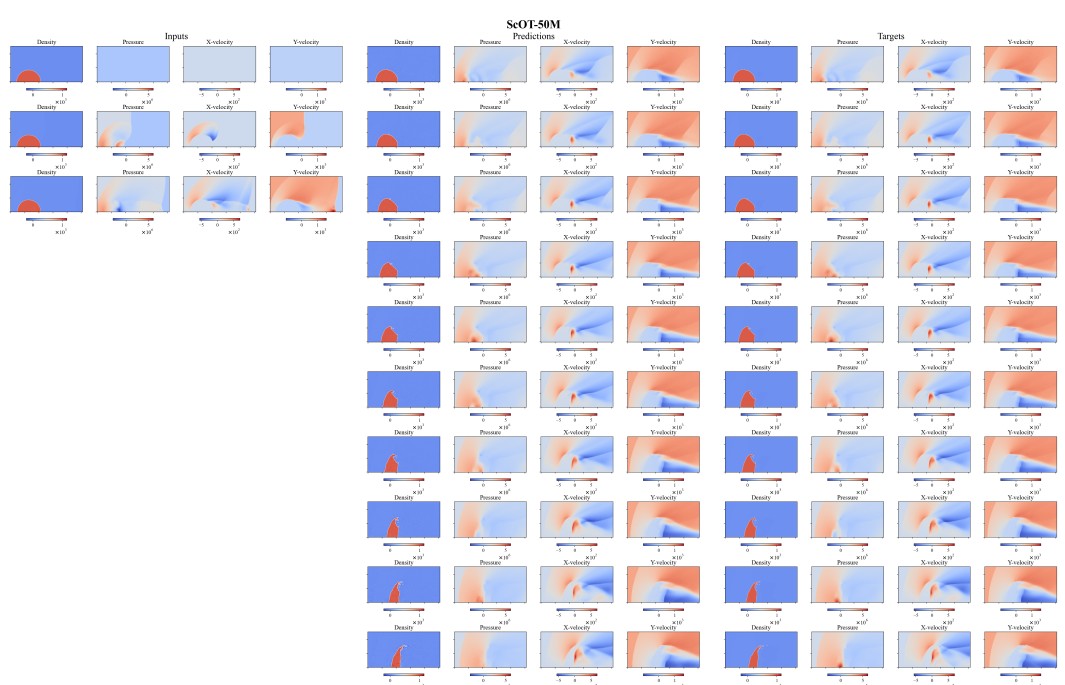

Figure 51: Rollout predictions for the SIDA-ID-Experiment PDUV_F_(3,1) with ScOT-50M.

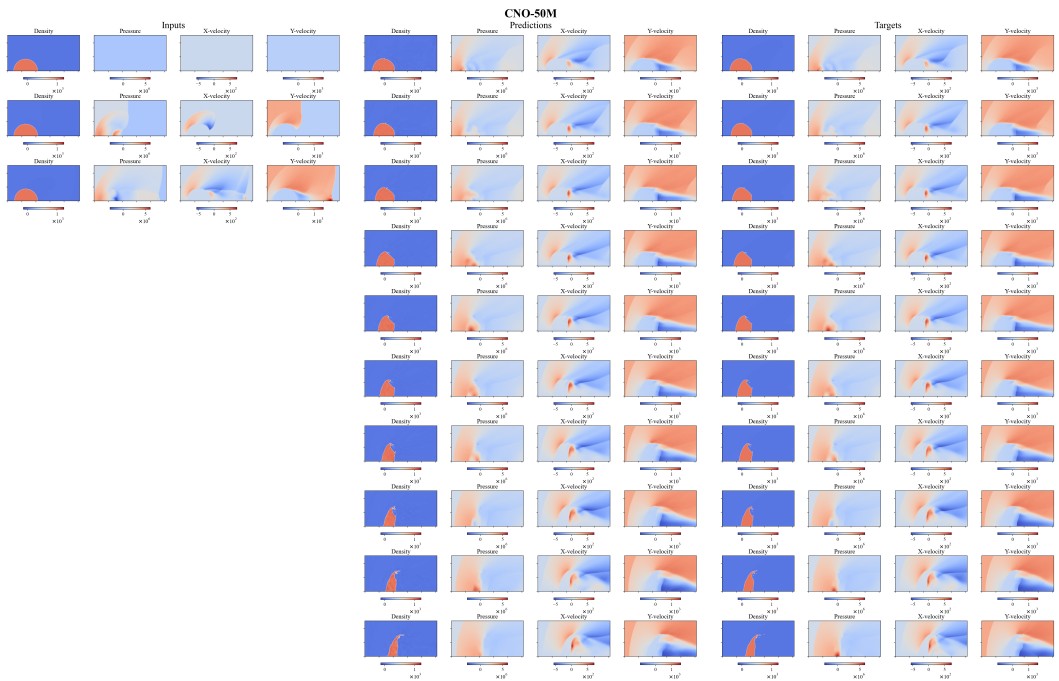

Figure 52: Rollout predictions for the SIDA-ID-Experiment PDUV_F_(3,1) with CNO-50M.

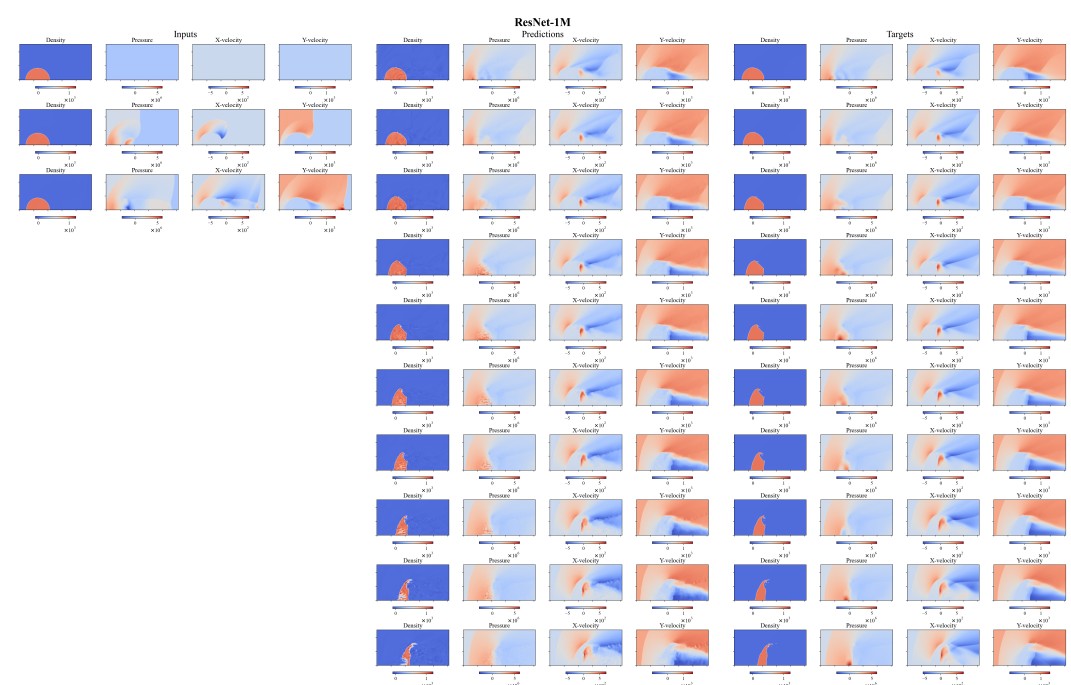

Figure 53: Rollout predictions for the SIDA-ID-Experiment PDUV_F_(3,1) with ResNet-1M.

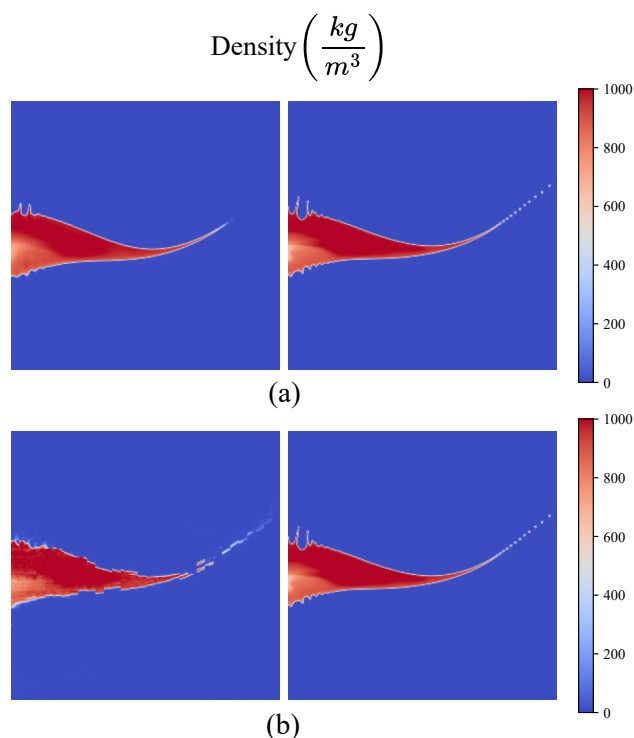

Figure 54: Comparison between UNet 50-M (a) and ViT 50-M (b) with target (for both at right) at the last rollout step for experiment PDUV_T_(3,1).

## G  LARGE LANGUAGE MODEL (LLM) USAGE

Large Language Models (LLMs) were utilized to polish the writing and find suitable words in some scenarios.

