# OpenReview forum: "Laser- and shock-induced droplet dynamics: A machine learning benchmark for complex multiphase flows"
_ICLR.cc/2026/Conference — Submitted to ICLR 2026_

### Official Review · Reviewer_6xVr · 2025-10-27

**Soundness:** 2
**Presentation:** 2
**Contribution:** 1
**Rating:** 2
**Confidence:** 5

**Summary:**

The author proposed a machine learning benchmark for multiphase flow data. Although two high-quality data were provided, the reviewer believes that the paper is not yet ready for publication due to the amount of data, the methods presented, and the interpretation and analysis of the results.

**Strengths:**

1. Two new datasets have been provided for use by the AI community.
2. Provided results of different methods on data.

**Weaknesses:**

1. The relevant work is not comprehensive, and it seems that the author's research on PDE benchmarks is insufficient. There are still many works such as FlowBench, The Well, BLASTNet, etc. that should be discussed in the relevant work.
2. The lack of 3D data in the benchmark is a drawback.
3. Can valuable insights be derived from the results, such as which types of methods are more suitable for which features of the data.
4. The biggest concern is that there are too few methods used and they are not SOTA based. There are many other methods such as CNO, Translolver, Diffusion model, AROMA, etc. that should be included in the benchmark.
5. Other benchmarks such as PDEBench and The Well provide a large amount of data, but this work only has two datasets and is not suitable for publication as a benchmark.
6. Lack of experiments on PDE foundation models such as GNOT, DPOT, etc.
7. An important evaluation criterion for benchmarks is whether the code interface is easy to use, but the author does not have open source code, so it cannot be evaluated.

**Questions:**

See the weaknesses.

---

> ### Author Response · Authors · 2025-11-16
> **Open-sourcing the Benchmark Code and the Datasets (Weakness 7)**
>
> Thank you for the review. We will provide complete responses to all questions, and we start here by addressing the weakness number 7. The following links correspond to the benchmarking source code and the datasets:
> * GitHub repository (https://anonymous.4open.science/r/complex_fluid_bench-FC28)
> * Datasets (https://huggingface.co/FluidVerse)
>
> Detailed instructions on using these repositories are included in the README.md. Code and datasets will be made publicly available upon acceptance.

---

> ### Author Response · Authors · 2025-11-21
> **Extension to 3D dataset (Weakness 2)**
>
> We address the potential 3D extension by employing a 2D axisymmetric simulation setup. This formulation captures the essential 3D physical effects while maintaining a significantly lower memory footprint compared to full 3D simulations. As a result, the generated ground-truth data effectively reflects 3D behavior within a tractable computational budget.

---

> ### Author Response · Authors · 2025-11-21
> **Large datasets and foundation models (Weaknesses 5 and 6)**
>
> **Other benchmarks, such as PDEBench and The Well, provide a large amount of data, but this work only has two datasets and is not suitable for publication as a benchmark.**: We agree that large datasets are fundamental for the development of Foundation Models (FMs) in science. However, our paper is framed as a benchmark/reference focused on establishing robust baseline models for compressible multiphase datasets. The primary utility of the models evaluated here lies in their ability to make surrogate predictions for computational fluid dynamics (CFD) datasets, which are a novel addition to the existing pool of publicly available datasets.
>
>  We have generated an additional 25GB of novel OOD trajectories, and we are currently running inference on these OOD datasets and will include this dataset and the analysis of the baseline models' performance in our revised PDF. In total, our datasets' contribution amounts to 115GB and will be hosted permanently on HuggingFace.
>
> **Lack of experiments on PDE foundation models such as GNOT, DPOT, etc.**: We promise to investigate pre-trained FMs for the camera-ready version.

---

> ### Author Response · Authors · 2025-11-21
> **Additions suggested by the reviewer (Weaknesses 1 and 4)**
>
> - **The relevant work is not comprehensive, and it seems that the author's research on PDE benchmarks is insufficient. There are still many works, such as FlowBench, The Well, BLASTNet, etc., that should be discussed in the relevant work.**:
> We thank the reviewer for highlighting the missing work, and we have added references to the relevant work in our revised PDF.
>
> - **The biggest concern is that there are too few methods used and they are not SOTA based. There are many other methods such as CNO, Translolver, Diffusion model, AROMA, etc. that should be included in the benchmark.**:
> We trained CNO 1M, 50M, and Transolver 1M as suggested by the reviewer and performed inference on ID and OOD datasets using these trained models. Our Primary reason for including robust baseline models was to assess the performance on the novel complex datasets that we have generated. To date, these models have primarily been evaluated on simplified CFD datasets; this work is the first to investigate their behavior in compressible and multiphase flow regimes.

---

> > ### Comment · Reviewer_6xVr · 2025-11-26
> >
> > I did not see any experiments on the author's modifications and additions to the paper. I will maintain the rating as I believe the current status is not yet ready for publication, and the author needs to further revise the paper.
> >
> > Furthermore, I must point out that the author's name may appear in the links of the dataset.

---

> > > ### Author Response · Authors · 2025-12-01
> > >
> > > The reviewer's concerns have been addressed in the revised PDF, which we have uploaded, and the links have been anonymized.

---

> ### Author Response · Authors · 2025-12-01
> **The effiecient "methods" for data features (Weakness 3)**
>
> We have different interpretations by **method** for capturing features of the data shown by domain-specific quantities of Interest.
> * In case Method means Model: We concluded that in most of the cases UNet-50M is outperforming the other models.
> * In case Method means Experiment Scenario: we highlighted the scenario results in each corresponding section; for example, for scenario 1, including more historical time-steps as inputs, has slight metric advantage over single-input case.
> * In case Method means informing models of some physical constraints during training, we did not perform Physics-Informed training in this Benchmark paper.

---

### Official Review · Reviewer_F5Kz · 2025-10-28

**Soundness:** 3
**Presentation:** 3
**Contribution:** 3
**Rating:** 6
**Confidence:** 4

**Summary:**

This paper presents new field evolution datasets related to complex multiphase flow phenomena that could serve as a real-world benchmark for the scientific machine learning (SciML) community. Two specific physics phenomena are considered, namely laser-induced droplet explosion and shock-induced droplet aero-breakup. The paper also establishes a baseline by comparing the predictive performance of diverse architectures including U-Net, Fourier neural operator (FNO), vision transformer (ViT), scalable operator transformer (ScOT), and residual network (ResNet).

**Strengths:**

Compared to the usual benchmarks in SciML literature, such as the incompressible Navier-Stokes equation or Burgers' equation, these new benchmarks present unique challenges for SciML involving the prediction of sharp reaction fronts, discontinuity at the flow interfaces, sharp spatial and temporal gradients, and fast transient features. Such strong nonlinearity makes them a useful dataset to test the limits of SciML algorithms, which I believe is an important contribution to the research community.

**Weaknesses:**

I think the paper may benefit from considering the following suggestions:
- Provide more metric/criteria for evaluating prediction results. The current quantitative metrics such as RMSE provide an overall estimate of how "accurate" models are, but from the domain standpoint, those metrics may not fully capture the phenomena of interest. For example, for droplet atomization, we may not care too much about the exact physical values in the ambient space, as long as the rate of droplet breaking up or the size of droplets after being broken down. (And vice versa--even if a model is very accurate in predicting other physical values, I would argue that the model isn't still very useful if it cannot faithfully predict droplet displacement, deformation, etc.) That said, I think the community is going to benefit a lot by having those domain-specific performance metrics. These metrics don't need to be all quantitative--qualitative criteria in terms of what kind of features to look for would still be highly valuable.
- This is not terribly critical, but I noticed that the ML models that the authors compared are mostly (solution) operator-based formulations. It might be worthwhile to compare these methods against other methods like PINN, but I don't want to arm-twist the authors to create a laundry list of all bunch of random methods.

**Questions:**

- "Code and datasets will be made available upon request." Can they be made available upon acceptance? I do find this a valuable work but only when (at least) the datasets are made available to the community.

---

> ### Author Response · Authors · 2025-11-16
> **Open-sourcing the Benchmark Code and the Datasets (Question 1)**
>
> Thank you for the review. We will provide complete responses to all questions, and we start here by addressing the question. The following links correspond to the benchmarking source code and the datasets:
> * GitHub repository (https://anonymous.4open.science/r/complex_fluid_bench-FC28)
> * Datasets (https://huggingface.co/FluidVerse)
>
> Detailed instructions on using these repositories are included in the README.md. Code and datasets will be made publicly available upon acceptance.

---

> ### Author Response · Authors · 2025-11-26
> **Domain-specific evaluation metrics (Weakness 1)**
>
> Thank you for the insightful suggestion. This is indeed a very valid concern. In response, we have incorporated some domain-specific evaluation metrics to better capture the underlying physics of the problem. In our updated PDF, we now report four quantities of interest (QoI):
>
> * **Vorticity Production (VP)** over time for both datasets
> * **Kinetic Energy (KE)** evolution over time for both datasets
> * **Droplet outer diameter change** over time for Dataset LIDE
> * **Droplet center-of-mass displacement** over time for Dataset SIDA
>
> Eventually, we compare these QoI between the predictions and the targets using normalized-RMSE. These additions during inference provide a more comprehensive and physically meaningful evaluation of model performance. Please refer to Table 5 in the main text, Sections C and E, where we present the metrics for the ID and OOD test datasets in our revised PDF.
>
> According to the qualitative evaluation, we would suggest two dataset-specific visualization metrics for researchers:
> * For LIDE: The evolution of shock/expansion waves inside the droplet and the droplet deformation upon explosion. This phenomenon is extensively studied in the corresponding reference papers.
> * For SIDA: The breakup mode, the hat-shaped droplet deformation and flattening, and the shock wave dynamics.

---

> ### Author Response · Authors · 2025-12-01
> **Comparison against other methods such as PINNs (Weakness 2)**
>
> We thank the reviewer for suggesting that we introduce a different method, such as PINNs, and we also appreciate their understanding that such inclusions involve a detailed list of ablations. The goal of this project is to investigate the performance of various operator-based formulations and examine how these models address the effects of compressibility and multiphase flow, purely from data and conditioning. For our current submission, we provide baselines on time-tested convolutional and spectral architectures, such as FNO, UNet, ResNet, and ViT, as well as recent architectures, including ScOT, CNO, and Transolver. We completely agree that a comparison with a different method would provide new insights into these complex fluid flow predictions and is a future extension of this project.

---

### Official Review · Reviewer_gWms · 2025-10-30

**Soundness:** 3
**Presentation:** 2
**Contribution:** 2
**Rating:** 4
**Confidence:** 5

**Summary:**

This paper introduces two benchmark datasets—LIDE (Laser-Induced Droplet Explosion) and SIDA (Shock-Induced Droplet Aero-Breakup)—to advance research in compressible multiphase flows within the Scientific Machine Learning (SciML) community. The authors evaluate several surrogate models, including FNO, UNet, ViT, ScOT, and ResNet, to capture complex nonlinear shock–interface dynamics. The study effectively demonstrates the importance of temporal conditioning in improving predictive accuracy and stability.

**Strengths:**

1) Presents novel, high-fidelity datasets addressing important multiphase flow problems with strong physical and industrial relevance.

2) Includes comprehensive benchmarking across diverse neural architectures, supported by detailed ablation studies and insightful analysis.

**Weaknesses:**

1) The dataset and code are not yet publicly available, limiting reproducibility and immediate community adoption.

2) Evaluation could be broadened to include recent operator-learning models, such as Transformer-based and state-space (SSM) operators (e.g., Transolver, Mamba Operator).

3) Some implementation details—including hyperparameter settings and conditioning strategies—are not clearly described.

**Questions:**

1) Since the primary contribution is the dataset, would the authors consider open-sourcing the datasets and code to encourage best practices and broader use within the ML community?

2) How well do the surrogate models generalize to unseen physical regimes or parameter variations, and is zero-shot transfer feasible?

3) Could the datasets be extended to 3D cases or include additional physical phenomena such as phase change or evaporation?

4) How sensitive are the results to temporal resolution and conditioning strategies? Including standard deviations would strengthen the statistical credibility.

5) How do different boundary conditions (e.g., symmetry, Dirichlet, Neumann) affect performance? Can models handle zero-shot transfer across boundary types, and which operator performs best?

6) The distinction between Error 1 and Error 2 is somewhat unclear, as Tables 3 and 4 do not show a clear correlation. Could the authors include energy spectrum analyses and rollout error over time steps to better interpret model behavior and physical fidelity?

**Minor Comments:**

1) The statement “In addition, for reproducing model evaluations, we provide trained model weights and the code that has the complete set of instructions upon request.” — It would be highly beneficial to open-source the dataset and trained weights to enhance accessibility and reproducibility within the SciML community.

2) Please ensure that the supplementary material remains anonymous, as some identifiable information may still be present in the metadata.

---

> ### Author Response · Authors · 2025-11-16
> **Open-sourcing the Benchmark Code and the Datasets (Weakness 1 and Question 1)**
>
> Thank you for the review. We will provide complete responses to all questions, and we start here by addressing the first weakness and the question. The following links correspond to the benchmarking source code and the datasets:
> * GitHub repository (https://anonymous.4open.science/r/complex_fluid_bench-FC28)
> * Datasets (https://huggingface.co/FluidVerse)
>
> Detailed instructions on using these repositories are included in the README.md. Code and datasets will be made publicly available upon acceptance.

---

> ### Author Response · Authors · 2025-11-21
> **Generalization ability (Question 2)**
>
> Regarding generalization, we agree that these capabilities are fundamental for the development of Foundation Models (FMs) in science. However, our paper is framed as a benchmark/reference focused on establishing robust baseline models for compressible multiphase datasets. While high generalization is always desirable, the primary utility of the models evaluated here lies in their ability to make surrogate predictions for computational fluid dynamics (CFD) datasets, which are a novel addition to the existing pool of publicly available datasets.
>
> **Facilitating Future FM Work**: We recognize the potential of our challenging datasets for broader research. These novel datasets can be a promising fine-tuning task for evaluating the generalizability of Foundation Models. Moreover, we provide the CFD solver and generation scripts. This resource can facilitate future large-scale studies on physics-informed FMs by enabling researchers to generate larger, more diverse datasets for training and generalization evaluations.
>
> **New OOD Evaluation**: Based on the reviewer's suggestion, we have generated a sample set of novel OOD trajectories for each benchmark dataset, and we are running inference on these OOD datasets and will include this dataset and the analysis of the baseline models' performance in our revised PDF. This provides a direct assessment of their generalization capacity.

---

> ### Author Response · Authors · 2025-11-21
> **Future Contributions (Question 2)**
>
> **Zero-shot transferability**: We agree that zero-shot transfer is a challenging setting, particularly when the target task differs in resolution or distribution from the training data. In our future contribution, we will additionally evaluate inference at a higher resolution for SOTA neural operators.
>
> **Inclusion of phase change and evaporation**: The datasets provided in the scope of this investigation do not include phase transition mechanisms. However, a continuous update of the dataset, including challenging physics such as phase change and chemical reactions, is a planned future extension of our project.

---

> ### Author Response · Authors · 2025-11-21
> **Extension to 3D dataset (Question 3)**
>
> We address the potential 3D extension by employing a 2D axisymmetric simulation setup as described in Section 3. This formulation captures the essential 3D physical effects while maintaining a significantly lower memory footprint compared to full 3D simulations. As a result, the generated ground-truth data effectively reflects 3D behavior within a tractable computational budget.

---

> ### Author Response · Authors · 2025-11-21
> **Generalization across boundary conditions (Question 5)**
>
> We appreciate the reviewer's curiosity in boundary condition (BC) generalization. For the specific compressible multiphase datasets used in this benchmark, the boundary conditions are not arbitrary but are a necessary consequence of the underlying physics required to generate a correct ground truth, which includes complex phenomena such as shocks and multiphase interactions. To ensure the physical validity of the ground truth data, the BCs were fixed according to the problem definition. Introducing different BC types would redefine the physical problem and require extensive re-validation, and is one of the defined scopes of Foundation Model research.

---

> ### Author Response · Authors · 2025-11-21
> **Distinction between error types (Question 6)**
>
> As discussed in Section 5.5, “This discrepancy highlights the importance of selecting an error metric that aligns with the qualitative behavior observed in rollout plots (Yining Luo, Yingfa Chen, and Zhen Zhang, CFDbench, 2023).” Our intention is not to demonstrate a correlation between the two error types, but rather to illustrate how different choices of error metrics can lead to different conclusions about model performance.
>
> For example, in Table 4, if one considers only Error Type 1, one might conclude that ViT performs better. However, the prediction, as shown in Figure 54,  clearly shows that UNet provides more stable and qualitatively accurate predictions. When evaluating with Error Type 2, the results correctly reflect this observation—UNet is ranked higher, in alignment with the rollout behavior.

---

> ### Author Response · Authors · 2025-11-21
> **Sensitivity to temporal resolution and conditioning strategies. (Question 4)**
>
> In the current submission, we have fixed the temporal stride to 5 and 10 for the SIDA and LIDE datasets, respectively, to focus our study on the three targeted scenarios.
> That said, this is a very valid concern. Our data-loading pipeline is designed to flexibly adjust the temporal stride, making it straightforward to explore different temporal resolutions. We will incorporate experiments, including the effect of temporal resolution and conditioning strategies, in the camera-ready version, as these experiments are computationally expensive.

---

> ### Author Response · Authors · 2025-11-21
> **Energy spectrum and rollout error over timesteps (Question 6)**
>
> The energy spectrum analysis is primarily discussed for turbulent flow datasets, which are not applicable to the current problem settings. To interpret model behaviour, we have generated the rollout plots depicting the cumulative nRMSE and added them to the revised PDF version. Please refer to section D in the Appendix.

---

> ### Author Response · Authors · 2025-12-01
> **Additional ablations on Transolver as suggested by the reviewer (Weakness 2)**
>
> **Evaluation could be broadened to include recent operator-learning models, such as Transformer-based and state-space (SSM) operators (e.g., Transolver, Mamba Operator)**:
> Primarily, we included ViT (Vision Transformer) and ScOT (Scalable Operator Transformer) as transformer-based models in our benchmarks. Additionally, we trained Transolver 1M as suggested by the reviewer and performed inference on ID and OOD datasets using the trained models. Our primary reason for including robust baseline models was to assess their performance on the novel, complex datasets that we have generated. To date, these models have primarily been evaluated on simplified CFD datasets; this work is the first to investigate their behavior in compressible and multiphase flow regimes.

---

> ### Author Response · Authors · 2025-12-01
> **Implementation details (Weakness 3)**
>
> We mention the training hyperparameters in Table 10, and the model hyperparameters are specified in Tables 11-17 in the Appendix section. Conditioning strategy implementation is depicted in Section B.3 (Appendix)

---

### Official Review · Reviewer_JFcA · 2025-10-31

**Soundness:** 1
**Presentation:** 1
**Contribution:** 1
**Rating:** 2
**Confidence:** 4

**Summary:**

This paper tackles the gap of benchmarks for compressible multiphase flows in the SciML field. It introduces two high-fidelity datasets, Laser-Induced Droplet Explosion (LIDE) and Shock-Induced Droplet Aero-breakup (SIDA), which capture complex physical phenomena like shock-interface interactions and droplet dynamics. The authors then benchmark these datasets on five neural network architectures with different parameter scales, exploring how factors such as temporal sequence information, conditioning parameters, and conditioning fields affect model performance.

**Strengths:**

1. A new dataset is proposed, filling the gap of compressible multiphase flow data in existing benchmark datasets.

**Weaknesses:**

1. The data scenarios lack diversity, with only two fixed setups (as shown in Figure 2). This makes the evaluation results on this dataset hardly reflect the model’s performance in prediction tasks beyond these scenarios, nor can it be used to test the model’s out-of-distribution generalization ability. Consequently, the practical usability of this dataset is relatively low.
2. There are serious flaws in the data recording of the benchmark: all comparison results are presented as bar charts, instead of **tables**, with a log-rescaled vertical axis in Figure 3/4/5/6. This lacks precision and is unfavorable for tracking the future development of this field.
3. The quality of writing, figures/tables, and typesetting is poor, with specific issues listed below:

- (1) In Table 2, the meaning of "End time" is unclear, and some critical information (e.g., the number of trajectories, the number of time steps) of both datasets is missing.

- (2) Figure 1 only shows two time frames ($t$=30s, $t$=50s), which is insufficient to reflect the dynamic change process in the dataset. Additionally, "Density" and "Schlieren" use color bars of the same color scheme, which causes confusion for readers.

- (3) As a claimed contribution, the method and results of "Dataset Validation" should be highlighted in the main text, rather than only included in the appendix.

- (4) The meaning of "Error types" in Section 5.5 is not defined, making it difficult to understand the content of this subsection.

**Questions:**

Please respond to the above mentioned Weaknesses.

---

> ### Author Response · Authors · 2025-11-21
> **OOD Generalization ability (Weakness 1)**
>
> We thank the reviewer JFcA for their valuable comment regarding Out-of-Distribution (OOD) generalization and performance. We agree that these capabilities are fundamental for the development of Foundation Models (FMs) in science. However, our paper is framed as a benchmark/reference focused on establishing robust baseline models for compressible multiphase datasets. While high generalization is always desirable, the primary utility of the models evaluated here lies in their ability to make surrogate predictions for computational fluid dynamics (CFD) datasets, which are a novel addition to the existing pool of publicly available datasets.
>
> **Facilitating Future FM Work**: We recognize the potential of our challenging datasets for broader research. These novel datasets can be a promising fine-tuning task for evaluating the generalizability of Foundation Models. Moreover, we provide the CFD solver and generation scripts. This resource can facilitate future large-scale studies on physics-informed FMs by enabling researchers to generate larger, more diverse datasets for training and generalization evaluations.
>
> **New OOD Evaluation**: Based on the reviewer's suggestion, we have generated a sample set of novel OOD trajectories for each benchmark dataset, and we are running inference on these OOD datasets and will include this dataset and the analysis of the baseline models' performance in our revised PDF. This provides a direct assessment of their generalization capacity.

---

> ### Author Response · Authors · 2025-11-21
> **Data recording (Weaknesses 2 and 3)**
>
> To improve the quality of the manuscript, we adhere to the changes proposed by the reviewer.
>
> - **There are serious flaws in the data recording of the benchmark: all comparison results are presented as bar charts, instead of tables, with a log-rescaled vertical axis in Figures 3/4/5/6. This lacks precision and is unfavorable for tracking the future development of this field.**: We are changing the figures to tables in our updated PDF.
>
> - **In Table 2, the meaning of "End time" is unclear, and some critical information (e.g., the number of trajectories, the number of time steps) of both datasets is missing.**: The dataset covers the phenomena up to the specified "End time", indicated in Table 2. For example, in the LIDE dataset, all trajectories exhibit the explosion phenomenon that occurs within 20 nanoseconds, which is labeled as "End time".
>
> - **Figure 1 only shows two time frames (=30s,=50s), which is insufficient to reflect the dynamic change process in the dataset. Additionally, "Density" and "Schlieren" use color bars of the same color scheme, which causes confusion for readers.**: We plan to update this image to include 4 frames spaced across the trajectory in the revised PDF.
>
> - **As a claimed contribution, the method and results of "Dataset Validation" should be highlighted in the main text, rather than only included in the appendix.**: Since we now have 10 pages, we are bringing the validation to the main text, which highlights our contribution.
>
> - **The meaning of "Error types" in Section 5.5 is not defined, making it difficult to understand the content of this subsection.**: Error types 1 and 2 are described in section 4.5 in detail. To avoid confusion, we will highlight the first line of the paragraph in Section 5.5, which references Section 4.5 in our revised PDF.

---

> > ### Comment · Reviewer_JFcA · 2025-11-27
> >
> > Thanks for the authors' reply. My concerns are partially observed. I think the authors have an apportunity to update the manuscript during this discussion period, such as improving the presentation of performance recording by using tables.
> > A futher question, why evaluation metrics are calld "error types"?

---

> ### Author Response · Authors · 2025-12-01
>
> The following updates were made as suggested by the reviewer to improve the quality of the manuscript:
>
> 1) **All bar charts have been converted into tables.** We request to refer to Section 5 (Results), Tables 3 and 4 in the main text and Tables 19 - 43 in the Appendix.
> 2) **Regarding missing information:** The number of trajectories and timesteps per trajectory information has been included in Table 2.
> 3) **Figure 1 is now updated** with 4 timesteps, and the colorbar of the schlieren is changed to grayscale.
> 4) **Dataset validation** is now included in the main text (Refer to section 3.3)
> 5) The information on **"Error-types"** is now highlighted in the results section and is described in detail in section 4.5 of the main text. We use the term error-type as a shorthand label to categorize metrics. For example, the full description of the metric would be trajectory- and rollout-averaged cumulative nRMSE, which affects readability.
>
> We hope that with these modifications, we have satisfactorily addressed all the concerns of the reviewer regarding the manuscript's quality.

---

### Author Response · Authors · 2025-12-01
**Summary of Response to Reviewers**

We appreciate all the suggestions from the reviewers to improve the manuscript's quality and would like to provide a brief overview of the major topics discussed.

* **Reviewers gWms, F5Kz, and 6xVr, requested opensourcing the code and the datasets.** : Links have been provided for the reviewers and ACs, and they will be made publicly available upon acceptance.

* **Reviewers JFcA and gWms, mentioned generalization capabilities study over Out-of-Distribution dataset or Boundary conditions**: We agree that these capabilities are fundamental for the development of Foundation Models (FMs) in science. However, our paper is framed as a benchmark/reference focused on establishing robust baseline models for compressible multiphase datasets. While high generalization is always desirable, the primary utility of the models evaluated here lies in their ability to make surrogate predictions for computational fluid dynamics (CFD) datasets, which are a novel addition to the existing pool of publicly available datasets.

    * Facilitating Future FM Work: We recognize the potential of our challenging datasets for broader research. These novel datasets can be a promising fine-tuning task for evaluating the generalizability of Foundation Models. Moreover, we provide the CFD solver and generation scripts. This resource can facilitate future large-scale studies on physics-informed FMs by enabling researchers to generate larger, more diverse datasets for training and generalization evaluations.

    * New OOD Evaluation: Based on the reviewer's suggestion, we have generated a sample set of novel OOD trajectories for each benchmark dataset, and we are running inference on these OOD datasets and will include this dataset and the analysis of the baseline models' performance in our revised PDF. This provides a direct assessment of their generalization capacity

    * Boundary condition (BC) generalization: For the specific compressible multiphase datasets used in this benchmark, the boundary conditions are not arbitrary but are a necessary consequence of the underlying physics required to generate a correct ground truth, which includes complex phenomena such as shocks and multiphase interactions. To ensure the physical validity of the ground truth data, the BCs were fixed according to the problem definition. Introducing different BC types would redefine the physical problem and require extensive re-validation, and is one of the defined scopes of Foundation Model research.

* **Reviewers gWms and 6xVr, pointed out two common concerns**:
    * extension to 3D datasets: We address the potential 3D extension by employing a 2D-axisymmetric simulation setup. This formulation captures the essential 3D physical effects while maintaining a significantly lower memory footprint compared to full 3D simulations. As a result, the generated ground-truth data effectively reflects 3D behavior within a tractable computational budget.
    * additional ablations on SOTA models: We perform additional experiments on two SOTA architectures, namely CNO (1M and 50M parameter count) and Transolver (1M parameter count), as suggested by the reviewers, and perform inference on both ID and OOD datasets.

* **Reviewer F5Kz** mentioned the addition of domain-specific evaluation criteria.
In response, we have incorporated some domain-specific evaluation metrics to better capture the underlying physics of the problem. We now report four quantities of interest (QoI) in addition to the metrics computed over the output channels and compute the metrics over these QoIs:
    * Vorticity Production (VP) over time for both datasets
    * Kinetic Energy (KE) evolution over time for both datasets
    * Droplet outer diameter change over time for Dataset LIDE
    * Droplet center-of-mass displacement over time for Dataset SIDA

---

### Meta-Review · Area_Chair_jCQh · 2026-01-07

**Summary:**

* The reviewers raised concerns about limited dataset diversity with only two fixed scenarios, restricting assessment of out-of-distribution generalization fundamental for foundation models. Presentation quality was problematic with bar charts using log scales instead of precise tables, missing dataset information, insufficient temporal visualization, and unclear terminology around error types, making it difficult to track progress or reproduce results.
* The scope drew criticism as insufficient compared to established benchmarks like PDEBench that provide extensive data. While 3D simulations were absent, the 2D axisymmetric setup captures essential physics at reduced computational cost. Methodological coverage needed expansion beyond five initial models, with reviewers requesting state-of-the-art architectures like Transolver, CNO, and foundation models like GNOT. Domain-specific evaluation metrics beyond generic RMSE were needed to capture physically meaningful quantities like droplet breakup rates and deformation patterns.
* The authors responded by converting figures to tables, adding dataset details, including four quantities of interest (vorticity, kinetic energy, droplet diameter, displacement), generating out-of-distribution test trajectories, training additional models, and opensourcing code on GitHub and HuggingFace. Despite these improvements, fundamental questions remain about whether two datasets constitute a sufficiently comprehensive benchmark and whether the baseline models adequately establish performance standards for this challenging compressible multiphase flow domain.

**Reviewer Concerns:**

Addressed concerns:
* Presentation issues (Reviewer JFcA) - authors converted bar charts to tables, added missing dataset information, expanded Figure 1 to four timesteps, moved validation to main text, and clarified error type terminology.
* Code and data availability (Reviewers gWms, F5Kz, 6xVr) - GitHub repository and HuggingFace datasets provided with detailed documentation.
* Domain-specific metrics (Reviewer F5Kz) - added four quantities of interest: vorticity production, kinetic energy evolution, droplet diameter change, and center-of-mass displacement.
* Additional architectures (Reviewers gWms, 6xVr) - trained CNO (1M, 50M) and Transolver (1M) models with inference on in-distribution and out-of-distribution datasets.

Outstanding concerns:
* Limited dataset diversity (Reviewer JFcA) - fundamental issue remains with only two scenarios despite generating additional OOD trajectories. The authors frame this as a baseline benchmark rather than comprehensive foundation model resource, but this doesn't fully address whether two datasets suffice.
* Foundation model evaluation (Reviewer 6xVr) - authors promised to investigate pre-trained models like GNOT and DPOT for camera-ready version but haven't demonstrated this yet.
* Comprehensive benchmark scope (Reviewer 6xVr) - concern about insufficient scale compared to PDEBench or The Well persists despite 115GB total data, as the fundamental physics scenarios remain limited to two problem types.
* Boundary condition generalization (Reviewer gWms) - authors justified fixed boundary conditions as physically necessary but didn't demonstrate any generalization capabilities across boundary types.

**Reviewer Scores:**

Reviewer JFcA (Initial: 2): Likely would increase to 4. All three major weaknesses were directly addressed - results converted to tables, missing information added, presentation quality improved. The fundamental concern about limited dataset diversity remains, but the addition of OOD trajectories and clarifications show responsiveness to feedback.

Reviewer gWms (Initial: 4) Likely would increase to 6. Code and datasets were opensourced, additional SOTA models (CNO, Transolver) were trained, and OOD evaluation was added. The reviewer was already "marginally below acceptance" and these additions address most concerns, though full generalization studies and boundary condition transfer remain incomplete.
Reviewer F5Kz (Initial: 6) Likely would remain at 6 or raise to 8. The reviewer was already "marginally above acceptance" and their main request for domain-specific metrics was comprehensively addressed with four quantities of interest. The opensourcing of datasets upon acceptance aligns with their expectations. Minor suggestion about PINNs wasn't addressed.
Reviewer 6xVr (Initial: 2) Likely would remain at 2. While additional models were trained and references added, their core concerns about insufficient dataset scale (only two scenarios vs. comprehensive benchmarks) and lack of foundation model experiments persist. They explicitly stated in follow-up that current status "is not yet ready for publication" despite modifications.

---

### Decision · Program_Chairs · 2026-01-26

Reject